# Recent ground thermo-hydrological changes in a Southern Tibetan endorheic catchment and implications for lake level changes

Léo C.P. Martin[1,2,3], Sebastian Westermann[2,4], Michele Magni[1], Fanny Brun[1,5], Joel Fiddes[6], Yanbin Lei[7,8], Philip Kraaijenbrink[1], Tamara Mathys[9], Moritz Langer[10,11], Simon Allen[12] and Walter W. Immerzeel[1]

1. Faculty of Geosciences, Utrecht University, Utrecht, The Netherlands
2. Department of Geosciences, University of Oslo, Blindern, 0316 Oslo, Norway
3. Aix Marseille Univ, CNRS, IRD, INRAE, CEREGE, Aix-en-Provence, France
4. Center for Biogeochemistry in the Anthropocene, Oslo, Norway
5. Université Grenoble Alpes, CNRS, IRD, Grenoble INP, IGE, Grenoble, France
6. WSL Institute for Snow and Avalanche Research SLF, Davos, Switzerland
7. Key Laboratory of Tibetan Environment Changes and Land Surface Processes, Institute of Tibetan Plateau Research, Chinese Academy of Sciences, Beijing 100101, China
8. CAS Center for Excellence in Tibetan Plateau Earth System Sciences, Beijing 100101, China
9. Department of Geosciences, University of Fribourg, Fribourg, Switzerland
10. Alfred Wegener Institute Helmholtz Centre for Polar and Marine Research, 14473 Potsdam, Germany
11. Department of Geography, Humboldt Universität zu Berlin, 12489 Berlin, Germany
12. Department of Geography, University of Zurich, Zürich, Switzerland

*Correspondence to:*    Léo Martin        (leo.doug.martin@gmail.com)
                         Walter Immerzeel    (w.w.immerzeel@uu.nl)

Abstract:    421 words
Main:        13,981 words, 11 figures, 2 tables
Appendices:  446 words, 3 figures, 1 table

# Abstract

Climate change modifies the water and energy fluxes between the atmosphere and the surface in mountainous regions such as the Qinghai-Tibet Plateau (QTP), which has shown substantial hydrological changes over the last decades, including rapid lake level variations. The ground across the QTP hosts either permafrost or seasonally frozen and, in this environment, the ground thermal regime influences liquid water availability, evaporation and runoff. Consequently, climate-induced changes in the ground thermal regime may contribute to variations in lake levels, but the validity of this hypothesis has yet to be established.

This study focuses on the cryo-hydrology of the catchment of Lake Paiku (Southern Tibet) for the 1980-2019 period. We use TopoSCALE and TopoSUB to downscale ERA5 data, in an effort to account for the spatial variability of the climate in our forcing data. We use a distributed setup of the CryoGrid community model (version 1.0) to quantify thermo-hydrological changes in the ground during this period. Forcing data and simulation outputs are validated with data from a weather station, surface temperature loggers and observations of lake level variations. Our lake budget reconstruction shows that the main water input to the lake is direct precipitation (310 mm per year), followed by glacier runoff (280 mm per year) and land runoff (180 mm per year). However, altogether these components do not offset evaporation (860 mm per year).

Our results show that both seasonal frozen ground and permafrost have warmed (0.17 °C per decade 2 m deep), increasing the availability of liquid water in the ground and the duration of seasonal thaw. Correlations with annual values suggest that both phenomena promote evaporation and runoff. Yet, ground warming drives a strong increase in subsurface runoff, so that the runoff/(evaporation + runoff) ratio increases over time. This increase likely contributed to stabilizing the lake level decrease after 2010.

Summer evaporation is an important energy sink and we find active layer deepening only where evaporation is limited. The presence of permafrost is found to promote evaporation at the expense of runoff, consistent with recent studies suggesting that a shallow active layer maintains higher water contents close to the surface. However, this relationship seems to be climate-dependent and we show that a colder and wetter climate produces the opposite effect. Although the present study was performed at catchment scale, we suggest that this ambivalent influence of permafrost may help to understand the contrasting lake level variations observed between the South and North of the QTP, opening new perspectives for future investigations.

# Main text

## 1. Introduction

Climate change is amplified in mountainous environments, with major consequences for ecosystems, landscapes, hydrology, human communities and infrastructure (IPCC, 2019). Station observations show that global warming is elevation dependent, with the strongest warming rates being observed at high elevations (Pepin et al., 2015; Wang et al., 2014). Over the Qinghai-Tibet Plateau (QTP), a significant increase in surface air temperatures has been recorded since the 1980s, in particular in the North of the plateau (Zhang et al., 2022). This has been accompanied by a decrease in wind speed, humidification of the air, and a general increase in precipitation, although with a strong spatial variability (Bibi et al., 2018). Altogether, these changes have affected the surface energy balance of the plateau through a shift of the Bowen ratio towards more latent heat fluxes, limiting the sensible surface warming (Yang et al., 2014a).

These changes in water and energy fluxes between the atmosphere and the surface have the potential to alter the hydrological cycle of the QTP, which is the headwater region for major Asian rivers. As such, increasing trends of evaporation over land have been measured (3.8 mm per decade since the 1960s) with strong spatial variability both in absolute values and increase rates (Wang et al., 2020b). Changes in the seasonality of river discharge (Cao et al., 2006) and groundwater discharge (Niu et al., 2011) were reported for the same period. Overall glacier shrinkage has also been observed since the 1960s with a persistent increase in glacier mass loss rates (Bhattacharya et al., 2021; Hugonnet et al., 2021).

The QTP also features more than 1,000 lakes larger than 1 km$^2$ (Zhang et al., 2017), most of them located in endorheic catchments. Lake volume changes are therefore attributable to climatic and hydrological changes occurring within the lake catchment, such as glacier melt, ground ice melt, precipitation, evaporation or runoff patterns. The majority of these lakes have experienced a pronounced increase in water levels since the 1990s (Lei et al., 2013, 2014), a trend that was suggested to be mainly driven by changes in precipitation and evaporation patterns (Yao et al., 2018) rather than by an increase

in glacier mass loss and runoff (Brun et al., 2020; Zhang et al., 2021a). Nevertheless, lake level
variations are not uniform across the QTP and exhibit important spatial variability. Whereas the
northern and central QTP have recorded lake expansion, the southern parts of the plateau have
experienced lake shrinkage (Qiao et al., 2019; Zhang et al., 2021a, 2020a). Shrinking lakes have
received less attention in the literature than rising lakes because they are fewer. For this reason the
drivers of this shrinkage are still unclear. Qiao et al. (2019) reported that recent lake shrinkage over the
QTP could be driven by local precipitation decrease and/or evaporation increase (in relation to air
temperature increase). Zhang et al. (2020a) suggests that the divergent trends in lake level variations
across the QTP could be linked to the contrasting evolution of moisture transport between the north and
south of the plateau. On longer timescales, lake shrinkage over the QTP during the Holocene seems to
be related to variations in the intensity of the Asian monsoon (Chen et al., 2013). Overall, such a
complex pattern of rising and shrinking lakes challenges our understanding of the hydrological changes
occurring in these high Asian watersheds.

In this regard, new insights on hydroclimatic changes over the QTP can emerge from the

investigation of the coupled energy and water fluxes between the ground surface/subsurface and the
atmospheric boundary layer. These fluxes are driven by the climate and have a major impact on cold-
region hydrology (Pomeroy et al., 2007; Gao et al., 2021; Bring et al., 2016). Indeed, hydrological
variables (precipitation, evaporation, runoff) affect the soil water content, which changes its thermal
properties, the distribution between latent and sensible fluxes and thus substantially influences the
ground thermal regime (Bring et al., 2016; Koren et al., 1999; Martin et al., 2019). In turn, the ground
thermal regime modifies the relative proportion of frozen and liquid subsurface water, influencing
infiltration possibilities and the amount of water available for evaporation and surface/subsurface runoff
(Yi et al., 2006; Carey and Woo, 2001).

So far, climate induced thermo-hydrological changes over the QTP have received limited attention.

Large-scale modeling studies reported changes in the seasonal ground freezing cycles characterized by
a reduction of the frost depth and duration of the frozen period since the 1960s (Qin et al., 2018; Wang
et al., 2020a) and notable ground warming trends in summer and winter (Qin et al., 2021). Similar
ground warming trends were reported in the regional modeling study from Qin et al. (2017), along with
an increasing trend in evaporation and a decrease of the runoff coefficient over time. Plateau-scale
surface energy balance modeling from Wang et al. (2020b) reported that increasing trends in
evapotranspiration could be mainly explained by variations in air temperature and net radiation at the
surface.

Complementary to seasonally frozen ground, permafrost is also a distinctive feature of climate-

surface interactions in cold regions. Large-scale permafrost modeling suggests that it covers a
significant part of the QTP, mainly as continuous permafrost in the north of the plateau and as
discontinuous or sporadic in the south (Obu et al., 2019). Permafrost on the QTP usually has a low ice
content due to limited precipitation and strong evaporation (Wu et al., 2005; Yang et al., 2010).
Borehole temperature measurements show that it is a relatively warm type of permafrost (Biskaborn et
al., 2019; Wu and Zhang, 2008) and its exposure to high solar radiations makes it sensitive to changes
in surface conditions and climate change (Yang et al., 2010). Since the 1960s, climate change has driven
permafrost warming across the plateau (Ran et al., 2018; Shaoling et al., 2000). Ran et al. (2018) reports
that most of the plateau exhibits a warming trend of the ground comprised between 0.26 and 0.74 C
per decade and half of the plateau warms at a rate higher than 0.5 °C per decade. This warming is
accompanied by upward migration (of around 100 m between the 1960s and 2000s) and shrinkage of
permafrost covered areas (24% of the permafrost extent lost between the 1960s and the 2000s, Ran et
al., 2018).

Permafrost grounds are characterized by a strong interplay between the ground thermal regime and

the land hydrology. Seasonal thawing and freezing of the active layer are driven by the surface energy
balance which, in return, influences surface and subsurface runoff (Kurylyk et al., 2014; Walvoord and
Kurylyk, 2016; Sjöberg et al., 2021) and evaporation (Gao et al., 2021). In this regard, both large-scale
and regional modeling indicate that thawing permafrost enhances evapotranspiration (Qin et al., 2017;
Wang et al., 2020b). Qin et al. (2017) also report that the increase in evaporation is logically
concomitant with a decrease in the runoff coefficient. Additionally, permafrost stores water as ground
ice and its thawing can trigger the release of liquid water in the watershed, contributing up to 15% of
the annual river streamflow (Cheng and Jin, 2013; Yang et al., 2019).
These hydrological changes are tied to various interdependent climate-driven physical processes
happening at the ground surface and subsurface (e.g. surface energy balance, infiltration, water phase
change, heat conduction...). Because these processes exhibit a strong spatial variability in high mountain
environments, it is challenging to represent them accurately together on large spatial scales. Therefore,
a deeper understanding of the impact of ground thermo-hydrological changes on the High Asia water
cycle can be gained through small-scale physical modeling of these processes. Yet, for now, physics-
based approaches at the catchment scale aiming to connect the ground thermo-hydrological regime and
the observed hydrological changes on the QTP (such as lake level changes) remain scarce. They are
however a powerful approach to tackle the question: how much might climate-driven ground thermal
changes affect the water cycle in high mountain headwater regions? In this study, we use physical land
surface modeling to quantify the ground thermo-hydrological changes in an endorheic Tibetan
catchment over the last 40 years as a response to climate change. We show the interplay in the water
and energy fluxes occurring between the atmosphere, the surface and the subsurface and discuss their
impact on the hydrology of the catchment and their implication regarding lake level variations.

## 2. Study area: the Paiku catchment

The Paiku catchment is located in south-western Tibet, China, close to the border with Nepal (28.8°N - 85.6°E, Fig. 1). Its southern edge lies 7 km from the Shishapangma peak (8027 masl). The catchment is endorheic and spans over 78 km from North to South, 66 km from East to West and covers 2 400 km$^2$. The median elevation of the catchment is 4872 masl, ranging from 7272 masl to its lowest point, lake Paiku at 4580 masl. Geologically, the catchment is mainly located in the Tethys Himalayan, and thus, an important part of the formations underlying the catchment are metamorphized sedimentary series (Appendix B, Fig. B1). The southern part of the catchment crosses the Southern Tibetan Detachment, and thus, the southern ridges of the massif belong to the High Himalayan metamorphic formations in the west and to the High Himalayan leucogranites of the Shishapangma massif on the east. The north and north-east ridges are formed by granite intrusions surrounded by metamorphic domes. The inner part of the catchment presents Plio-Quaternary formations such as alluvial fans close to the ridges and inclined alluvial plains in its inner parts (Aoya et al., 2005; Searle et al., 1997; Wünnemann et al., 2015).

Automatic Weather Station (AWS) observations (5033 masl, Oct 2019 – Sept 2021, Fig. 1) show that the climate in the catchment is characterized by a relatively small temperature amplitude during the year (around 20 °C, JJA being the warmest months and DJF the coldest) and significant daily amplitude (up to 10 °C during the warm season). The mean annual temperature is -1.5 °C at the AWS, where night freezing can occur until the beginning of June and resume at the beginning of October. The catchment is dry (200-300 mm year$^{-1}$) and precipitation mostly falls as rain during the monsoon (JJAS).

Around 5% of the catchment is covered by glaciers (RGI Consortium, 2017), which are concentrated in its southwestern part. They feed several proglacial lakes that can reach up to 6 km in length. Geodetic glacier mass budgets show that, similar to other glaciers in the region, glaciers of the Paiku catchment have undergone sustained mass loss at least since the 1970s, with an average mass balance of -0.3 m w.e.a$^{-1}$ until the beginning of the 2000s and around -0.4 m w.e.a$^{-1}$ thereafter (Bhattacharya et al., 2021). There are more than 10 rivers that drain the catchment towards the lake and most of them only exhibit a seasonal activity during the monsoon months. The three main ones are

(Fig. 1), Daqu (glacier-fed, 450 km$^2$), Bulaqu (glacier-fed, 325 km$^2$) and Barixiongqu (non-glacier-fed,
703 km$^2$, Lei et al., 2018).
In the north-west of the catchment, Lake Paiku covers approx. 280 km$^2$ (11.5% of the catchment
surface area) and spans over 27 km from North to South. It has a mean water depth of 41 m, with a
maximum water depth of 73 m (Lei et al., 2018). It receives water from direct precipitation and from
land and glacier runoff which can be routed at the surface via the river systems or the subsurface via
the alluvial formations. Because it is hydrologically closed, the lake mainly loses water through
evaporation. Previous studies reported lake level fluctuations over different time scales. It reached 4665
masl (85 m higher than the present level) prior to 25 ka BP and at the onset of the Holocene (11.9-9.5
ka BP), afterwards, the lake shrank gradually (Wünnemann et al., 2015). More recently, the lake level
decreased by 3.7 m between 1972 and 2015, losing 4.2% of its surface and 8.5% of its volume.
Measurements have been performed since the end of the 1970s and allow to accurately know the
evolution of the lake level until today (Lei et al., 2021, 2018), they are used in this study to validate our
hydrological results (Sect 3.2.1, Fig. 5D and 6B). At the seasonal scale, the lake level cycle has an
amplitude of ~ 0.4 m. It is marked by a strong increase during the monsoon period (JJAS) supported by
direct precipitation, glacier melt and land runoff. From October and until the next monsoon period,
evaporation dominates the lake mass budget and the level decreases rapidly until January and at a slower
rate afterwards (Lei et al., 2021).

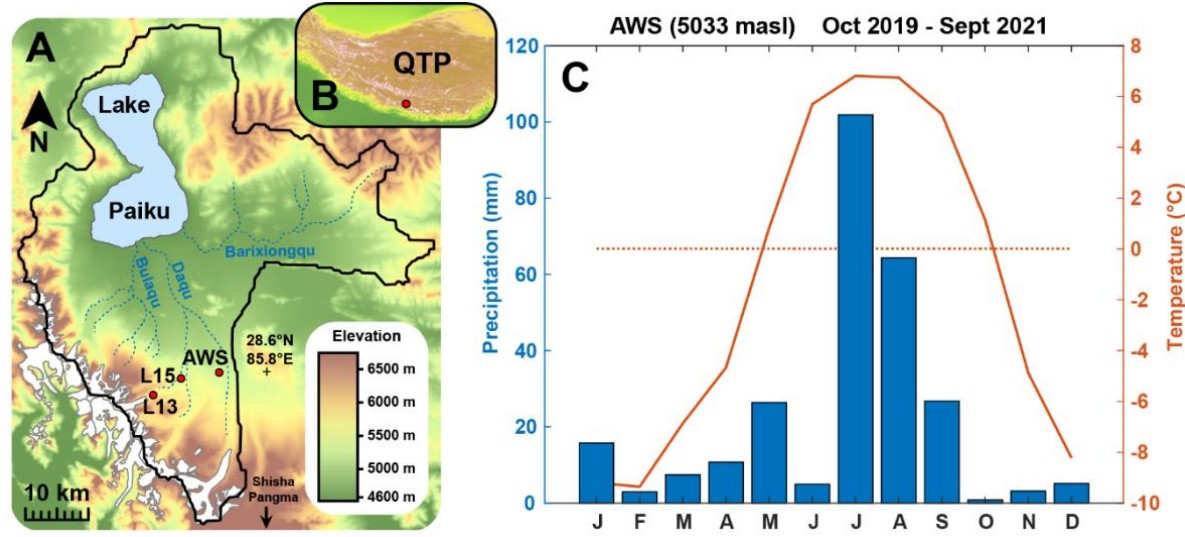

*Figure 1. The Paiku Catchment. A: Topographic and hydrologic map of the catchment with the glaciers*
*in white, the ephemeral rivers in dark blue and the lake in light blue (elevation: SRTM data courtesy of*
*the U.S. Geological Survey). AWS: Automatic Weather Station. L13 and L15 are surface temperature*
*loggers (Sect. 3.1.). B: Localization of the catchment over the QTP. C: Monthly temperature and*
*precipitation recorded at the AWS between October 2019 and September 2021.*

# 3. Material and methods

## 3.1.    Field measurements

An AWS was set up in October 2019 in the South of the catchment at an elevation of 5033 masl (Fig. 1). It is equipped with various sensors which record air temperature, pressure, relative humidity, wind speed, incoming and outgoing long and short wave radiations and precipitation every 15 minutes. The meteorological record extends to September 2021 and covers a period of nearly 2 years. We used it to evaluate and correct the distributed downscaled climatic forcing on which we rely in our modeling framework (Sect. 3.2.2.).

Two temperature loggers recorded the surface temperature in the vicinity of the AWS location. Logger 15 (L15) is located at 5055 masl, 6 km west of the AWS. Logger 13 (L13) is located at 5356 masl, 12 km west of the AWS (Fig. 1). Both loggers were buried 10 to 15 cm below the surface to avoid direct solar radiation on the sensors and recorded surface temperature at a 20-minute timestep from October 2017 to October 2018. These surface temperature records were used to evaluate the simulations (Sect. 3.2.4.).

## 3.2.    Catchment thermo-hydrological modeling

### 3.2.1. Conceptual hydrological model for the catchment

To understand the level variations of lake Paiku over the last 40 years (1980-2019 period), we develop an approach at the catchment scale. Because the catchment is hydrologically closed, the lake receives water input via direct precipitation, land surface and subsurface runoff, and glacier runoff. Conversely, it loses mass via evaporation. Because the quantification of water flows between the lake and potential aquifers surrounding it is difficult (Rosenberry et al., 2015), our approach assumes that these flows are negligible. The present study requires quantification of the different terms of the hydrological balance. Under these assumptions, the hydrological balance of the lake is given by the following equation:

$\Delta z_{Lake} = Precipitation_{Lake} + Runoff_{LandSurf} + Runoff_{LandSub} + Runoff_{Glacier} - Evaporation_{Lake}$
The production of forcing data for the catchment (including precipitation) is detailed in Sect. 3.2.2.
The land hydrology processes are quantified using the CryoGrid community model (version 1.0)
(Westermann et al., 2023) as described in Sect. 3.2.3. Distributed 1D simulations are used to quantify
land evaporation and runoff. The routing of water in the catchment is not represented and the runoff
computed for a given simulation is directly accounted as a water input for the lake. The evaporation
from the lake is simulated using the CryoGrid3-Flake model (Langer et al., 2016) as described in
Sect. 3.2.5. Glacier melt is not modeled, but estimated for the study period (1980-2019) from remote
sensing observations. From these observations, glacier yield is calculated as described in Sect. 3.2.6.
Our catchment-scale approach to represent the hydrological balance of the lake is summarized in Fig. 2.
Based on this approach, we can evaluate the performance of our framework (Sect. 4.1.2.), by comparing
the simulated lake balance with the one derived from the detailed observations of lake level variations
over the study period (Lei et al., 2018, 2021).

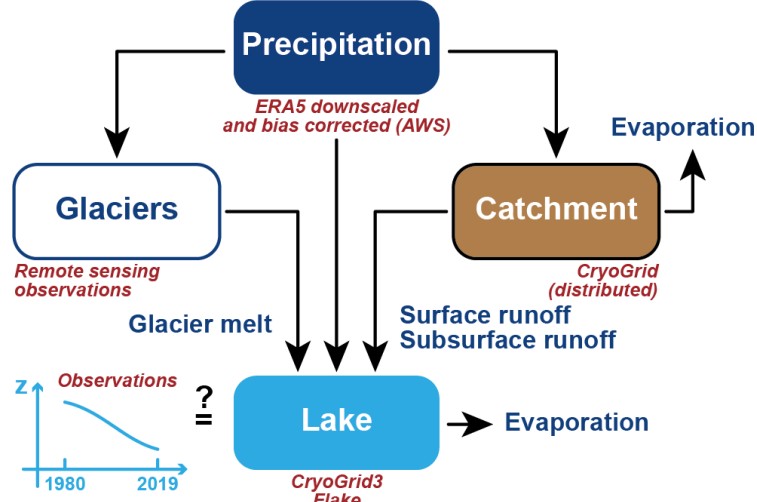

*Figure 2. Conceptual hydrological framework for the study.*
*3.2.2.   Forcing data production and validation*
In high mountain environments, topography creates strong spatial variability of temperature and
incoming radiation, which impact the surface energy balance (Klok and Oerlemans, 2002) and the
ground thermo-hydrological regime (Magnin et al., 2017). Our approach requires forcing data that (i)
captures this variability, (ii) includes numerous variables such as air temperature, incoming long and
short wave radiations, wind speed, specific humidity, rain and snowfall and (iii) covers the 40 years
study period at a sub-daily timestep. The TopoSCALE approach (Fiddes and Gruber, 2014) was
developed for this purpose and allows to downscale reanalysis products like ERA5 (Hersbach et al.,
2020) at high resolution (here ~ 100 x 100 m).
Additionally, because working at a $10^{-2}$ $km^2$ spatial resolution over a 2400 $km^2$ catchment would
require more than 200,000 forcing files and simulations, we rely on the TopoSUB method (Fiddes and
Gruber, 2012) to reduce computational costs. This method uses a SRTM30 Digital Elevation Model to
explore redundancies in physiographic parameters of the study area such as elevation, aspect, slope and
sky-view factor and to identify groups of high-resolution pixels (100 x 100 m) sharing similar values
for these parameters. From there, all the high-resolution pixels belonging to such a group are only
described as a single TopoSUB point, for which climatic variables can be downscaled to create one
single dataset of climatic timeseries. The degree of similarity required by TopoSUB to identify groups
of high-resolution pixels with redundant physiographic parameters can be adjusted by choosing the final
number of TopoSUB points (and thus climate datasets) that should be used to cover the area
corresponding to one ERA5 pixel. The Paiku catchment intersects 8 ERA5 pixels at 30 km resolution
and we chose to use 50 TopoSUB points within each ERA5 pixel to cover the spatial variability created
by the topography on small-scale climate. Ultimately, 368 TopoSUB points are used to cover the
catchment. The average level of redundancy (i.e. the average number of high-resolution pixels
represented by a single TopoSUB point) is $723 \pm 745$ ($1\sigma$, median: 506, min: 1, max: 4347). Appendix
C, Fig. C1 shows the distribution of the TopoSUB points and a reconstruction of the topography of the
catchment based on this approach. The period covered by the forcing datasets starts on $1^{st}$ January 1980
and ends on $31^{st}$ August 2020 (40 years and 8 months).
In the TopoSCALE statistical downscaling approach, we do not rely on the AWS data and thus the
downscaled ERA5 data can be biased, as is often the case over Asia (Jiang et al., 2021, 2020; Jiao et
al., 2021; Orsolini et al., 2019). Comparison against the available AWS observations (Appendix D, Fig.
D1) indeed highlights notable differences in variables such as air temperature and precipitation. From
these differences, we derived monthly bias correction factors that we applied systematically to all of the
368 climate forcing datasets. The catchment averages for precipitation and air temperatures are shown
in Fig. 3. In this figure and across the rest of the study, we use p-values to evaluate the significance of
linear trends in the temporal evolution of certain variables (temperature, precipitation, evaporation…).
This p-value tests the null hypothesis which supposes that the value of the slope is equal to zero. The
hypothesis is tested using the Student's t-test, by comparing the distance between the estimated slope
and 0, relative to the standard error of the slope. We did not report trends when this p-value (probability
of a null slope) was higher than 0.005.

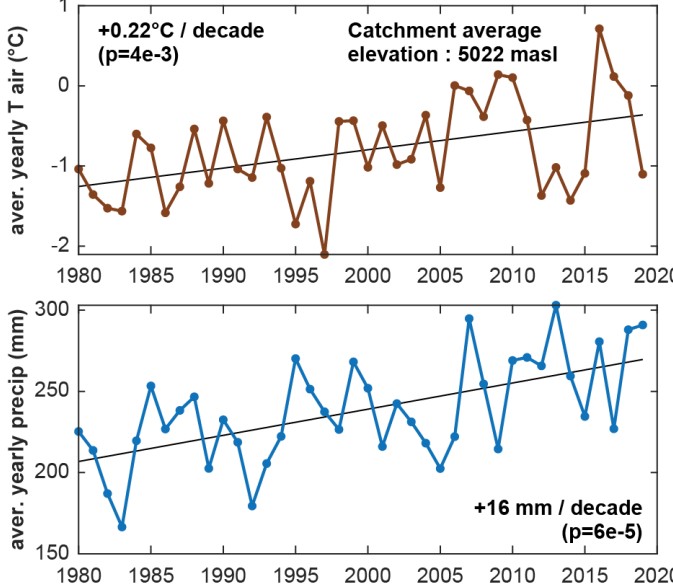

*Figure 3. Climate forcing data for the land and lake modeling. Annual catchment-average air*
*temperature (2 m above ground) and annual total precipitation for the study period. Note that the model*
*is also forced by incoming short and long wave radiations, humidity, windspeed and air pressure.*
*Details about the spatial and temporal resolution of the distributed forcing data are presented in Sect.*
*3.2.2.*
*3.2.3.  The CryoGrid community model (version 1.0)*
To simulate the ground thermo-hydrological regime, we use the CryoGrid community model
(Westermann et al., 2023). The CryoGrid community model (CG) is a land surface model designed for
applications in cold regions where seasonal frozen ground or permafrost may occur. The model
implements heat transfer in a 1D soil column, accounting for freeze-thaw processes of soil water using
effective heat capacity (Nakano and Brown, 1972). To do so, soil freezing curves are based on
Dall'Amico et al. (2011) as detailed in Westermann et al. (2013). Vertical water movement in the soil
column is based on Richards equation (Richardson, 1922; Richards, 1931). The soil matric potential
and hydraulic conductivity follow van Genuchten (1980) and Mualem (1976). Additionally, to represent
the obstruction of connected porosity by ice formation, the hydraulic conductivity is reduced by a factor
dependent on the local ice content, following Dall'Amico et al. (2011). The model features the
snowpack module called *CG Crocus* described in Zweigel et al. (2021) that adapts the snow physics
parameterizations from the CROCUS scheme (Vionnet et al., 2012) to the native snow module of
CryoGrid3 (Westermann et al., 2016). At the surface, the model uses a surface energy balance module
to calculate the ground surface temperature and water content. The turbulent fluxes of sensible and
latent heat are calculated using a Monin–Obukhov approach (Monin and Obukhov, 1954). Evaporation
is derived from the latent heat fluxes using the latent heat of evaporation and is adjusted to the available
water in the soil. It occurs in the first grid cell only, but water can be drawn upwards due to matric
potential differences. Because vegetation is very scarce in the catchment, we do not expect transpiration
to have a strong imprint on evapotranspiration and our calculations do not unravel evaporation from
transpiration.

*3.2.4.   Model setup and validation*

The setup of the CryoGrid community model for the land is presented in Fig. 4. To capture the

high spatial variability of mountainous climate, our approach relies on the 368 climate forcing datasets
to cover the catchment (see Sect. 3.2.2.). This approach enables us to perform spatially distributed
modeling. All of the 368 simulations are independent and use the same parameterization. In absence of
direct observation of the soil stratigraphy within the catchment, the soil column was designed to agree
with field observations in the region (Yuan et al., 2020; Wang et al., 2009; Hu et al., 2020; Luo et al.,
2020; Yang et al., 2014b; Wang et al., 2008), to be consistent with similar modeling approaches across
Tibet (Chen et al., 2018; Song et al., 2020) and to be consistent with input datasets (Shangguan et al.,
2013, 2017). Thus, the soil stratigraphy is divided into 3 units: a top soil (0.3 m thick), a bottom soil
(1.7 m thick), and a bedrock unit (extending beyond the depth of interest of the study). An overview of
the parameters for each unit, their source and the way they are calculated is presented in Appendix A,
Tab. A1.

Regarding the processes implemented in the model (Sect. 3.2.3.), infiltration according to Richards

equation only occurs in the top and bottom soil units. The bedrock unit has a static water content.
Unraveling surface from subsurface flow is an ongoing challenge in catchment-scale hydrology
(McDonnell, 2013) and this distinction is important in mountain terrains where these two flows can
behave differently due to the complex topography (Seibert et al., 2003; Gao et al., 2014; Hu et al.,
2020). For this study, we rely on a simple approach that is based on thresholds regarding the soil water
content (porosity and field capacity). This kind of approaches are thus based on soil properties and have
often been used in hydrological modeling studies (Vörösmarty et al., 1989; Shaman et al., 2002;
Kelleners et al., 2010; Kampf, 2011; Samuel et al., 2008). In detail, we compute surface and subsurface
flow as follows.
On the one hand, surface runoff is computed relative to the saturation level of the soil column.
When the entire soil column is saturated (WC = porosity), additional water input from precipitation or
snowmelt is directly counted as surface runoff. On the other hand, subsurface runoff is computed
relative to the field capacity of the ground, which is an input parameter of the model. When the water
content (WC) of a ground cell exceeds this field capacity (FC), the amount of water corresponding to
WC-FC is available to produce subsurface runoff. We use the lateral boundary condition
LAT_WATER_RESERVOIR from the CryoGrid community model (Westermann et al., 2023) to
account for this subsurface runoff. The speed at which this available water exits the soil column towards
the lake is calculated with Darcy's law, using the hydrological conductivity of the ground and the mean
slope of the catchment as hydraulic slope. Because the model couples thermal and hydrological fluxes,
all of these changes in the soil water content can be driven by precipitation input, evaporation but also
water phase change in the ground such as ice melt.
Because we do not have knowledge of the distributed thermal state with depth over the catchment
at the beginning of the simulations, we assume temperature profiles were in equilibrium with the climate
of the 5 first years of modeling (1980-1984). To do so, we start our simulations with a 60-year spin-up
of these first 5 years (12 repetitions), which is sufficient to establish a stable temperature profile over
the first 9 to 80 meters depending on the simulations, extending beyond the hydrologically active part
of the ground (the first 2 meters).

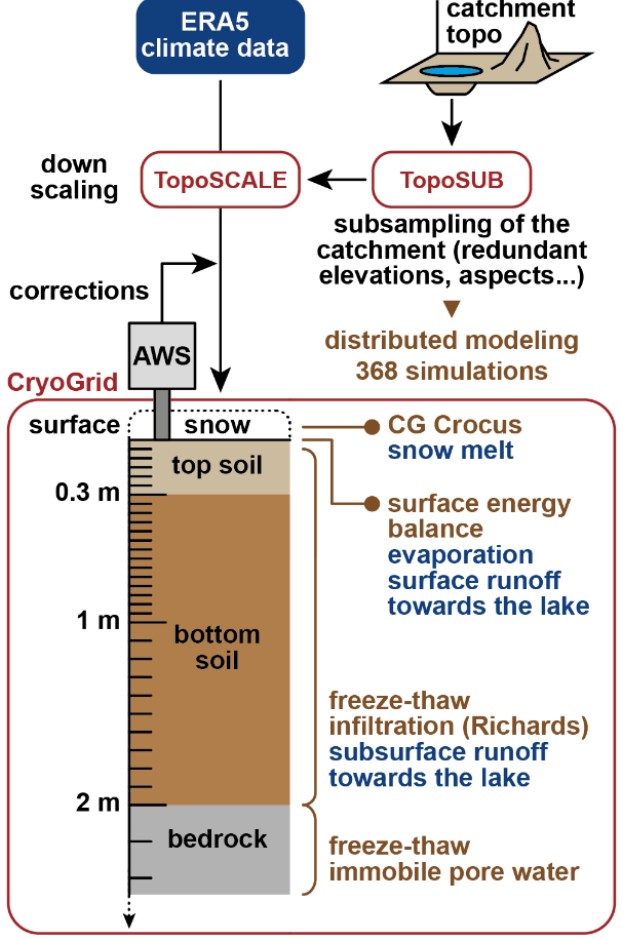

*Figure 4. Modeling framework for the land hydrology. ERA5 data are downscaled using the TopoSUB*
*and TopoSCALE approaches (Fiddes and Gruber, 2014, 2012). The downscaled data are bias-*
*corrected based on the AWS observations. Distributed 1D simulations are performed using the*
*CryoGrid community model (Westermann et al., 2023). The vertical resolution is indicated with the tick*
*marks on the depth axis.*
To validate model simulations, the simulated ground surface temperatures (GST) are compared to
the two temperature logger time series acquired in the vicinity of the AWS (Sect. 3.1.). We used this
comparison to calibrate the surface roughness used for the surface energy balance calculations in the
model.

The following method is used to produce area-averaged evaporation and runoff (in mm water

equivalent) in a zone of interest. For a given TopoSUB point in this zone, the model produces
hydrological values in $m^3$ using the area of a TopoSUB pixel on the catchment map. Then these values
are multiplied by the number of pixels in the zone corresponding to this TopoSUB point in particular,
and this for all the relevant TopoSUB points covering the zone (e.g. evaporation in warm permafrost).
Then the area of interest is calculated by counting the number of pixels in the zone of interest and
multiplying this number by the area of a pixel. Then the total volume is divided by the total surface for
the zone of interest to obtain the final value in mm.
*3.2.5. Lake modeling*
The lake thermo-hydrological response to the climatic forcing data is simulated using the
CryoGrid3-Flake model (Langer et al., 2016). The two models were coupled by Langer et al. (2016) to
simulate the thermal regime of thermokarst lakes (including surficial water freezing and melting) and
underlying ground. Here we use the coupled models mainly to quantify evaporation at the lake surface.
In the coupled model, the native surface energy balance module of CryoGrid3 (Westermann et al., 2016)
was amended to account for processes tied to free water surface energy balance: (i) the dependence of
the albedo of a water surface to solar angle (and thus time of the day) and wind speed (and wave
formation), (ii) the dependence of the surface roughness length to wind speed (and wave formation) and
(iii) the exponential decay of incoming radiation with depth in the water column. Similar to the land
simulations, the lake simulations were forced by the downscaled ERA5 data (with the TopoSUB and
TopoSCALE methodology), with the corrections derived from the AWS data (Sect. 3.2.2.). The
simulations were initiated with a 20-year spin-up of the 1980-1984 climate. The simulation results
corresponding to the four ERA5 tiles covering the lake were then averaged using the respective spatial
footprint of each tile on the lake.
*3.2.6. Quantification of glacier mass change*
Multiple studies quantified the volume change of the glaciers located within the Paiku catchment
in the recent past (1970s to 2020). To our knowledge, there are no field based measurements of glacier
mass balance available in this catchment. As a consequence, we rely solely on geodetic mass balance
studies (Brun et al., 2017; Maurer et al., 2019; King et al., 2019; Shean et al., 2020; Hugonnet et al.,
2021). All these studies estimated glacier volume changes over periods of 20-30 years from satellite
derived DEMs. As a consequence, we can only estimate the average annual glacier mass balance, and
not the year-to-year variability. Glaciers occupy approximately 113 $km^2$ in the Paiku catchment. They
have shrunk for the past fifty years at a rate of 0.44 % $y^{-1}$, from an area of 132 $km^2$ in 1975 to 122 $km^2$
around 2000 and to their current extent (King et al., 2019; Bolch et al., 2019). The average mass
balances for the period 1975-2000 and 2000-2020 are -3.9 ± 2.1×$10^{10}$ kg $y^{-1}$ and -5.4 ± 2.4×$10^{10}$ kg $y^{-1}$,
respectively (-4.6 ± 2.5 $10^7$ m$^3$ and -6.4 ± 2.8 $10^7$ m$^3$ with a 850 kg m$^{-3}$ density). These mass balances
correspond to specific mass balances of -0.31 ± 0.17 m of water equivalent per year (w.e. y$^{-1}$) and -0.47
± 0.21 m w.e. y$^{-1}$, respectively.

Regarding glacial runoff, it was estimated to 320 ± 4 mm per year for the 2001-2010 period by

Biskop et al. (2016) using a temperature-index approach for ice melt. For the 2000-2018 period, Zhang
et al. (2020b) derived a runoff value of 52 ± 12 mm per year (1.24 ± 0.29 $10^8$ m$^3$ per year that we scaled
to the basin area). The value we derive of 39 ± 13 mm per year thus finds good consistency with the
latter one (Sect. 4.1.1.).

# 4. Results

## 4.1.	Model validation and hydrological budget of Lake Paiku

### 4.1.1.	Model validation

Simulated daily ground surface temperatures are in good agreement with the observed ones, showing a bias of -0.2 °C and 0.6 °C and a RMSE of 1.4 °C and 1.6 °C for loggers 15 and 13, respectively (Fig. 5A and 5C). Most of this RMSE is explained by a mismatch between model and observations in the tails of the temperature distribution, whereas intermediate temperatures exhibit the best agreement with observations.

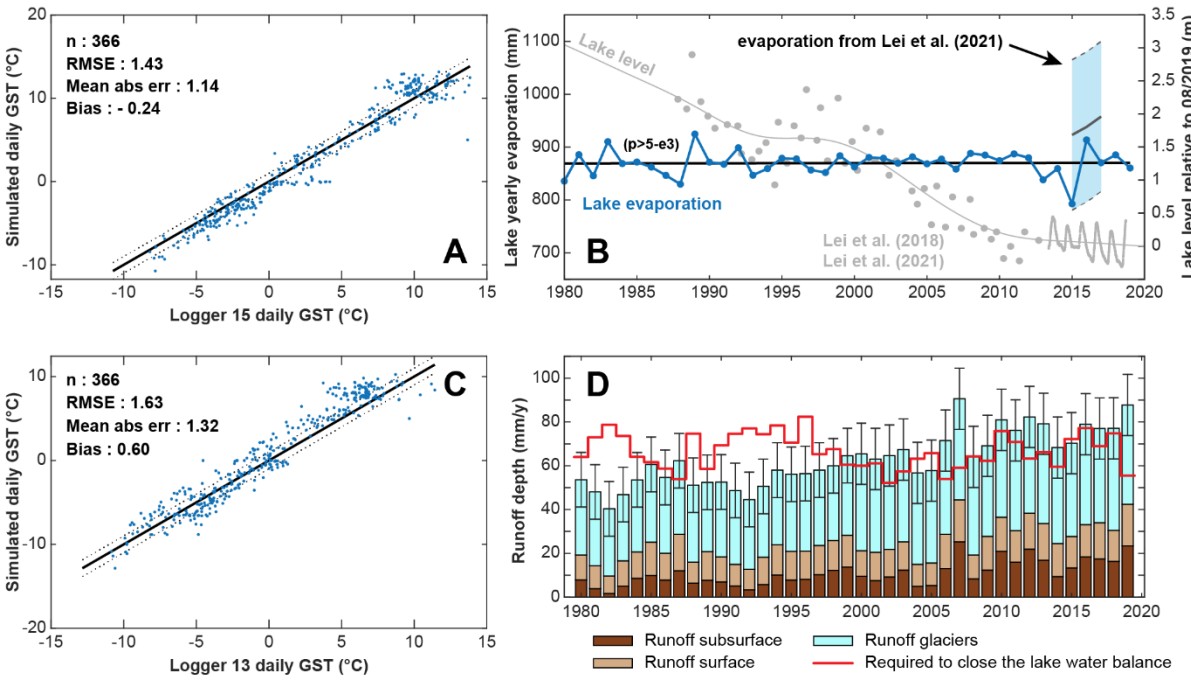

*Figure 5. Model validation. A and C: modeled mean daily ground surface temperatures compared to measured ground surface temperatures for logger 15 and logger 13 (location on Fig. 1). B: modeled annual lake evaporation (blue curve) and comparison with values calculated by Lei et al. (2021) in the light blue zone. The gray curve shows the smoothed lake level relative to August 2019 based on observations from Lei et al. (2018) (gray points) and Lei et al. (2021) (gray oscillating line). D: Comparison between the runoffs required to reproduce the observed lake variations (red curve, derived from lake level, lake area, forcing data and lake evaporation) and the sum of the glacier and land runoff we derive from remote sensing observations and modeling respectively (Sect. 3.2.). Error bars are associated to the glacier values and come from the geodetic results. Runoff values are expressed as heights scaled to the land surface of the Paiku catchment.*

Annual lake evaporation mainly ranges between 800 and 900 mm per year (Fig. 5B), with a mean value of $870 \pm 23$ mm ($1\sigma$). Lake evaporation does not exhibit a linear trend of increase or decrease and is mostly dominated by year-to-year variability. Though slightly lower, our evaporation results are in

good agreement with the values from Lei et al. (2021), which are derived from local and regional
meteorological observation and lake budget calculation (Fig. 5B). We used the simulated evaporation
together with the lake level data and lake area data from Lei et al. (2018) and Lei et al. (2021) and the
precipitation forcing datasets (3.2.2) to derive the total runoff (land + glacier) required as an input to
the lake budget to reproduce the lake variations. This required runoff corresponds to the red line of
Fig. 5D. The required runoff volumes are scaled to the land area of the catchment to be comparable
with the other variables. Fig. 5D also presents the runoff values derived from the land cryo-hydrological
modeling and from the glacier remote sensing investigations. Annual volumes are expressed as mm
over the land part of the catchment (excluding the lake). As presented in Sect. 3.2.6., glacier mass
balance values are considered constant for the 1980-2000 period and the 2000-2019 period and are
respectively equal to -4.6 $\pm$ 2.5 $10^7$ and -6.4 $\pm$ 2.8 $10^7$ m$^3$ per year. The addition of annual precipitation
to these values to quantify the total glacier runoff introduces year-to-year variability to the glacier
runoff. At the catchment scale, the average glacier runoff over the 40 years is 39 $\pm$ 13 mm per year.

Over the 40 years, the average annual land runoff value (surface + subsurface) we model is 24 $\pm$ 8

mm. Summed together, the land and glacier runoff find a partial agreement with the runoff that is
required to close the lake water balance. Annual values are compatible within error bars for 28 out of
the 40 years of simulations. The glacier and land runoff are slightly too small to close the lake water
balance during the first 20 years and slightly too large for the last 20 years of simulation. Over the whole
period, the sum of the glaciers + land runoff produces 95% of the required runoff. Land runoff is further
described in Sect. 4.3. and lake results in the following section.

*4.1.2.    Hydrological budget of Lake Paiku*

Our observations, climate data, simulations, geodetic data and the lake level data from Lei et al.

(2018, 2021) enable us to quantify the different terms of the lake hydrological budget. We present these
results in m of lake level change based on the average slope of the Volume = f(level) relationship
(Fig. 6). As the unique output term, evaporation dominates the lake budget with an average annual value
of 0.86 m (34.6 m per 40 years, Fig. 6A). Direct precipitation in the lake is the dominant input with an
average annual value of 0.31 m (12.3 m per 40 years), followed by glacier runoff (0.28 m per year, 11.3
m per 40 years) and land runoff (0.18 m per year, 7.0 m per 40 years). When compared with lake volume
observations over the 40 years of the simulation, the simulated lake budget is 1.04 m too negative.

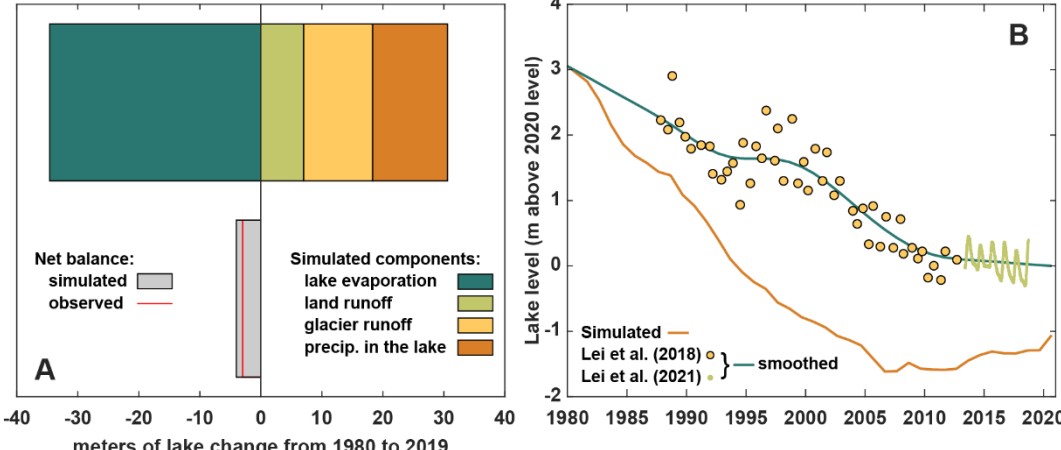

*Figure 6. Budget and level of lake Paiku for the simulation period (1980-2019). A. The different*
*components of the hydrological budget of the lake according to our framework. Results are given in m*
*of lake change based on the average slope of the Volume = f(level) relationship. B. Lake level data.*
*Points correspond to observations from Lei et al. (2018, 2021) that we smoothed (green curve, based*
*also on observation points older than 1980). The simulated lake level appears in orange.*
Based on our results, we also reconstructed lake level variations that we compare with the observed
variations (Fig. 6B). Following our framework, our values are presented at an annual timestep. They
qualitatively reproduce the overall lake level decrease but tend to overestimate this decrease and show
an increasing mismatch with the observations from 0 in 1980 to 2 meters in 2005. This mismatch is
later compensated by an increasing lake level trend in our simulation from 2005 to 2019. At the end of
the simulation period, the mismatch is 1.04 m, consistent with the budget values (Fig. 6A) and the fact
that our approach provides 95% of the required runoff to close the lake budget (Sect. 4.1.1.). This pattern
of a too strong decrease followed by an increase is consistent with the comparison between simulated
and required runoff presented on Fig. 5D.
## 4.2.    Ground thermal results
Based on our temperature results, we define four categories of ground thermal regimes (Fig. 7A).
*Cold permafrost* are the areas of the catchment for which the deepest thaw depth did not exceed 1 m
over the 40 years of simulation. For cold permafrost, frozen conditions dominate the first meters of the
ground most of the year and surficial thawing during summer can be interrupted by ground freezing

from the surface to the top of the permafrost at night. *Warm permafrost* are the areas of the catchment presenting permafrost for the whole duration of the simulation and which are not part of the *cold permafrost*. These areas are characterized by a distinct seasonal pattern of frozen ground in winter and an active layer in summer. *Disappearing permafrost* are the areas of the catchment presenting permafrost at the beginning of the simulation and not at the end. *No permafrost* are the areas without permafrost at the onset of the simulation. The geographical characteristics of each ground category are presented in Tab. 1, and their distribution throughout the catchment is shown on Fig. 7A. These different ground categories are subsequently used to compare their cryo-hydrological behaviors during the simulation (consistent color code).

*Table 1. Cryological classification of the catchment based on the modeled ground temperatures.*

| Name | Characteristics | % of the catchment area | Elevation mean (masl) | Elevation range (masl) | Slope mean (°) |
|---|---|---|---|---|---|
| **Cold permafrost** | Max thaw depth over the 40 years < 1m | 3% | 6068 | 6946 5213 | 35±13 |
| **Warm Permafrost** | Max thaw depth > 1 m and permafrost present over the 40 years | 19% | 5480 | 5921 4877 | 20±9 |
| **Disappearing permafrost** | Permafrost present in 1980 but disappears during the simulation | 5% | 5274 | 5552 4882 | 18±9 |
| **No permafrost** | No permafrost from 1980 to 2019 | 73% | 4900 | 5463 4580 | 10±8 |

At the catchment scale, the 2 m depth temperature (Fig. 7B) shows a pronounced warming trend of 0.17 °C per decade ($p=1\times10^{-6}$). This trend is mainly supported by the *no permafrost* areas, which underwent a slightly stronger warming trend of 0.2 °C per decade ($p=7\times10^{-8}$). Areas with disappearing permafrost, warm permafrost and cold permafrost exhibit smaller trends around 0.1 °C per decade with decreasing p-values (respectively 0.00001, 0.006 and 0.05, i.e. non-significant for the last two).

From 1980 to 1989, permafrost covers 27% of the catchment and the mean active layer thickness (ALT) is $1.36 \pm 0.51$ m ($1\sigma$, minimum: 0.11 m and maximum: 2.37 m, Fig. 7C). From 2010 to 2019, permafrost covers 22% of the catchment. At the scale of the initial permafrost area, this change corresponds to a loss of 19%. The mean ALT is $1.29 \pm 0.49$ m ($1\sigma$, minimum: 0.11 m and maximum: 2.55 m, Fig. 7D) for this period. Permafrost disappearance (grey zones in Fig. 7D) mainly happens for

low-lying permafrost of the south and the center of the catchment. It occurs for the most part on the
outer slopes of the permafrost regions and at the bottom of steep glacial valleys.

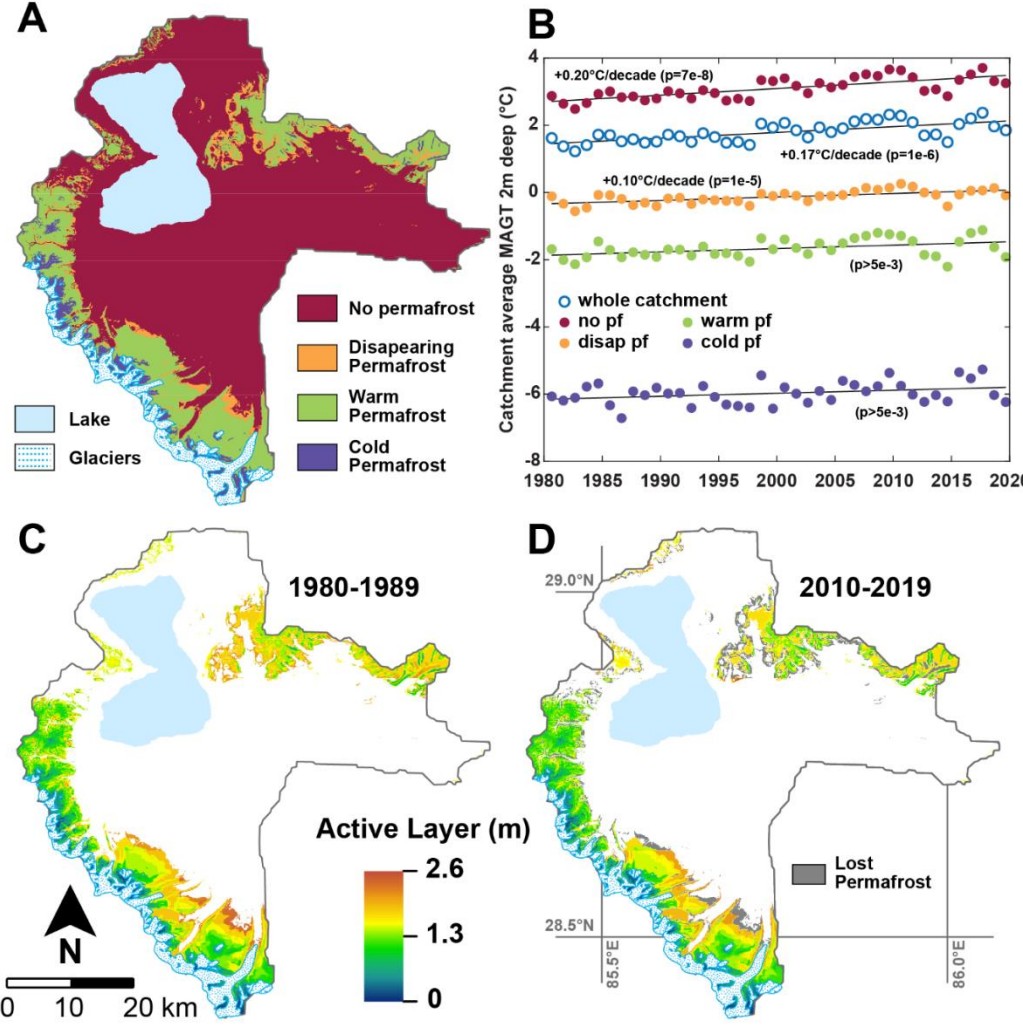

*Figure 7. A: Different cryological states of the ground throughout the catchment for the 1980-2019*
*period (see Tab. 1). B: Mean Annual Ground Temperature (MAGT) 2 m deep, averaged for the whole*
*catchment and for the different cryological states of the ground. C: Average active layer depth over the*
*1980-1989 period. D: Average active layer depth over the 2010-2019 period. Only locations presenting*
*permafrost at the end of the simulation are assigned a color on the map on C and D. Locations where*
*permafrost has disappeared are shown in gray on D.*
We also present the average duration of seasonal thaw at a depth of 70 cm averaged over the
catchment (Fig. 8A). Because at this depth some areas might present two (or more) consecutive years
without thawing (highest locations) or without freezing (lowest locations), these areas were excluded
from the averaging. In the end, the averaged results account for 89% of the catchment land area (i.e.
excluding glaciers and lake Paiku). The results show an increasing trend in the duration of the seasonal
thaw of +4.6 days per decade (p=$3\times10^{-4}$, blue line on Fig. 8A). When looking at the average start and
stop days of the seasonal thaw (Fig. 8A, grey lines) in the Julian calendar (day 150 is the 30th of May
and day 300 is the 27th of October), we note that this increase is mainly caused by a later ending date of
the thaw season (*Stop date* on Fig. 8A, +6.1 days per decade, p=8×10⁻¹⁰) and not by an earlier starting
date (non-significant trend).

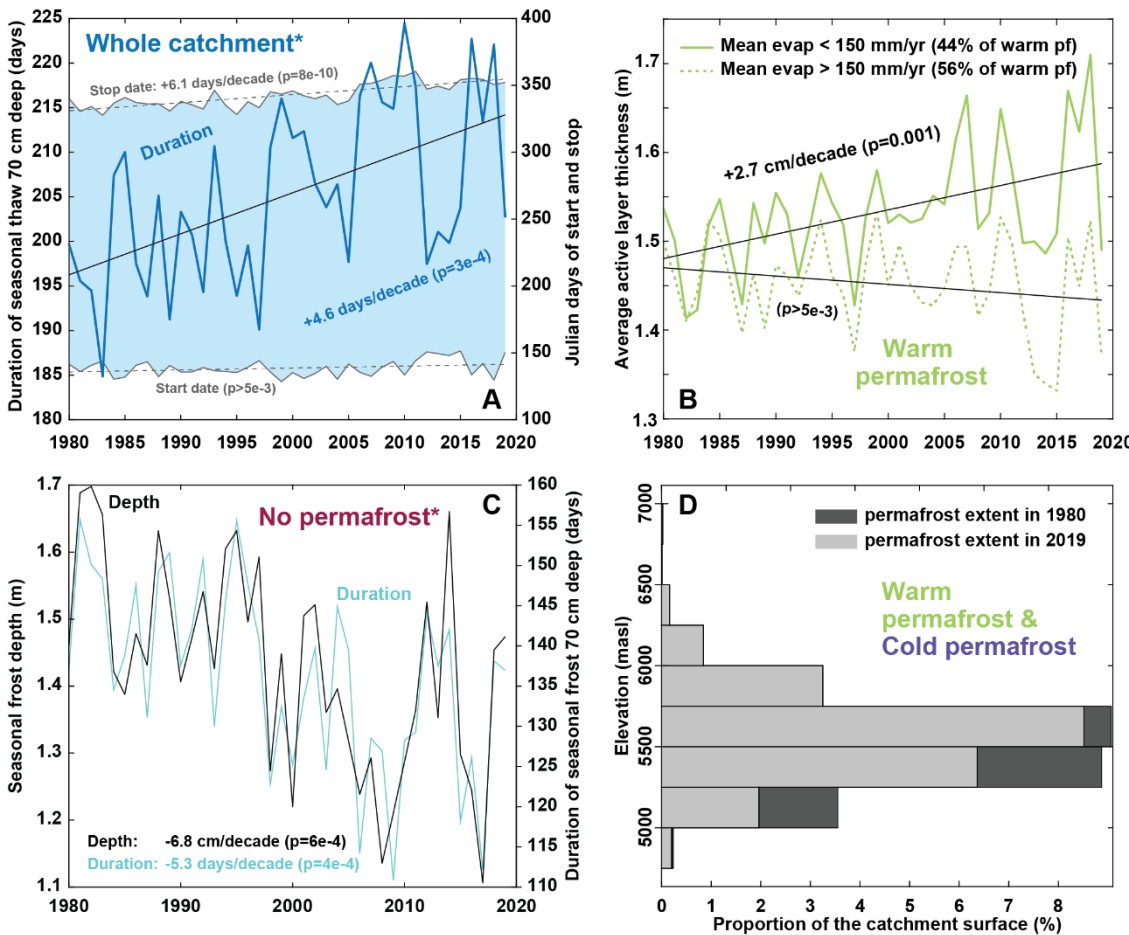

Figure 8. A: Duration of seasonal thaw 70 cm deep averaged over the catchment. The asterisk indicates
that the presented curves average 89% of the surface of the catchment (Sect. 4.2.). The gray curves and
the light blue area are associated with the right axis and indicate the average start and stop day of the
seasonal thaw in the Julian calendar. Values higher than 365 indicate that freezing conditions came
back after the 31st of December. B: Active Layer Thickness (ALT) evolution for warm permafrost. The
solid line shows the ALT for simulations experiencing an annual evaporation lower than 150 mm when
averaged over the 40 years. The dashed line shows the ALT for simulations with annual evaporation
higher than 150 mm. C: Temporal trends for seasonally frozen ground where there is no permafrost.
The asterisk indicates that simulations were excluded if one of the simulated years did not present
freezing conditions 70 cm deep (persistence of thawed conditions from one year to another). The
presented curves thus average 88% of the total permafrost-free areas of the catchment. D: Altitudinal
distribution of permafrost in 1980 and 2019. This distribution includes both cold and warm permafrost.

Within *warm permafrost*, we distinguished ALT for locations experiencing an average evaporation
lower or higher than 150 mm per year during the simulations (Fig. 8B). Whereas locations with average
evaporation below 150 mm per year record an active layer deepening trend of 2.7 cm per decade
(p=0.001), it is not the case for locations with an average evaporation higher than 150 mm per year
(non-significative trend). This threshold value of 150 mm per year is based on further investigations on
the relationships between evaporation and ALT provided in Sect. 5.3.1.

In the permafrost-free areas of the catchment, seasonal frozen ground (Fig. 8C) reaches a depth of

$1.43 \pm 0.15$ m on average and shows a decreasing trend of -6.8 cm per decade ($p=6\times10^{-4}$). At a 70 cm
depth, the average duration of seasonally frozen ground is $136 \pm 12$ days with a decreasing trend of -
5.3 days per decade ($p=4\times10^{-4}$). These values average 88% of the no permafrost areas since locations
showing persistent thawed conditions at this depth from one year to another were excluded (i.e. minimal
seasonal freezing depth over the 40 years lower than 70 cm).

When comparing permafrost spatial distribution between 1980 and 2019 (Fig. 8D), our results

show that permafrost distribution above 5750 masl has not been modified during the simulation.
Permafrost disappearance has mainly occurred between 5000 and 5750 masl, with the largest loss
reaching 2.5% of the catchment area between 5250 and 5500 masl.

## 4.3.    Hydrological results for the land


The mean annual evaporation (land area only) over the simulation time is $180 \pm 19$ mm ($1\sigma$, Fig.

9A). Evaporation shows an increasing trend over the 40 years of +1.01 mm per decade ($p=3\times10^{-7}$).
Average total runoff over the 40 years is $24 \pm 8$ mm per year (Fig. 9B) and exhibits an increasing trend
of +4.8 mm per decade ($p=8\times10^{-7}$). Similarly, surface runoff ($13 \pm 3$ mm per year) and subsurface runoff
($11 \pm 6$ mm per year) show increasing trends of +1.3 and +3.5 mm per decade ($p=6\times10^{-5}$ and $3\times10^{-7}$)
respectively (Fig. 9B). The surface runoff presented on Fig. 9B includes the snow melt that did not
infiltrate the ground. These linear trends we report are high compared to the absolute values of the
variables and their extrapolation backward in time would lead to null values in the recent past which is
unrealistic. This suggests a non-linear evolution of these variables over the XX[th] century.

We also present the catchment average of the *runoff / (runoff + evaporation)* ratio (Fig. 9C), which

is equivalent to *runoff / (rain + snow – snow sublimation)* given the negligible contribution of soil
storage variations. Hence it is the proportion of the water input to the ground surface that is converted

into runoff. This proportion is $11 \pm 2\%$ over the simulation time and shows an increasing trend of +1.23% per decade ($p=2\times10^{-7}$). Fig. 9C also shows the average theoretical ratio to maintain a steady lake level (of 17.6%). This ratio was obtained under the following hypothesis:

- Same climate forcing data, hence same lake evaporation
- The glacier contribution is (i) considered the same for the historical simulation and this scenario and (ii) taken as the difference between the total land surface runoff and the red curve of *required runoff* in Fig. 5, therefore independent of remotely sensed estimates.
- Under these conditions, the runoff increase needed to maintain the lake level is only supplied by land runoff (surface and subsurface) by shifting the *runoff / (runoff + evaporation)* ratio.

The ratio from the historical simulation starts significantly below the theoretical steady lake ratio (10.2% < 17.6%, Fig. 9C) and increases progressively to 16.0% in 2019.

Finally, Fig. 9D shows the annual proportion of *liquid / total* water averaged for the whole catchment. The value was computed based on the daily water content (liquid and frozen) of the first 2 m of the soil column (the hydrologically active part of the column, Sect. 3.2.4.) from which annual averages were derived and used to compute a catchment scale average. The graph shows that the proportion of liquid water in the total water content increases at around +1.41% per decade ($p=1\times10^{-4}$), indicating that water spends more and more time in the ground in a liquid form, being thus increasingly available for hydrological processes such as evaporation or runoff.

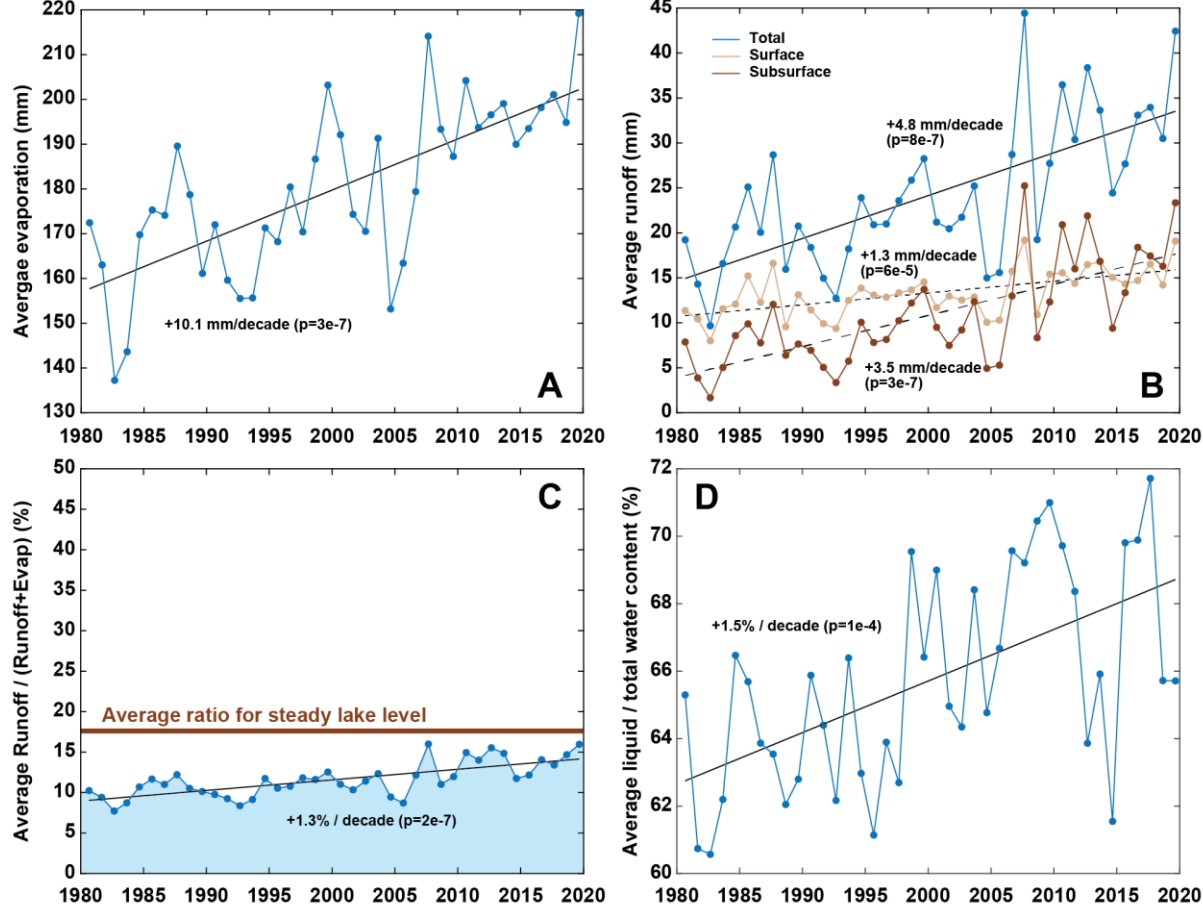

*Figure 9. Hydrological results. A: Annual evaporation averaged over the whole catchment (land area only). B: Annual runoff averaged over the whole catchment (land area only). The blue curve sums the surface and subsurface runoff. C: Ratio between runoff and (evaporation + runoff) averaged over the whole catchment (land area only). The brown line indicates the theoretical average ratio needed to maintain a steady lake level when considering an identical glacier contribution to runoff (details in Sect. 4.3.). D: Annual mean of the (liquid water)/(total water) ratio over the first 2 meters of ground, averaged over the whole catchment (land area only).*

## 4.4.    Sensitivity of evaporation and runoff

We conducted a simple sensitivity test on the climatic conditions (i.e. not a full-scale sensitivity test). We ran the same 40 years of simulations (with thermal initialization) for a climate 1 °C cooler and 30% wetter (more precipitation) than the historical scenario. We call this new scenario *colder and wetter* (to be compared with the *historical scenario*, i.e. the results of the present study presented in the rest of Sect. 4.). Results of this experiment are presented in Fig. 10 and Table 2. Because of the difference in climate forcing, the *colder and wetter* scenario produced a greater amount of *cold* and *warm permafrost* areas than the historical scenario, as presented on Fig. 10A. Fig. 10B shows the proportion of the

precipitation reaching the surface (rain + snow – snow sublimation) that produces runoff compared to
evaporation for the Paiku catchment. Fig. 10C aggregates over the whole catchment the distribution of
such precipitation input to the ground between runoff and evaporation for both scenarios. In between
them, it also includes the distribution associated with the steady lake level scenario of Fig. 9C, which
is based on the hypothesis listed as bullet points in Sect. 4.3. (climate forcing of the historical scenario,
same glacier contribution, only land runoff increases).

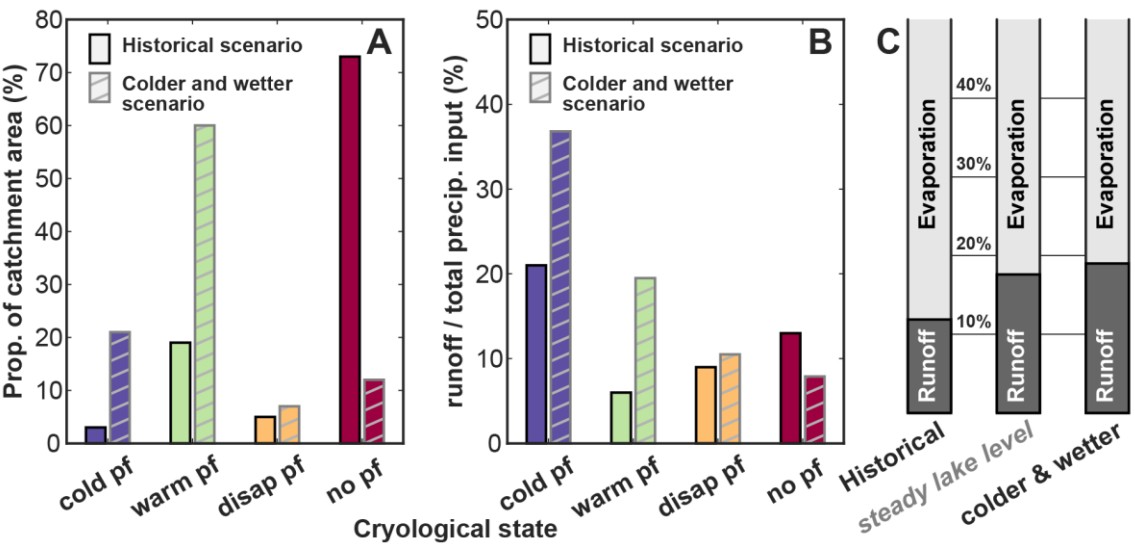

*Figure 10. Sensitivity of the distribution between runoff and evaporation to climate. A: distribution of*
*the different cryological states of the ground for the historical scenario (presented in Sect. 4.1. to Sect.*
*4.3.) and for an alternative scenario where the climate is 1 °C colder and brings 30% more*
*precipitation. B: runoff as a proportion of the precipitation input to the land (rainfall + snowfall – snow*
*sublimation) for the different cryological states of the ground and for the 2 climatic scenarios. C:*
*catchment scale ratio between runoff and evaporation for (i) the historical scenario, (ii) for a steady*
*lake level with the same glacier contribution (same as Fig. 9C), and (iii) for the colder and wetter*
*scenario.*
The *historical scenario* shows that *cold permafrost* areas produce the highest proportion of runoff,
which we attribute to the fact that the ground in these areas is most of the time frozen, turning a
substantial part of the snow melt and rainfall into surface runoff. When considering grounds with a
hydrologically active subsurface (*warm permafrost*, *disappearing permafrost* and *no permafrost*) in the
historical scenario, the proportion of runoff increases slightly from *warm permafrost* to *no permafrost*.
Such an evolution then corroborates the idea that the presence of permafrost tends to increase
evaporation at the expense of runoff, as modeled by Sjöberg et al. (2021). Yet, for the *colder and wetter*
scenario, runoff shows a regular decrease from *cold* to *no permafrost* with a more pronounced trend
than the historical scenario. Several factors can be at play in this transition and most likely involve (i)
a different extent and altitudinal distribution for each cryological type of ground, (ii) an overall reduced
intensity of evaporation due to cooler surface temperatures, (iii) a higher soil water content driven by
higher precipitation and (iv) difference in the seasonal timings. Altogether, these processes substantially
change the proportion of water that ends up as runoff water available for the lake, as highlighted by Fig.
10C.
*Table 2. Distribution between runoff and evaporation for the 2 scenarios*

| Ground cryological type | Historical Scenario | | | Colder and wetter scenario | | |
|---|---|---|---|---|---|---|
| | Precipitation input[1] | Runoff | Evaporation | Precipitation input[1] | Runoff | Evaporation |
| **Cold permafrost** | *100%* <br> 117 mm | *21%* <br> 24 mm | *79%* <br> 93 mm | 100% <br> 234 mm | 37% <br> 86 mm | 63% <br> 148 mm |
| **Warm permafrost** | *100%* <br> 183 mm | *6%* <br> 10 mm | *94%* <br> 173 mm | *100%* <br> 281 mm | *20%* <br> 55 mm | *80%* <br> 226 mm |
| **Disappearing permafrost** | *100%* <br> 211 mm | *9%* <br> 19 mm | *91%* <br> 192 mm | *100%* <br> 211 mm | *10%* <br> 22 mm | *90%* <br> 189 mm |
| **No permafrost** | *100%* <br> 218 mm | *13%* <br> 28 mm | *87%* <br> 189 m | *100%* <br> 200 mm | *8%* <br> 16 mm | *92%* <br> 184 mm |

*1. Precipitation input is the input to the ground, counted as rainfall + snowfall – snow sublimation*

# 5. Discussion

## 5.1. Limitation and potential of the approach

### 5.1.1. Data usage within the conceptual framework and data scarcity

Our approach relies on a variety of data regarding their scientific focus (glaciers, ground, lake, atmosphere), their type (in situ observations, remotely sensed data, reanalysis data), their characteristics (point wise data, distributed data, constant or with various time resolution) and the way they interact with our models (model parameters, forcing data, validation data, result data in case of the glacier runoff). Such a diversity arises from our goal to quantify both the ground thermo-hydrological regime and the different terms of the lake budget. This variety also makes it challenging to consistently merge these data into a unique framework. For example, our quantification of the glacier mass change reconstruction is made of two constant values for the study period (1975-2000 and 2000-2020), which limits the relevance of the comparison between the observed lake level variations and the simulated ones.

Yet, the lake level variations are the only hydrological observations available to evaluate the robustness of the runoff we compute. Therefore, we had to combine lake level observations with our precipitation forcing data and lake evaporation quantifications in a simple mass conservation calculation, to derive the land runoff to the lake required to reproduce the level variations (red curve on Fig. 5D). In this regard, the sum of the glacier and land runoff we derive over the 40 years correspond to 95% of the required runoff to the lake, indicating that the magnitude of our reconstruction is correct. Year-to-year comparison is less accurate and we suggest that this is the consequence of the aforementioned limitations and also of our modeling strategy as detailed below.

A main limitation regarding our usage of the data is related to the limited amount of available field observations required to provide robust model parameterizing, climate forcing and in-depth validation of the simulations, both hydrologically and thermally. Regarding climatic forcing data, our AWS measurement offers sound observations to evaluate and adjust the ERA5 data processed with TopoSUB and downscaled with TopoSCALE. Yet, a period of observations longer than 2 years would have enabled more robust corrections and could have allowed us to perform a more advanced statistical

downscaling approach, e.g. quantile mapping (Themeßl et al., 2011). As such, the spatiotemporal
domain of relevance of these corrections is insufficient to correct data for the whole catchment and the
40 years of simulations. Overall, considering the strong bias we observe in the raw ERA5 data (Fig.
D1), these corrections do represent an important first-order improvement.
Additionally, in absence of borehole data that would allow us to anchor our parameters into
observations, we rely on gridded values designed for hydrological and/or land surface modeling (Sect.
3.2.4. and Appendix A). Because these values might be less reliable than field observations, we chose
to average them over the catchment to derive some more robust values. Altogether, this scarcity of field
observations is likely to bring significant uncertainties to our analysis. Future efforts should focus on
acquiring additional data or developing validation methods based on remotely sensed observations.
*5.1.2.  Modeling strategy*
A limitation in our study is that lateral water flows between land simulation units is ignored. By
giving access to the timing of water transport across the catchment, water routing would allow to
investigate temporal hydrological patterns at a monthly or seasonal scale. Because we work at annual
and decadal time scales, this limitation has limited consequences on our results. The main consequence
is to ignore potential storage effects on the land that would delay the arrival of runoff to the lake. We
suggest that it is possible that this limitation partly explains the limited match between computed and
required runoff at the annual time scale (Fig. 5). Yet, our subdivision of the catchment based on the
different cryological states of the ground allows us to work with hydrological units that are smaller than
the catchment and thus present shorter hydrological response time to precipitation.
Additionally, our approach regarding the modeling of runoff is relatively simple, i.e. partition
between subsurface and surface runoff based on comparison between the soil water content and field
capacity and porosity, respectively. More complex approaches split runoff into more sophisticated
categories such as Horton overland flow, Dunne overland flow, subsurface stormflow… (e.g. Savenije,
2010; Gao et al., 2014; Mirus and Loague, 2013). However, over the last decade, the relevance of this
type of partitioning between different types of runoff has been questioned (McDonnell, 2013; Gao et
al., 2023). In the frame of our study, we find it important to distinguish between surface and subsurface
runoff because they generate flows with very contrasted speed. In a general perspective, this significant
difference in flow velocities impacts the hydrological system as a whole (e.g. river discharge,
evaporation…) and has various consequences throughout the catchment, such as the water availability
for vegetation, erosion and sediment transport.
In the particular case of a cryo-hydrological study, separating surface from subsurface runoff is
particularly relevant because both flows do not react in the same way to ground temperature changes.
As such, we see our approach as a middle way that allows us to make this distinction based on simple
hydrological considerations. Yet, we acknowledge that the classification and quantification of the
different types of runoff represent a valuable direction for future investigation on catchment-scale cryo-
hydrology in Tibet. Another potential improvement in our modeling approach could be to unravel
evaporation from transpiration. However, since vegetation is extremely scarce in the Paiku catchment,
which is largely dominated by barren lands, we suggest that this would not significantly affect our
results. However, this limitation should be explored in future field and modeling studies.
Conversely, our approach also conveys several important advantages regarding our goal to
describe and quantify the ground thermo-hydrological regime of the catchment. The use of TopoSUB
enables us to produce results at a resolution of 100 x 100 m over an area of nearly 2400 km$^2$ with
calculation costs 700 times lower than if each 100 x 100 m pixel was treated individually. Yet, thanks
to the clustering method used to produce the forcing dataset (Sect. 3.2.2.), the strong spatial variability
of the physiography and its impact on the climate and incoming radiations is significant in the forcing
data and has a major influence on the ground thermo-hydrological results, as exemplified by the strong
spatial variability of ground temperatures (Fig. 7). Beyond elevation, other physiographic parameters
such as aspect also influence the results. The mean values of 2 m-deep temperature and evaporation
over the 40 years for north-facing areas (averaged over the whole catchment and over the 40 years) are
1.3 °C and 163 mm while they reach 2.9 °C and 197 mm for the south-facing ones. This strong
dependence of modeled results on physiography highlights the necessity to take it into account when
modeling the thermo-hydrological regime of the ground in high mountainous environments. Finally,
our approach allows us to couple the physical processes governing both energy and water fluxes at the
surface and subsurface and highlight their interplay, as developed in Sect. 5.3.1.

## 5.2. Trends in the catchment and across the QTP

### 5.2.1. Lake hydrological budget and level variations

The total lake level change we simulate is a decrease of 4.11 m. This is qualitatively consistent with the overall observed trend. The mismatch with the observations is limited to a 1.04 m excess in the simulated level drop (Fig. 9A). Our reconstruction shows a decrease of 4.66 m from 1980 to 2007, which is an overestimation of the initial drop. Afterwards, while observations indicate a gradual slowdown of the lake level decrease, we simulate a stabilization followed by a slight increase (0.55 up between 2013 and 2019). The reason for the overall mismatch of 1.04 m can arise from bias (i) in the forcing data (and mainly in the precipitation) used for the land and lake simulations, (ii) in the glacier mass balance estimate and/or (iii) in the quantification of hydrological processes for the land or for the lake (evaporation, runoff). On top of these potential biases, the difference in trends for the end of the simulation time can be influenced by (i) our estimates of glacier mass changes, which are made of two time averages (one for the 1980-2000 period and one for the 2000-2020 period) and therefore produce very smoothed glacier runoff values that cannot capture variations at the scale of the decade of less and (ii) the absence of water routing that prevent us from accounting for delays of storage effects on the water supply from the land to the lake.

Additionally, our approach ignores potential water fluxes between the lake and a surrounding aquifer. This can be a possible reason for this mismatch. In the context of a decreasing lake level, an aquifer surrounding the lake can create an additional water inflow when the lake level passes below the piezometric level of the aquifer (Yechieli et al., 1995). We suggest that such an inflow could mitigate the lake level decrease and thus explain the missing water in our reconstruction (Fig. 6B). It could also explain the gradual stabilization of the lake level that our model does not reproduce. This flow is not part of our conceptual hydrological framework even though it likely exists in reality, especially since there is no permafrost near the lake (as we simulate it here), allowing for the existence of such an aquifer (Walvoord and Kurylyk, 2016). Groundwater has been identified as a potential contributor to lake level rise in other regions of the QTP (Lei et al., 2022). In the long run, lake-aquifer systems commonly follow oscillations of the net atmospheric flux of water (Precipitation – Evaporation) and of the runoff

that forces its mass balance (Watras et al., 2014). During these oscillations, the lake can "pump" water
from the aquifer or feed it depending on the relative difference of piezometric level between them
(Almendinger, 1990; Liefert et al., 2018). Yet, this potential effect is difficult to account for and its
magnitude remains unclear. Therefore, the reasons for the mismatch between observed and simulated
lake levels could also be connected to other aspects of our methodology such as bias in the climatic
forcing data and other shortcomings arising from the lack of field data, or hydrological processes, as
developed in Sect. 5.1.1. and 5.1.2.
Our reconstruction of the lake budget is informative regarding the respective contribution of the
different inputs and outputs. Regarding lake evaporation, our mean value of $870 \pm 23$ mm is close to
the one modelled by Yang et al. (2016) with the Flake model for lake Nam ($832 \pm 69$ mm) for the period
1980-2014 but we do not report a significant increasing trend in our results. Yet for the same lake (Nam
Co) and a similar period (1980-2016) Zhong et al., (2020) reported an average value of $1149 \pm 71$ mm
(along with an increasing temporal trend) using the Penman formula (Penman, 1948), thus highlighting
the potential dependence of the results to the methodology. In our results, direct precipitation to the lake
represents 40% of the inputs, followed by glacial runoff (35%) and land runoff (25%). Glaciers are
therefore a particularly important contributor to the runoff towards the lake (60% of the total runoff, vs.
40% for land runoff), what contrasts with the results from Biskop et al. (2016) who calculated that the
runoff input to the lake Paiku was dominated by land runoff (70% vs. 30% for the glacier contribution).
Here again, these differences likely arises from important differences in input data and methodologies
to quantify the different hydrological processes (evaporation, runoff, snow and glacier melt). Yao et al.
(2018) reported that, at the QTP scale, the balance between precipitation and evaporation (over land
and lake) was dominant over glacier melt to understand both lake storage increases and decreases. Our
reconstruction does not give us access to significant temporal variation of the glacier contribution but
the above-mentioned proportions in the contributions to the lake (40%, 35% and 25%) show that the
glacier contribution does not dominate the input terms. At the catchment scale, these proportions can
vary significantly depending on the glacier coverage. For Lake Selin, Zhou et al. (2015) reported that
runoff towards the lake, evaporation from the lake and on-lake precipitation altogether explained 90%
of the lake storage variations for the 2003-2012 period. The catchment of lake Selin has a very limited
glacier coverage, corresponding to 0,63% of its area (Lei et al., 2013), compared to the Paiku (5%).
### 5.2.2.  *Permafrost and ground temperature trends*
Our results indicate that permafrost coverage in the Paiku catchment evolved from 27 to 22% of
the land area during the simulated period. Such a coverage corresponds to sporadic permafrost (10-50%
of the area) and is consistent with recent large-scale estimates of permafrost in the Northern Hemisphere
(Obu et al., 2019) and across the QTP (Zou et al., 2017; Ran et al., 2018). This decrease corresponds to
a 19% shrinkage of the 1980 permafrost area, which is higher than the 9% reported by Gao et al. (2018),
a value determined by catchment-scale numerical modeling in the upper Heihe catchment (northeastern
QTP) over a similar period. It is also slightly higher than the 13% decrease modeled from 1971 to 2015
for the Qinghai Lake catchment with a similar approach by Wang and Gao (2022). Yet, it is smaller
than the 34% loss modeled by Qin et al. (2017) from 1981 to 2015 for the Yellow River Source Region
(YRSR, North Eastern QTP).
Active layer (AL) evolution is contrasting throughout the catchment and a deepening signal is only
visible for the locations with limited evaporation (<150 mm per year). Given the strong drive of summer
climate on ALT, this overall lack of a deepening trend highlights how evaporation can act as an energy
intake at the surface (Yang et al., 2014a), limiting the surface and subsurface heat fluxes and thus AL
deepening. In this regard, our results fall in line with the conclusions of Fisher et al. (2016) when
observing evapotranspiration and ALTs in boreal forests and also confirm the modeling experiments of
Zhang et al. (2021b) on permafrost wetting in arid regions of the QTP. Besides, the lack of an overall
deepening trend is consistent with observations from Luo et al. (2018) in the YRSR over the last decade
and with the modeled AL from Zhang et al. (2019) at the scale of the QTP for the last 40 years. Where
evaporation is limited, we report an AL deepening trend of 2.7 cm per decade, which is smaller than
the 4.8 cm per decade trend modeled by Song et al. (2020) for the YRSR for the same period, and
smaller than the 4.3 cm modeled by Gao et al (2018) in the upper Heihe catchment. Yet it is comparable
to the 2 cm per decade value modeled by Wang and Gao (2022) for the Qinghai Lake catchment from
1971 to 2015. Connection between AL deepening and evaporation are discussed in Sect. 5.3.1.
In *no permafrost* areas, our simulations show that the thickness of seasonally frozen ground shrinks
at a rate of 6.8 cm per decade. This rate is faster than the rate of 3.1 cm per decade quantified by Qin et
al. (2018) using the Stefan solution for the YRSR (1961-2016) and faster than the 3.2 cm per decade
modeled by Gao et al. (2018, Heihe catchment). However, it is similar to the 6 cm per decade rate
modeled by Wang and Gao (2022) in the Qinghai Lake catchment from 1971 to 2015 and smaller than
the 12 cm per decade modeled by Qin et al. (2017) for the YRSR (1981-2015). All these values fall
within the wide range of 3 to 29 cm per decade reported by Wang et al. (2020a) when studying
seasonally frozen ground over the whole QTP with in-situ observations. Regarding timing, we report a
decreasing trend of 5.3 days of frozen conditions (70 cm deep) per decade which is consistent with the
decrease of 6.7 days per decade reported by Wang et al. (2020a) just below the surface.
Regarding the timing of seasonal ground thaw, our results highlight that the increase in the duration
in the seasonal ground thaw (at 70 cm) is mostly driven by a progressive delay of the end date of the
thaw period. This result contrasts with those from Song et al. (2020) for the same period in the YRSR
who also modeled an increase of the seasonal thaw (at a 2 cm depth), although driven by an advancing
trend of the start date of the seasonal thaw.
Our warming trends at a 4 m depth for permafrost areas is 0.1 °C per decade, which is substantially
smaller than the 0.43 °C per decade observed at this depth between 1996 and 2006 in permafrost
boreholes along the Qinghai-Tibetan Highway in the North East of the QTP (Wu and Zhang, 2008).
Zhang et al. (2019) reported a 0.13 °C per decade of warming of the permafrost top during winter that
is consistent with the trend of 0.14 °C per decade we observe at 2 m depth (mean AL between 1.4 and
1.7 m in our simulations) for the months of December, January and February.

*5.2.3.  Evaporation and runoff trends*

Our results are characterized by (i) an increase of both evaporation and runoff (Fig. 9A and 9B),
mainly driven by an increase in precipitation (Fig. 3 bottom), (ii) a runoff/(runoff+evaporation) ratio
exhibiting an increasing trend as a result of ground warming and permafrost disappearance that both
enable more subsurface runoff along time (Fig. 9C and 10D) and (iii) an increase in the proportion of
liquid water in the ground compared to ice (Fig. 9D). Regarding all these points, our results find a good
consistency with the evolution reported by Gao et al. (2018) for the upper Heihe catchment
(northeastern QTP) using a similar approach for a comparable period (1971-2013). The increasing
trends in evaporation and runoff they report for the thawing season (dominant period for both processes)
are comparable with the annual values we report: +10.0 mm per decade for evaporation (our study:
+10.1 mm per decade) and +3.3 mm per decade for runoff (our study: +4.8 mm per decade). Similar
evolutions are also reported by Wang and Gao (2022) for the Qinghai Lake catchment and by Qin et al.
(2017) for the YRSR (1981-2015). These increases in runoff (especially surface runoff) are likely to
have an influence on sediment transport. For instance, Li et al. (2021) showed that current precipitation
increase over High Mountain Asia is driving a runoff increase, which contributes to a significant rise in
fluvial sediment fluxes. Regarding differences, Qin et al. (2017) modeled a stronger evaporation
increase (14.3 mm per decade) linked to a decreasing runoff coefficient. Similar to Li et al. (2019), we
see that an important part of snow melt (49%) infiltrates the ground and later contributes to runoff and
evaporation.
## 5.3. Cryo-hydrological couplings at catchment scale and implication for
lake level variations
### 5.3.1. Interdependence of thermal and hydrological variables
Our simulation results enable us to explore the interplay between the fluxes of energy and water at
the surface and subsurface. In this regard, we tested the correlation of evaporation with the proportion
of liquid/total water in the ground for cold and warm permafrost, as well as the correlation between
evaporation and the duration of seasonal thaw at a 70 cm depth (Fig. 11, A and B). For permafrost areas
(*cold permafrost* and *warm permafrost*), evaporation shows a strong correlation with the seasonal
distribution between liquid and frozen water, similar to previous modeling works for the region (Cuo
et al., 2015). As such, this correlation suggests that the intensity of seasonal ground thaw plays a role
in enabling higher or lower evaporative fluxes. This is likely due to cold surface temperatures strongly
reducing water loss from the surface and because moisture delivery to the surface is inhibited when the
ground is frozen. We suggest that this dependence is particularly important in the Paiku Catchment
because evaporation is strong (88% of the precipitation input to the surface evaporates on average) and
because frozen water is the dominant form of water in the ground in permafrost areas (Fig. 11A, the
calculation includes the first 2 meters below the surface).
Similarly, evaporation in *no permafrost* areas shows a significant correlation with the duration of
the seasonal thaw (Fig. 11B). We suggest that this result arises from the fact that frozen ground limits
the evaporative fluxes and thus years during which the subsurface seasonal thaw is shorter are associated
with reduced evaporative fluxes. We also tested the relationship between the linear trend of active layer
deepening and the mean evaporation (over the 40 years of simulation) for *warm permafrost* areas (Fig.
11C). Thus, this graph does not present annual values and one point corresponds to one of the 92
TopoSUB points classified as *warm permafrost* (values averaging the 40 years). The graph highlights
that TopoSUB points showing an Active Layer (AL) deepening trend are associated with low
evaporation and precipitation. From there, TopoSUB points with stronger evaporation show no
deepening trend or even a shrinkage of the AL. This relationship is contradicted by the highest level of
evaporation (>240 mm per year) observed for *warm permafrost*, for which AL deepening is observed
again (dark blue points of the graph). These TopoSUB points with the highest levels of evaporation also
correspond to those receiving the largest amount of precipitation.
Runoff also shows a strong connection with the ground thermal regime (Fig. 11D). At the
beginning of the simulation, years with an average 2 m-deep temperature below 0 °C are associated
with limited subsurface runoff (< 5 mm per year). Over the years, as the ground warms up and
permafrost disappears, subsurface runoff increases and can reach 20 to 45 mm per year. This result is
consistent with increased subsurface connectivity expected when permafrost thaws (Kurylyk et al.,
2014; Gao et al., 2021) that has been both observed (Niu et al., 2016) and modeled (Lamontagne-Hallé
et al., 2018; Huang et al., 2020; Gao et al., 2018). We suggest that these substantial changes in
subsurface runoff, associated with changes in the ground temperature in Fig. 11D support the hypothesis
of a modification in the hydrological pathways as permafrost thaws.
Altogether, these results suggest a dependence of key variables quantifying the catchment
hydrological balance (evaporation, runoff) to the seasonal characteristics and interannual trends of the
ground thermal regime (temperature, liquid vs frozen water content). Similar to previous studies (Ding
et al., 2020; Wang and Gao, 2022), we think these results advocate for the necessity to couple thermal
and hydrological modeling to improve our ability to understand and quantify changes in the
hydrological balance of high mountain catchments. To our best knowledge, along with Gao et al. (2022),
our study represents to date the most complete effort to include the variety of coupled climatological,
surface and subsurface processes characterizing the climate, hydrology and ground thermal regime of
high-mountain catchments in Tibet at a small scale with a high spatial resolution.

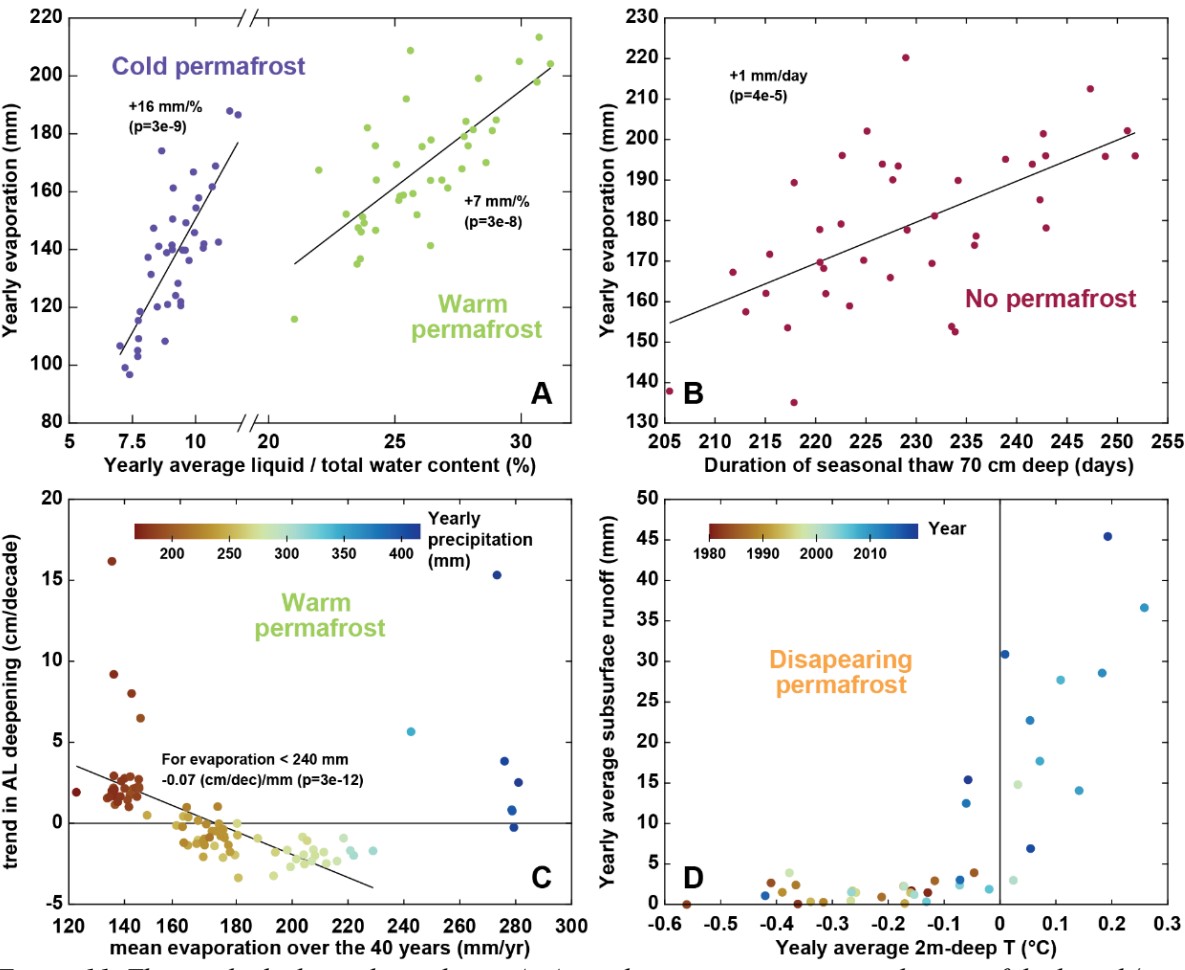

*Figure 11. Thermo-hydrological couplings. A: Annual evaporation vs. annual mean of the liquid / total*
*water ratio over the first 2 meters of ground, averaged for simulations corresponding to cold permafrost*
*and warm permafrost (one dot per year for each permafrost category). B: Annual evaporation vs.*
*duration of seasonal thaw at a 70 cm depth averaged for simulations corresponding to locations without*
*permafrost (one dot per year). C: Active layer deepening trend vs. mean evaporation over the 40-year*
*for each simulation corresponding to warm permafrost (here one dot corresponds to one TopoSUB*
*point). The color of the dots shows the precipitations averaged over the 40 years for each simulation.*
*The linear regression excludes simulations exhibiting annual evaporation higher than 240 mm. D:*
*Annual subsurface runoff vs Annual 2 m-deep temperature averaged for simulations corresponding to*
*locations with disappearing permafrost (one dot per year). The color of the dot indicates the year of*
*the simulation.*

### 5.3.2. Influence of the ground thermal regime on the distribution between runoff and evaporation

Our results indicate that evaporation is particularly strong in the Paiku catchment. Over the 40 years of simulation, 10% of the total precipitation is converted into runoff, and the rest of the water is either directly returned to the atmosphere from the snowpack via snow sublimation or from the ground surface via evaporation. Comparatively, Gao et al. (2018) observed and modeled a ratio of around 35% for the Heihe catchment; Qin et al. (2017) reported an average ratio of 33% for the YRSR and Li et al. (2014), a ratio of 83% for the Qugaqie catchment (central QTP) but modeling hydrological fluxes only.

Our sensitivity test on evaporation and runoff for a slightly different climates (Sect. 4.4.) highlights the fact that the role of permafrost regarding the runoff/evaporation distribution is a complex question, as it has already been discussed in the literature (e.g. Bring et al., 2016). Some studies have suggested that landscape-scale permafrost thaw would trigger more evaporation (Walvoord and Kurylyk, 2016). This phenomenon was modeled by Wang et al. (2018) in the upper Heihe River Catchment, for which they reported that the thickening of the active layer increased the ground storage capacity and led to a decrease in runoff and an increase in evapotranspiration. Wang et al. (2020b) also reported that permafrost thawing accelerated evapotranspiration (1961-2014).

Conversely, Zhang et al. (2003) and Carey and Woo (1999) reported that shallow frozen ground conditions (such as a shallow active layer) maintain higher water contents close to the surface, promoting higher evaporation. Sjöberg et al. (2021) modeled this phenomenon with a fully coupled cryo-hydrological model including surface energy balance calculation. They modeled a slope with a simplified geometry in 2D for different permafrost coverages. They found that hillslopes with continuous permafrost have twice as high rates of evapotranspiration compared to hillslopes with no permafrost.

As such, the interplay between the runoff/evaporation distribution and the ground thermal regime in areas where permafrost coverage shows a spatiotemporal variability is difficult to apprehend (Fig 10). This complexity is most likely due to a strong sensitivity to the drainage conditions (fast flows of steep mountain environments vs. slow flows of lowland catchments) and to the climate setting, both at

the annual scale (arid regions vs. wet regions) and at the seasonal time scale (relative timing of
temperature variations, rainfall, snowfall, snow melt and ground freeze/thaw).
Because it can both promote evaporation or runoff depending on the setting, the ground thermal
regime of the catchment seems to have the possibility to create a positive feedback, both towards lake
level decrease or increase. Further studies should therefore focus on comparing the thermo-hydrological
regime of different Tibetan catchments with contrasting lake level changes and permafrost coverage, to
test to which extent these differences can contribute to explain the spatial patterns of lake level changes
across the QTP.

## 5.4.    Implications for lake level changes


At the scale of the Paiku catchment and in regard of lake level variations, the results we present
highlight that:
• The sum of the direct precipitation in the lake, the land runoff and the glacier runoff are not
enough to compensate for the lake evaporation over the study period, hence driving the
observed lake level decrease.
• Long-term hydrological trends in the catchment are led by trends in climate; and
precipitation increase, jointly with glacier melt, provides enough water to drive a
concomitant increase of runoff and evaporation.
• Ground thermal changes increase the distribution of liquid vs. frozen water in the ground
and the duration of seasonal thaw, correlations suggest that these modifications increase
evaporation. The warming of the ground is also related to the increase of subsurface runoff
towards the lake.
• Ground warming and permafrost thawing promote subsurface runoff over time,
contributing to an increase in the runoff/evaporation ratio of the catchment.
• Over the 40 years we studied, the presence of permafrost seems to promote evaporation at
the expense of runoff. Yet this trend appears to be climate-dependent and the cryological
state of the ground might shift the runoff/evaporation distribution in the other direction
under colder and wetter climates.
At the scale of the QTP, these results have several implications. First, a better understanding of the
recent and future lake level variations will come with a better knowledge of spatial patterns and
temporal trends in precipitation. Second, climate changes are modifying the ground thermal regime of
Tibetan catchments. Ground warming may lead to active layer deepening, permafrost disappearance
and/or changes in the seasonal freeze/thaw cycles, affecting evaporation, runoff volumes and pathways
and overall, changing the hydrological functioning of Tibetan catchments (and the waterflow provided
to the lakes). Finally, the effect of permafrost on the distribution between evaporation and runoff seems
to be dependent on the climate settings and the permafrost coverage of the catchment. Further studies
should investigate this phenomenon and how it might contribute to explaining the contrasting lake level
evolutions across the QTP.

# 6. Conclusion

Our study quantifies the different terms of the Lake Paiku budget over the past 40 years. Direct precipitation to the lake represents 40% of the inputs, followed by glacial runoff (35%) and land runoff (25%). Glaciers are therefore a particularly important contributor to the runoff towards the lake.

We also confirm that the ground of the Paiku catchment presents different types of cryological states, from seasonally frozen ground to permafrost. Permafrost coverage shrinks from 27 to 22% of the land area of the catchment from the 1980s to the 2010s (19% loss of the 1980 permafrost area). The whole catchment warms up at a rate of 0.17 °C per decade (2 m deep), with a substantial elevation-dependent variability. This warming is concomitant with an increase in the duration of the seasonal thaw, mainly supported by a progressive delay of the end date of the thaw period. Where permafrost is present, active layer deepening is only observed where evaporation is relatively low (<150 mm yr$^{-1}$).

Over the simulation period, we also report an increase in evaporation (+10.1 mm per decade), surface and subsurface runoff (+1.3 and +3.5 mm per decade respectively). Together, this leads towards an increase of the runoff/(runoff + evaporation) ratio of +1.2% per decade. Our results also highlights the strong interdependence between the ground thermal and hydrological regimes and the necessity to jointly represent them to accurately quantify evaporation and runoff in this type of environment.

Over the last 40 years, the presence of permafrost seems to promote evaporation at the expense of runoff. Yet this trend appears to be climate-dependent and the cryological state of the ground might shift the runoff/evaporation distribution in the other directions under colder and wetter climates. Further studies should investigate this phenomenon and how it might contribute to explain the contrasted lake level evolutions across the QTP.

# Appendix A: model parameters

*Table A1. Parameters of the model.*

| Depth | Layer | Parameter | Values | Source | Calculation |
|---|---|---|---|---|---|
| 0.0 m | Surface | Albedo | 0.24 | Modis MCD43A3.006 | November mean, 4600-5100 masl |
| | | Emissivity | 0.95 | Modis MCD43A3.006 | November mean, 4600-5100 masl |
| | | Roughness | 0.024 | - | Adjusted to fit loggers T values |
| 0.0 m | Top soil | Thickness | 0.30 m | HiHydro Soil v1.0 | modeling framework |
| | | Porosity | 0.5 | Shangguann et al. 2013 | mean |
| | | Organic | 8.60% | HiHydro Soil v1.0 | catchment mean |
| | | Mineral | 41.40% | - | subtraction (100 - porosity - orga) |
| 0.3 m | | Soil type | Sand | Shangguann et al. 2013 | dominant fraction |
| | | Field capacity | 0.32 | HiHydro Soil v1.0 | catchment mean |
| | | Hydro cond | 0.000030 m s$^{-1}$ | HiHydro Soil v1.0 | catchment mean |
| | | Alpha | 0.028 cm$^{-1}$ | HiHydro Soil v1.0 | catchment mean |
| 0.3 m | | n | 1.481 | HiHydro Soil v1.0 | catchment mean |
| 0.3 m | Bottom soil | Thickness | 1.70 m | Shangguan et al. 2017 | truncation, consistent with literature |
| | | Porosity | 0.4 | Shangguann et al. 2013 | catchment mean |
| | | Organic | 4.20% | HiHydro Soil v1.0 | catchment mean |
| | | Mineral | 55.80% | - | subtraction (100 - porosity - orga) |
| 1.7 m | | Soil type | Sand | Shangguann et al. 2013 | dominant fraction |
| | | Field capacity | 0.32 | HiHydro Soil v1.0 | catchment mean |
| | | Hydro cond | 0.000016 m s$^{-1}$ | HiHydro Soil v1.0 | catchment mean |
| | | Alpha | 0.062 cm$^{-1}$ | HiHydro Soil v1.0 | catchment mean |
| 2.0 m | | n | 1.707 | HiHydro Soil v1.0 | catchment mean |
| 2.0 m | Bedrock | Thickness | 98.3 m | - | - |
| | | Porosity | 0.03 | - | - |
| 98 m | | Organic | 0% | - | - |
| | | Mineral | 97% | - | - |
| | | Soil type | Sand | - | - |
| 100 m | | Field Capacity | 0.03 | - | equal to porosity |

# Appendix B: Geological map of the catchment

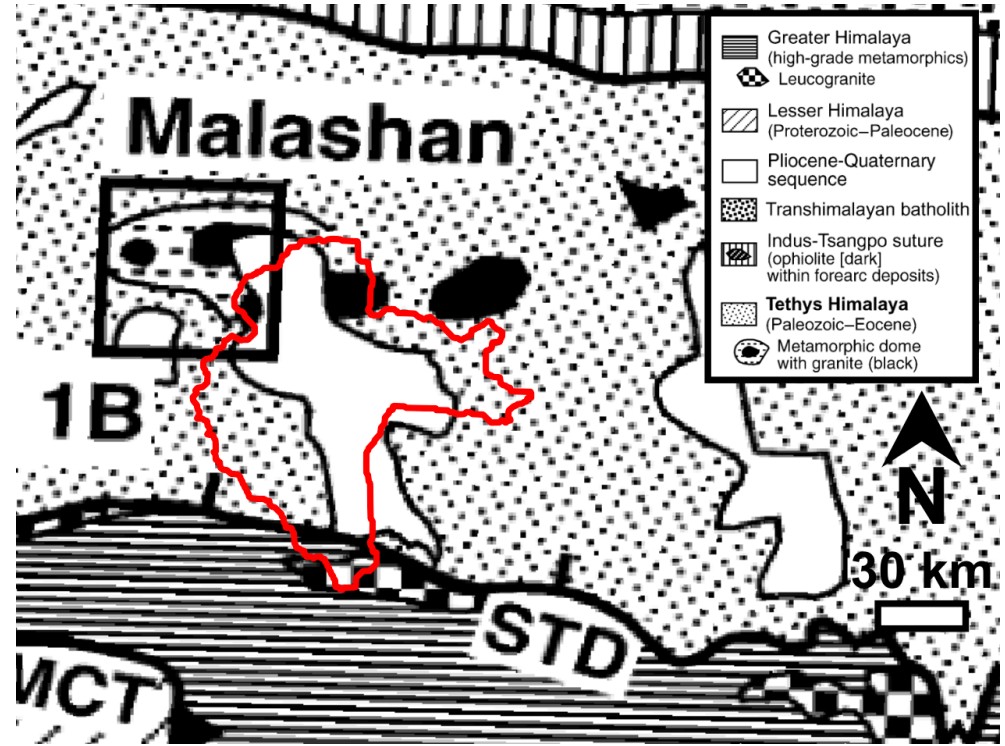

*Figure B1. Geology of the catchment. Modified from Aoya et al. (2015). The red contour indicates the limits of the Paiku catchment.*

# Appendix C: TopoSUB subsampling of the catchment

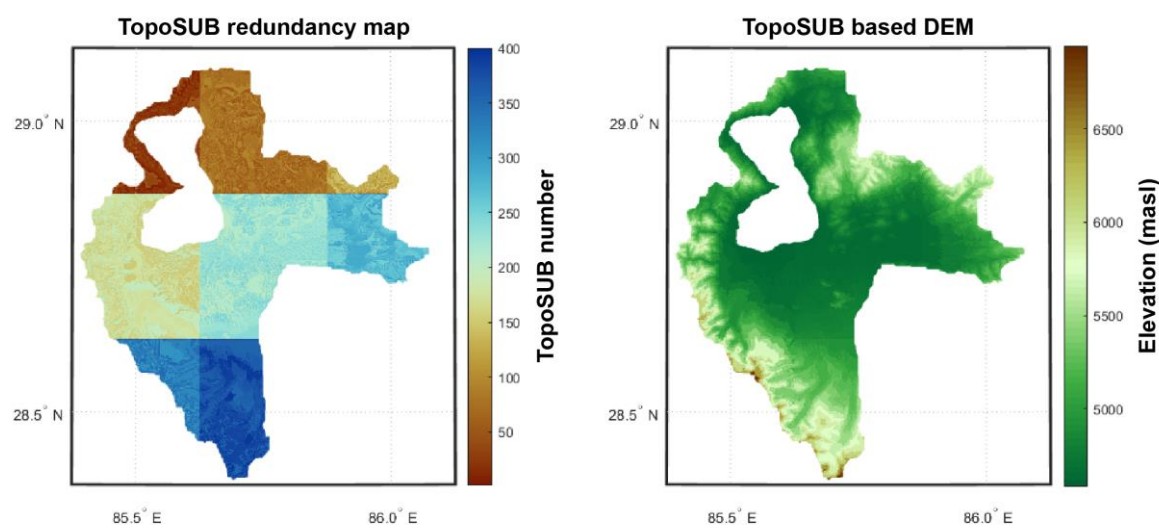

*Figure C1. Application of the TopoSUB clustering method (Fiddes and Gruber, 2012) in the Paiku catchment. Left: number of the TopoSUB points. Strong color changes reflect the footprint of the 8 ERA5 pixels that the catchment intersects. Small color changes within a given of these zones show the distribution of the 50 TopoSUB points covering each tile (Sect. 3.2.2.) B: topographic map reconstructed using the TopoSUB approach.*

 # Appendix D: Evaluation of forcing data

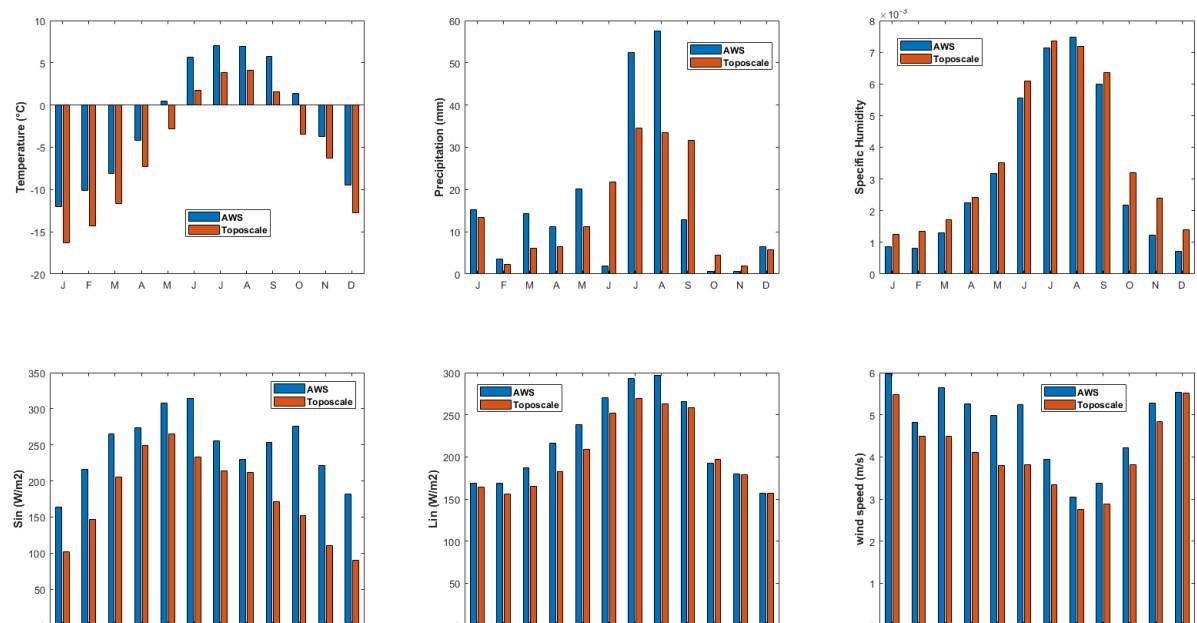

962
*Figure D1. Comparison between the AWS data and the model forcing data downscaled from ERA5 with
the TopoSCALE and TopoSUB approaches. Based on the AWS data, a monthly correction factor is
applied to the downscaled data so that monthly data matches for the observed period for each variable
(methodological details in Sect. 3.2.2.).*

**Code availability.** The CryoGrid community model (version 1.0) and related documentation are available at: https://github.com/CryoGrid/CryoGridCommunity_source.

**Data availability.** Field data have been saved on Zenodo.org and will be published with a DOI upon acceptance of the manuscript.

**Author contribution.** L.M, W. I. and S.W. designed the study. L.M. and M.M. conducted the numerical simulations. S.W., M.L. and L.M. contributed to the model development. F.B., W.I., Y.L. ad S.A. acquired field data. L.M., F.B., M.M., P.K., Y.L. and T.M. analyzed and processed the data. J.F. provided downscaled forcing data for the model. All authors contributed to result interpretation and to manuscript preparation.

**Competing interests.** The authors declare that they have no conflict of interest.

**Acknowledgements.** This study was funded by the open program of the Dutch Research Council (NWO) (ALWOP.467) and by the Strategic Priority Research Program of the Chinese Academy of Sciences within the Pan-Third Pole Environment framework (grant agreement no. XDA20100300). The land surface and lake simulations were performed on Utrecht Geosciences computer cluster. Sebastian Westermann acknowledges funding by European Space Agency Permafrost_CCI (https://climate.esa.int/en/projects/permafrost/). We are very grateful to the reviewers for their input which significantly improved this manuscript.

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
