# Peer review of "Recent ground thermo-hydrological changes in a Southern Tibetan endorheic catchment and implications for lake level changes"

_Hydrology and Earth System Sciences, 2022_

## Author Comment (AC1)

We are very grateful to Reviewer 1 for the in-depth reading and for the thorough review we received. We present below our detailed answer to the discussed points. The reviewers' comments appear in black and our responses appear in brown. Quotes from the manuscript are in *brown italic*, modified parts from the revised version are in **bold font**.

**Review 1**

The manuscript by Martin et al. describes the application of a coupled hydro-thermal modeling approach to a high-altitude catchment. The authors relate long-term lake level variations in an endorheic basin of Lake Paiku, Southern Tibet, to changes in both water balance and permafrost distribution across the catchment (finally, and this is my major concern, the manuscript lacks such relation). This modeling effort is based on ERA5 reanalysis data as driving climate forcing, downscaled and distributed across the homogenous response units with TopoSCALE/TopoSUB, the CryoGrid3-Flake as lake module and a distributed CryoGrid CM model as basic hydro-thermal model for both permafrost and hydrology in the basin. The results presented in the manuscript is scientifically sound and the obtained results enhance our understanding of permafrost hydrology and change in a high-altitude catchment with limited direct anthropogenic pressure. However, the interpretation of the results, or what exactly our understanding gains, is not of immediate evidence to me because of the issues raised below, and I suggest this manuscript is subject to major revision.

The evidences of both cryologic and hydrologic change are presented, but overall reasoning behind the conclusions is unconvincing from the hydrological perspective. First, the manuscript, since its title, aims at relating cryohydrologic change to the Paiku Lake level

We understand the reviewers' concern. As detailed below in our answers, we have further processed our simulation results to be able to present the lake hydrological budget and discuss it. Therefore, we hope that the new results, figures and discussion developed along this line make the title more consistent with the manuscript content.

– nonetheless, the simulated lake level data are not presented in the manuscript. Figure 5D showcases the runoff needed to close the lake water balance based on observed data, but it might be useful to covert runoff directly to lake level fluctuations, given that stage-volume relation is known to the authors.

Regarding Fig. 5, our intention is to provide a validation figure. Connecting the simulated runoff to observations is important for our study considering our focus on land cryo-hydrology. Yet, we understand the reviewer's comment regarding the need for more results to present and discuss the lake hydrological budget. Modifications in these directions are presented in the new manuscript and in the response to the next comments.

The manuscript beyond Section 4 discusses secondary effects without relating them to the modeled lake level change; this is the reason why finally Section 5.4 'Implications for lake level change' is so faceless and merely doubles the Section 6 'Conclusions'.

In the revised version of the manuscript, we now present our results on the lake budget in a new section (Sect. 4.4, 4.4.    Hydrological budget of Lake Paiku) with a new figure (Fig. 9). The new text and figure show the reconstructed lake level 
[revised manuscript text omitted]

Moreover, following the comment on the difference between correlation and causation, we moved the part on connections between the ground thermal regime and the hydrology. It was initially presented in the results, it is now presented in the discussion (section 5.1.4. of the new manuscript) and fully reworked (detailed answer on this point below).

Second, behind all modeling exercises, lake level variations in an endorheic basin ΔH are described by a three-member water balance equation, where ΔH is on the left, and on the right, the three members are: (1) lake surface balance, described as a (P-E) term, (2) catchment runoff, split into river runoff and side inflow. In this equation, when all the members are conditionally known, the unmeasured components, e.g., loss to the deep subsurface through infiltration, can be deduced. This basic hydrological approach has only limited use in the manuscript, i.e., in the sections where water balance components are presented and discussed, there is always a component that is missing, so that overall catchment water balance cannot be closed through mental calculation. See, e.g., Section 3.2.1, Section 4.1.

The new section 4.4. and Fig. 9 now detail the different components of the hydrological budget of the lake. Regarding deep subsurface infiltration, we believe that we cannot use the method provided by the reviewer because the data we have limits us in the way we can validate our simulations. Our understanding of the reviewer's suggestion is to say:

Δlake_level = lake_precipitation – lake_evaporation + land_runoff + glacier_runoff – deep_infiltration

We have real world controls on: the lake level (observations), the precipitation (AWS data), the evaporation (Lei et al., 2021), the glacier runoff (geodetic observations) but neither on the land_runoff nor on the deep_infiltration. From there, having to quantify two variables is problematic. As discussed earlier, we decided to compute an "observation-derived" land_runoff to control our simulated runoff because it appeared important for the study. To do so, we had to assume losses to the deep subsurface to be null (as well as any type of lake-aquifer interactions). It is one of our working hypotheses to be able to implement our approach. We see that this hypothesis was not detailed enough in the initial manuscript so we modified it to clarify this point in section 3.2.1 (Conceptual hydrological model for the catchment):

**"Because the quantification of water flows between the lake and potential aquifers surrounding it is difficult (Rosenberry et al., 2015), our approach assumes that these flows are negligible."**

We also added a mention to this hypothesis in the discussion (pasted in the answer to the previous part).

It is sufficient to give long-term values for 1980-2020, and show how the balance is not closing and why; then how permafrost thaw promoted subsurface runoff (or ground ice thaw?) to finally stabilize the lake water level around its present reference level. Or the like. I don't know.

We are not sure we fully understand what the reviewer means. The calculation we did for Fig. 5D shows indeed that a term of land runoff is needed to close the lake budget (i.e. lake_precipitation – lake_evaporation + glacier_runoff is not enough) but we believe that this can be true regardless of the state of the permafrost in the catchment. Given the data we have access to, we think that deriving the land runoff from the lake level variations (and other variables) is the best we can do and that this value alone does not give access to information on the role of permafrost on the land hydrology.

Third, the manuscript draws into vague conclusions ignoring the 'correlation is not causation' axiom. In this respect, Figure 7C

Figure 7C presents the evolution of the active layer over time (similarly to other graphs presenting temporal evolution of different simulated variables). It does not aim at correlating 2 modeled variables like those of Fig. 9 (of the former manuscript ) further mentioned by the reviewer.

and Figure 9A-C are illustrative. A (sometimes not so) strong correlation between E and physiographical features may well reflect a spurious correlation, i.e., when there is a third common factor that correlates to both variables (Pearson, 1897). E.g., in Figure 9B which is incorrect in itself – there is no seasonal thaw in 'no permafrost' points

Seasonal freezing and seasonal thaw are the two sides of the same coin. As much as seasonal freezing, seasonal thaw is a straightforward and valid physical value that we derive from our ground temperature results.

– both variables might well be related to an increase in mean air/ground surface temperature, and juxtaposed control in precipitation, so that ground remains frozen longer at 0.7m when air temperature is low and precip is high causing most sensible heat available to be spent on evaporation and ground cooling. A common variable – or a multivariable set – vaguely explains this relation. Or not – but you have the data at hand to disprove my reasoning. Since this physics drives the CryoGrid model, as well as many other models, I think I am not too wide in my perception.

We understand the concern of the reviewer on these questions of correlation and causation and we have significantly modified the manuscript accordingly. We detail here our perception on this question. These correlations compare the variability of key variables for the study and test relatively straightforward and simple hypotheses regarding physics, therefore we think they are worth being part of the manuscript. Since this material (initially presented in section *4.4. Thermo-hydrological couplings)* allows us to discuss connections between key variables (ground temperature, active layer depth, evaporation, runoff), we think it represents relevant discussion content. We therefore moved this section to the Discussion, part 5.1.4. (The interdependence of thermal and hydrological variables). We also largely re-worked it and rephrased the presentation of the graphs with more cautious wording in order to avoid the confusion between correlation and causation:

*"5.1.4. The interdependence of thermal and hydrological variables*

[revised manuscript text omitted]

In the same fashion, on Figure 9C, less evaporation means faster active layer deepening exclusively in dry simulations with P < 200 mm. The AL reduction is driven by an increase in BOTH evaporation and precipitation, which presumably means that across simulations (TopoSUB points?) the evaporation is in fact moisture-limited not energy-limited (Haghighi et al. 2018, https://doi.org/10.1002/2017WR021729). This is however a speculative conclusion as I do not have all the data in hand and not intended indeed to fully reproduce this research from eleven contributing authors.

We are not sure to understand what puzzles the reviewer here. But we fully agree to the other look she/he gave at this result in its specific point to L496, where she/he expresses the message we want to convey and concludes that it seems plausible to her/him.

Finally, on several occasions, the authors were particularly imprecise in interpreting the references. See below, comments to L71-72 and L671-672. I was not up to verifying the correctness of all references, but hope that the authors will do so during the revision.

We are sorry for these imprecisions. We are grateful to the reviewer for such an in-depth verification. We went through all the references of the draft to ensure they were used correctly.

Multiple line-by-line comments are also provided:

L39: Bibi et al. 2018 does not refer to Bowen ratio or latent heat fluxes; also, should be (Yang et al. 2014a)

We split the references to avoid misunderstanding (Yang et al., 2014a speaks of Bowen ratios).

L71-72: this is an incorrect citation; Qin et al. 2017 found that evaporation is increasing along with an increase in both precipitation and air temperature (Qin et al. 2017, Figure 5, p. 837). Then, "The annual precipitation first decreased <…> from 1981 to 2002 and then increased <…> from 2002 to 2015. The annual runoff exhibited a trend similar to that of precipitation, but the runoff coefficient displayed a decreasing trend" (Qin et al. 2017, p. 839). In (Wang et al., 2020b), their Figure 5c, d (p. 8 of 13) does not show a runoff decrease; Figure 5c shows an upward trend since the mid-1990s, and Figure 5d shows variations similar to those shown by (Qin et al. 2017). So the claim that runoff is found to decrease is straightforwardly incorrect, and not supported by the references.

We modified the text for:

*"Similar ground warming trends were reported in the regional modeling study from Qin et al. (2017) along with an increasing trend in evaporation and a decrease of the runoff coefficient over time. Plateau-scale surface energy balance modeling from (Wang et al., 2020b) reported that increasing trends in evapotranspiration could be mainly explained by variations in air temperature and net radiation at the surface."*

Additionally, I find slightly controversial the two claims presented in both the manuscript and the cited literature, that (1) the change in Bowen ratio decreases as latent heat fluxes limit sensible ground warming, in other words, increased evaporation limits ground warming, and (2) ground warming promotes evaporation. This is the reasoning of a kind, "more cheese (warming) = more holes (evaporation), more holes = less cheese, more cheese = less cheese", and I struggle to find a correct line of thinking to get the logic right.

As with the cheese, we believe that both propositions 1 and 2 are correct and that the sophism arises when trying to assemble them under different working hypotheses, like different changes of parameters.

The effect "increased evaporation limits ground warming" supposes that under a similar climate signal, grounds that are not or less limited by water availability (compared to those who are) will enable a greater evaporative flux during summer, an energy loss that will counterbalance the seasonal energy intake.

The effect "ground warming promotes evaporation" supposes that, for a given level of total water availability, a ground exposed to a warmer climate (when compared to a colder one) will enable more evaporative fluxes than the one exposed to the colder climate because more liquid water is available.

From there we believe that both effects are relevant to interpret our results, provided that they are used within the correct frame. In the case of the Paiku catchment, these effects only marginally affect the main trends

of ground warming and evaporation increase imposed by the surface energy balance in response to the climate. Indeed, the energy input from the climate signal was strong enough to warm the ground, despite an increasing evaporation trend. Effect 1 only affects the magnitude of this warming between wetter and dryer locations experiencing similar climatic conditions, as shown on figures 7C and 9C. Similarly, at a given location, we think that our results support the idea that the increasing proportion of liquid water (compared to ice) over time contributes to promote evaporation, along with other changes of the surface energy budget.

L82-83: for consistency and clarity, please express all trend rates across the manuscript in units per decade, not per century.

We applied this modification throughout the text and figures

L91-92: see above; Qin et al. 2017 reason on decrease in runoff coefficient, not runoff itself.

We replaced "*runoff*" by "*runoff coefficient*".

L93: here, and elsewhere in the manuscript, replace 'yearly' with 'annual'; the former is most used as an adverb, while the latter, as an adjective.

Modification applied in the text and figures

L122; here, and throughout the manuscript, better use (Appendix B, Figure B1) to refer to Appendix data, otherwise your current reference style causes confusion, i.e., later in the manuscript, L215, and particularly L363.

Done

L128-129: better provide the range than a single value, also 200 mm is significantly lower than your Figure 1C, Figure 3C, and multiple figures throughout the manuscript, i.e., Figure 9C.

We replaced "*(~ 200 mm year$^{-1}$)*" by "*(200-300 mm year$^{-1}$)*".

L178-179: this is a proper line to place the water balance equation and present its terms – this will structure the presentation in the following sub-section.

This part now reads:

"*As such, the present study requires quantification of all these terms of the hydrological balance. **The hydrological balance of the lake is given by the following equation:***

$$\Delta z_{Lake} = Precipitation_{Lake} + Runoff_{Land} + Runoff_{Glacier} - Evaporation_{Lake}$$"

L185: 'in Section 3.2.5'

Done

L203: SRTM30 is known to be highly imprecise in mountainous regions to the degree it is red-flagged to be used 'as is' e.g., in the Himalayas (Mukul et al. 2017, https://www.nature.com/articles/srep41672). Please comment on the potential uncertainties of your approach, or, otherwise, how was SRTM30 data treated to limit such uncertainty.

The topographic parameters (elevation, slope, aspect, skyview factor) that we derive from the SRTM DEMs are clustered by the TopoSUB algorithm. As a consequence, they are smoothened and averaged by a convergence on a cluster mean value, limiting the impact of specific errors at specific DEM pixels . Additionally, the catchment presents a very strong spatial variability regarding all of these parameters so that we do not need a higher level of precision to capture these gradients. In this regard, the method manages to reconstruct the catchment topography in a reliable way (Fig. C1). Finally, this approach has, for now, no equivalent regarding its ability to capture the complexity of mountain terrains with limited computational costs for applications requiring surface energy balance calculation. Thus, even with the 10m additional error above that published for SRTM DEMs in undulating terrains (reported by Mukul et al., 2017), we believe that our methodology is still a substantial improvement compared to others that would be based on the delimitation of sub regions of the catchment or any type of response units that would be much bigger than a 100 x 100 m pixel and thus would tackle the topography in a simplified and less accurate way.

L220-224: am I correct to understand that: the observed data from one-year long record, October 2019 to September 2020, was monthly-averaged compared to a 40-year monthly-averaged ERA5 data (for a pixel/TopoSUB point where the AWS is located?), then correction factors were obtained bringing ERA5 monthly data to the AWS data, and they were applied then to other TopoSUB points? So to say, longer records were corrected by a shorter record, and regional data were corrected by punctual correction factors? If so, a largely uninspiring Section 5.1, notably sub-sections 5.1.1 and 5.1.2, can be animated with discussions on the applicability of this approach and potential uncertainties implied.

The reviewer understood correctly. There is no other meteorological measurements available to perform the necessary corrections. We extended the initial paragraph on this topic in section 5.1.:

*"Regarding climatic forcing data, our AWS measurement offers sound observations to evaluate and adjust the ERA5 data processed with TopoSUB and downscaled with TopoSCALE. **Yet, a period of observations longer than 2 years would have enabled more robust corrections and could have allowed us to perform a more advanced statistical downscaling approach, e.g. quantile mapping (Themeßl et al., 2011). As such, the spatiotemporal domain of relevance of these corrections is insufficient to correct data for the whole catchment and the 40 years of simulations. Overall, considering the strong bias we observe in the raw ERA5 data (Figure D1), these corrections do represent an important first-order improvement***.*"

L226, Figure 3: if providing a p-value for a trend, explain how it was obtained, in the separate Statistics paragraph in the Methods section. This applies to this figure and to multiple occasions across the manuscript. Were the trend tests performed, and if yes, which exactly. Mann-Kendall test would roughly give p-value of 5e-4, though consistent Sen's slope.

We added the following information to section 3.2.2 (right before the first appearance of a p-value Fig.3 presenting the forcing data):

*"**In this figure and across the rest of the study, we use p-values to evaluate the significance of linear trends in the temporal evolution of certain variables (temperature, precipitation, evaporation…). This p-value tests the null hypothesis which supposes that the value of the slope is equal to zero. The hypothesis is tested using the Student's t-test, by comparing the distance between the estimated slope and 0, relative to the standard error of the slope. We did not report trends when this p-value (probability of a null slope) was higher than $5 \cdot 10^{-3}$.***"*

L231, Section 3.2.3: evapo(transpi)ration from the land surface is not presented in this section. However, this variable plays an important part in your reasoning throughout the manuscript! Was it E or ET, is your basin al bare soil, or vegetation is present?

This comment is similar to the comment of reviewer 2. In response to both we have modified the end of section 3.2.3 (The CryoGrid community model) which now reads as:

*"**At the surface, the model uses a surface energy balance module to calculate the ground surface temperature and water content. The turbulent fluxes of sensible and latent heat are calculated using a Monin–Obukhov approach (Monin and Obukhov, 1954). Evaporation is derived from the latent heat fluxes using the latent heat of evaporation and is adjusted to the available water in the soil. It occurs in the first grid cell only, but water can be drawn upwards due to matric potential differences. Because vegetation is very scarce in the catchment, we do not expect transpiration to have a strong imprint on evapotranspiration and our calculations do not unravel evaporation from transpiration.**"*

L291-292: for the TopoSUB, the lake surface is a homogenous surface hence represented by four TopoSUB points, one for each ERA5 pixel? Explanations are needed, otherwise unclear how lake climate forcing was assembled.

To be precise, we got 7 TopoSUB points for the lake, 1 corresponding to one ERA5 pixel and 3 pairs for the 3 other ERA5 pixels. For each pair of TopoSUB points, the two datasets were almost identical and produced highly similar simulation results. We then averaged the obtained results based on their spatial footprint. We believe that, methodology wise, what is important for the reader is the averaging of results associated to 4 different ERA5 pixels and not the details on the pairs of nearly identical TopoSUB points. So we added the following explanation to the manuscript at the end of section 3.2.5 (Lake modeling):

*"Similar to the land simulations, the lake simulations were forced by the downscaled ERA5 data **(with the TopoSUB and TopoSCALE methodology)**, with the corrections derived from the AWS data (Sect. 3.2.2). The simulations were initiated with a 20-year spin-up of the 1980-1984 climate. **The simulation results corresponding to the four ERA5 tiles covering the lake were then averaged using the respective spatial footprint of each tile on the lake.**"*

L294, Section 3.2.6: data from L342 belongs here.

These data are presented in section 3.2.6. To make clear that their mention line 342 is a reminder, we modified the following sentence:

"**As presented in section 3.2.6,** *glacier mass balance values are considered constant for the 1980-2000 period and the 2000-2019 period and are respectively equal to -4.6 ± 2.5 $10^7$ and -6.4 ± 2.8 $10^7$ m$^3$ per year.*"

L309, Section 4.1: see general comments. In this section, besides model validation, the summary of the hydrological results is partially given, but incompletely. The water balance equation approach would help structuring the narration, and interpreting the results. In general, all members of the lake water balance equation are written first in absolute values, i.e., volumetric units, km3, then converted to layer units, mm, scaled either to lake surface or, less often, to catchment area. See, e.g., (Szesztay, 1974; https://doi.org/10.1080/02626667409493872) for reference water balance equation for an endorheic basin. From this approach, deep groundwater component can be roughly estimated as well. Besides, this approach allows the derivation of lake level time series which can be directly comparable to the observed data. Isn't this, according to the title, an important aspect of your study?

Following our response to the main comment and the comment on on line 178-179, the equation for the lake budget is now given in section 3.2.1. Additionally, each component of the lake budget and the reproduced lake variations are presented in the new result section 4.4. Following our response to the previous comment on the topic, our approach does not give us access to a quantification of the deep groundwater component.

L316, Figure 5: on Figure 5C, does the scale refer to lake level, or lake level change? If this is change, is it change to previous year? If it is lake level, explain the reference level – which level is taken as zero. Also, order of figures is different from other figures, Figure C is top right, while on other figures, it is in the bottom left. This is acceptable, but potentially confusing.

We modified the Y axis that now states lake level relative to August 2019, we also modified the order of the letter.

L336-337: This is unclear, rephrase and explain. Otherwise, it is evident that in lake water balance, the catchment input is important.

We removed the sentence.

L341-342: see above.

We modified the text according to the previous comment of the reviewer (comment on line 294).

L342-343: is it correct that only the annual precipitation over the glacier area was considered? Am I right to understand that all precipitation over glacier area was flushed toward the lake at all altitudes, so to say there was no glacier feeding during this time above the ELA?

To be precise, when the total precipitation equals the glacier yield, the glacier loses exactly what it gained and the mass balance over the year is 0. In this case there is no feeding only if there is also no ablation which is unlikely. When there is a negative mass balance, the glacier yield can be counted as the precipitation plus the glacier mass loss, as we presented it in the methods.

L348-349: simulated lake level curve would be more informative on this matter.

We now present the simulated lake level in the new result section 4.4.

L363: why 8m? I am curious since the model had a spin-up period of 60 years to reach the steady-state conditions at the first 2m only (L268-269). Does this mean that below 2m the model was not in the steady state after the spin-up period and hence at least some change at 8m can be attributed to non-steady-state evolution?

The motivation was to show results of temperature changes where the year to year variability is limited and where most of the signal comes from long term trends. Yet following the next remark from the reviewer we no longer display these maps.

Relatively to the steady state at depth, we realize now that our wording was misleading. The effectiveness of thermal initialization was checked for all the runs and ranged from 9 to more than 80 meters of depth, depending on the magnitude of phase changes during the initialization and the difference between the values at $t_0$ and the steady state temperature values. We rephrased those lines which now read:

"*To do so, we start our simulations with a 60-year spin-up of these first 5 years (12 repetitions), which is sufficient to establish a stable temperature profile **over the first 9 to 80 meters depending on the simulations, extending beyond** the hydrologically active part of the ground (the first 2 meters).*"

L372, Figure 6C, D: as change is not immediately deducible from this pair of images, would not it be more informative to provide one figure with change in DJF temperature between the two time periods?

We agree with the reviewer and think that this map was not the most relevant way to present our results. We have now reworked the figures presenting the thermal results. We removed these temperature maps and present instead the maps of the ground thermal regimes along with the temperature trends for each thermal type of ground (initially presented in the next figure of temperature related plots). Since this freed a space in the other temperature-related plot figure, we included the depth and duration of seasonal freezing (these 2 operations allowed to remove to appendices):

[Figure]

*Figure R1.3 (Fig. 6 of the revised manuscript). A: Different cryological states of the ground throughout the catchment for the 1980-2019 period (see Tab. 1). B: Annual 2 m deep ground temperature averaged for the whole catchment and for the different cryological states of the ground. C: Average active layer depth over the 1980-1989 period. D: Average active layer depth over the 2010-2019 period. Only locations presenting permafrost at the end of the simulation are assigned a color on the map on C and D. Locations where permafrost has disappeared are shown in gray on D.*

[Figure]

*Figure R1.4 (Fig. 7 of the revised manuscript). A: Duration of seasonal thaw 70 cm deep averaged over the catchment. The asterisk indicates that the presented curves average 89% of the surface of the catchment (Sect. 4.2). The gray curves and the light blue area are associated with the right axis and indicate the average start and stop day of the seasonal thaw in the Julian calendar. Values higher than 365 indicate that freezing conditions came back after the 31st of December. B: Active Layer Thickness (ALT) evolution for warm permafrost. The solid line shows the ALT for simulations experiencing an annual evaporation lower than 150 mm when averaged over the 40 years. The dashed line shows the ALT for simulations with annual evaporation higher than 150 mm. C: Temporal trends for seasonally frozen ground where there is no permafrost. The asterisk indicates that simulations were excluded if one of the simulated years did not present freezing conditions 70 cm deep (persistence of thawed conditions from one year to another). The presented curves thus average 88% of the total permafrost-free areas of the catchment. D: Altitudinal distribution of permafrost in 1980 and 2019. This distribution includes both cold and warm permafrost.*

L382-383: what is a 'distinct active layer season'? Same in L385. The active layer is relatively thin in cold permafrost, but the winter-summer temporal pattern holds for cold permafrost as well.

Locations corresponding to cold permafrost are located very high up and show important amplitude of the daily temperature cycles. As a consequence, the ground can freeze from the surface to the top of the permafrost (which is shallow, smaller than a meter) overnight during the warm season. This does not happen for the warm permafrost. We now phrase this better:

*"For cold permafrost, frozen conditions dominate the first meters of the ground most of the year and surficial thawing during summer **can be interrupted by ground freezing from the surface to the top of the permafrost at night**."*

L406: Here, and throughout the manuscript, if the trend is not significant, avoid presenting trend rates as they do not convey reliable information and can be misleading. See, e.g., Figure 7C.

We removed all y=ax+b and p-values from trends for which the p value was superior to $5 \times 10^{-3}$. We also now mention this threshold in the methods.

L420: if possible, avoid starting your paragraphs with presenting figures. Figures accompany the manuscript text and serve as references confirming your textual statements. When the figure is presented 'as is', decoupled from the main text flow, it loses its reference value. But is a scientific paper, there is no value for a figure other than a reference. Try to better integrate your figures in the text flow. Also, for Figure 7C, add precipitation time series for both high and low evaporation regions (TopoSUB points?).

We complied with the style preferences of the reviewer and reworked the text accordingly. We prefer not to apply the suggestion of the reviewer for figure 7C because we believe that the role of precipitation on evaporation and active layer deepening is covered in the more detailed figure 10C (formerly figure 9C) where the precipitation is represented with the color scale. This allows to not over complicate figure 7C.

L432-433: in other words, locations with average seasonal freezing depth was less than 0.7m, were excluded from calculations? Is it correct?

Rather than the average value, it is if the minimum value of seasonal freezing depth did not reach 0.7m that we excluded the points, we rephrased to:

*"These values average 88% of the no permafrost areas since locations showing persistent thawed conditions at this depth from one year to another were excluded (i.e. minimal seasonal freezing depth over the 40 years lower than 70 cm)."*

L436-437: in evaporation calculations (as well as other hydrological variables though), how were the layer units (mm) obtained? Are they direct model output for a TopoSUB point? Were they averaged over the TopoSUB point representative area?

Yes, for a given TopoSUB point, the model produces hydrological values in $m^3$ using the area of a TopoSUB pixel on the catchment map. We added a paragraph explaining this at the end of section 3.2.4. (model setup and validation):

*"The following method is used to produce area-averaged evaporation and runoff (in mm water equivalent) in a zone of interest. For a given TopoSUB point in this zone, the model produces hydrological values in m3 using the area of a TopoSUB pixel on the catchment map. Then these values are multiplied by the number of pixels in the zone corresponding to this TopoSUB point in particular, and this for all the relevant TopoSUB points covering the zone (e.g. evaporation in warm permafrost). Then the area of interest is calculated by counting the number of pixels in the zone of interest and multiplying this number by the area of a pixel. Then the total volume is divided by the total surface for the zone of interest to obtain the final value in mm"*

L443: Why not runoff coefficient?

We found different definitions of the runoff coefficient online but the most common definition seems to be that the runoff coefficient is calculated as **runoff / (rain + snow)**. For our study we calculate **runoff / (rain + snow – snow sublimation)** which is slightly different. We think it is a more relevant value because it is equivalent to **runoff / (runoff + evaporation)** so it focuses on the precipitation that makes it to the ground surface and splits it between the proportion that can produce runoff towards the lake and the proportion that goes back to the atmosphere through evaporation.

L460, Figure 8C: besides the steady lake level, it could be instructive to present the ratio values explaining the lake level variations, notably its observed gradual decrease since the 1980s. Also, Figure 8D: with +48mm per century trend, we can assume no runoff around 1950s, even earlier for the subsurface runoff.

We understand the point of the reviewer but given the data we have and our framework we are limited in our ability to produce the expected graph. The goal of our graph is to present hydrological outputs from our simulations with a reference point regarding the lake. We suggested this reference level for a steady state lake because we could design it as a value independent of our glacier runoff estimate. To do so, as explained in section 4.3, we assumed that the runoff and evaporation values are correct and that the actual glacier runoff is the runoff we need to add on top of the land runoff to reach the runoff required to close the lake budget (reproducing the lake level observations, as on validation figure 5D). If understood correctly, to add the graph suggested by the reviewer, we would need to use our glacier runoff estimates which are based on only 2 averaged values of glacier mass balance for the whole period and for which uncertainty bars neighbors the size of the values of land runoff. For this reason, we think this additional data on the graph would not improve the presentation of our hydrological results, which is the initial goal of this figure.

We agree that the trends we simulate take high absolute values and most likely express a non-linearity in the evolution of these variables. We added this idea to the first paragraph of Section 4.3. (Hydrological results):

*"**These linear trends we report are high compared to the absolute values of the variables and their extrapolation backward in time would lead to null values in the recent past which is unrealistic. This suggests a non-linear evolution of these variables over the XXth century.**"*

L466: isn't 'liquid/total' more correct, as shown on the Figure 8D?

We applied the correction.

L483: does this mean, that out of 368 TopoSUB points, 92 were classified as 'warm permafrost'? In other words, does 'simulations' refer to 'TopoSUB points' here?

Yes, we replaced simulation by TopoSUB point in the text (which now appears in the discussion, part 5.1.4.)

L484-485: also, AL deepening is associated with low precipitation!

We added "and low precipitation" in the text (same new location as previous point).

L474-476: 'correlation is not causation' holds here, and while Figure 9A shows correlation, it does not necessarily reasonable. What if this is a spurious correlation with precipitation as a driving variable? This must be tested otherwise can be highly misleading (see general comments)

As stated earlier, we understand the concern of the reviewer on this point. Therefore, we moved this outcome of our study in the discussion part and we now use careful wording such as "correlations suggest…" see the response to the main point above.

L498, Figure 9B: under 'no permafrost' condition, there is no seasonal thaw, but rather seasonal freezing. The manuscript contains the data required to produce the correct figure (Appendix E, Figure E, right), but whether such figure is useful, I am not convinced.

See our previous comment on the quantification of seasonal thaw based on ground temperatures.

L498, Figure 9C: see comment on L484-485. Dry locations = less evaporative loss (moisture-limited E) = less latent heat fluxes = higher sensible heating = deeper AL. Sounds plausible to me.

This is indeed the message we want to convey.

Also, combining Figures 9A and 9C, is it so that for the points (years) in Figure 9A, there must have existed points with average P over 400mm and E over 280mm, counterbalanced by points with much lower E values, so that annual E would not exceed 220mm? What are these points?

There is an important North-South precipitation gradient in the catchment. The first points the reviewer mentions are in the north where precipitation can reach 400 mm and evaporation can go beyond 250 mm, whereas in the south some points can receive less than 200 mm per year. Since figure 9A averages the point to point variability to calculate the average values for warm permafrost, this variability is toned down. Like this, the maximal annual evaporation over the 40 years (when averaging all the warm permafrost locations) is lower than the maximum evaporation (averaged over the 40 years) for the topoSUB points where a lot of evaporation is happening.

L510: Sections 5.1.1 and 5.1.2 are unimpressive at best. Yes, we know field data are scarce, but would it be catchier to discuss uncertainties arising from data assimilation techniques, not data absence. Some related questions are listed in the comments above.

Section 5.1.1 has been streamlined and largely reworked. It now includes discussion on the relation between the data and the modeling as suggested by the reviewer. Section 5.1.2 has also been shortened following the later comment of the reviewer on line 539. The beginning of Section 5.1.1. now reads as follow:

*"**Our approach relies on a variety of data regarding their scientific focus (glaciers, ground, lake, atmosphere), their type (in situ observations, remotely sensed data, reanalysis data), their characteristics (point wise data, distributed data, constant or with various time resolution) and the way they interact with our models (model parameters, forcing data, validation data, result data in case of the glacier runoff). Such a diversity arises from our goal to quantify both the ground thermo-hydrological regime and the different terms of the lake budget. This variety also makes it challenging to consistently merge these data into a unique framework. For example, our quantification of the glacier mass change reconstruction is made of two constant values for the study period (1975-2000 and 2000-2020), which limits the relevance of the comparison between the observed lake level variations and the simulated ones.**

**Yet, the lake level variations… [initial text]**"*

L529: Finally, there is no lake level variation curve generated as an outcome from this study, so no, the robustness was not evaluated against this directly observed variable.

The lake level variations are now presented in the result part of the revised manuscript (Sec. 4.4).

L532: in fact, not; red curve is not lake level fluctuations, but runoff required to close the observed annual water balance.

*The initial sentence in the submitted manuscript is:*

*"We combine lake level observations with our precipitation forcing data and lake evaporation quantifications in a simple mass conservation calculation to derive the land runoff to the lake required to reproduce the level variations (red curve on Fig. 5)"*

*Hence the "red curve" does not refer to "the level variations" but rather to "the land runoff to the lake required to reproduce the level variations", which is correct.*

L539: Water routing has minor importance on annual timescale (you admit it in L546-548). This paragraph can be omitted from the manuscript. In L544-545, the 95% argument is reiterated though it was just evoked in L532-533 to support the correctness of the magnitude.

*We deleted the paragraph.*

L578-579: Figure 9B is unrelated to frozen water content, maybe Figure 9A? Figure 8D looks contradictory in this scope; although it refers to the whole catchment dominated by non-permafrost areas.

*Indeed, the mention should have gone to Fig 9A, which is now Fig. 10A in the revised manuscript. We corrected this mistake.*

L671-672: this effect was not modeled by Wang et al. 2018 but it is represented in several global climate models in this way under RCP4.5 (see, e.g., their Table 3, p. 1159).

*From our understanding of this article, we believe this is incorrect. Wang et al. (2018) used outputs from GCM models to force their GBEHM model which calculates both phase changes at the surface and subsurface and runoff (among other physical values, see their section 3.2 that starts on page 1156).*

L721: Sections 5.4 and 6 are repetitive, they can be merged into one, otherwise, provide mode discussion concerning lake level changes in the respective section.

*Following the suggestion from the reviewer, we merged the two parts into a conclusion.*

L724-726: modeled data can not lead to observed lake level change. Also, how modeled data drives modeled lake level change, is not presented in the manuscript (a major flaw).

*We rephrased for:*

*"The sum of the direct precipitation in the lake, the land runoff and the glacier runoff are not enough to compensate for the lake evaporation over the study period, hence driving the observed lake level decrease."*

*Modeled lake level changes are now presented and discussed in the revised version of the manuscript.*

L730: 'affecting' stands for 'increasing' here? Also, L733-734 is not about change in permafrost but the presence of permafrost, which is different.

*Yes, we changed "affecting" for "increasing".*

*We also agree with the second sub-comment from the reviewer but we do not see how it can be seen as problematic, as we are simply listing results.*

L727-728 and L733-734 need to be consistent and better supported by results/discussion. E.g., the sequence "catchment loses permafrost (Figure 7D) = less ET (L733) = increase in P (Figure 3) = increase in runoff & runoff ratio (Figure 8B)" might be incorrect straight away because E is also increasing catchment-wise. But where ET is increasing most? There is no answer in Figure 8A nor in the manuscript text. Does the increase in ET coincide with TopoSUB points where permafrost was lost? The answer is relatively easy to answer.

*We are not sure to fully understand the reviewers point here. Regarding L727-728. We rephrased the sentence in a more cautious and complete fashion:*

*"Long-term hydrological trends in the catchment are led by trends in climate; **and precipitation increase, jointly with glacier melt, provides enough water to drive a concomitant increase of runoff and evaporation.**"*

*Regarding the second point, L733-734 is merely a summary of the sensitivity test presented in Section 5.3. (Evaporation vs runoff and sensitivity to climate conditions). It is written with careful wording and neither adds new discussion elements nor new implications on top of the simple results of this test (Fig. 11).*

L755: not where it is limited, but just were it is 'relatively' low compared to other TopoSUB points, for whatever reason; Figure 9C suggests that the main reason is low precipitation amount. Both figures are for warm permafrost.

*We changed "limited" for "relatively low"*

L766-767: see above.

Here again, we do not see why describing a result related to the presence or absence of permafrost poses a problem.

**References**

[revised manuscript text omitted]

---

## Author Comment (AC2)

We are very grateful to Reviewer 2 for the review we received. We present below our detailed answer to the discussed points. The reviewers' comments appear in black and our responses appear in brown. Quotes from the manuscript are in *brown italic*, modified parts from the revised version are *in bold font*.

**Review 2**

Martin, Immerzeel and their colleagues conducted a study to understand recent ground thermohydrological changes in a Tibetan endorheic catchment and implications for lake level changes. The authors did a lot of work on using model to understand the cold region hydrology. I think this is a comprehensive modeling study. The conceptual hydrological framework includes precipitation downscaling, remote sensing glacier observation, catchment hydrology, lake evaporation and water balance. For land hydrology, the authors used the TopoSUB to delineate catchment into different units, and used CryoGrid to simulate the complicated frozen soil hydrology. The presentation is clear for most part, and the study has kind of novelty. But there are still some major issues the authors should address before considering for publication.

The definition of surface and subsurface flow. The authors defined the runoff from top 0.3m soil as surface runoff, and 0.3-2m as subsurface runoff. This definition might need more rigorous study.

We understand the reviewer's concern on the reader's understanding of the model. In the model, the distinction between surface and subsurface runoff is not based on the stratigraphy but on the balance between soil water content and new water input. We now provide further explanations on this matter in section 3.2.4 (Model setup and validation) as presented in the quotation responding to the next comment.

In hydrology, the separation of surface and subsurface flow is a grand challenge, which is still not well solved in moderate climate catchments (McDonnell, 2013), not to mention this data scarce permafrost region. Also the surface water and subsurface water have different behaviors in different topography (Seibert et al., 2003; Gao et al., 2014), e.g. on hillslope or riparian area. I did find how the CryoGrid model takes this into account.

We agree with the reviewer and to better acknowledge the relative simplicity of our approach compared to specific literature on this point, we modified section 3.2.4 (Model setup and validation). The following quotation of the revised manuscript thus answers this comment and the previous one.

"Regarding the processes implemented in the model (Sect. 3.2.3), infiltration according to Richards equation only occurs in the top and bottom soil units. The bedrock unit has a static water content. **Unraveling surface** from subsurface flow is an ongoing challenge in catchment-scale hydrology (McDonnell, 2013) and this distinction is important in mountain terrains where these two flows can behave differently due to the complex topography (Gao et al., 2014; Seibert et al., 2003). For this study, we rely on a simple approach that computes surface and subsurface flow as follows.

On the one hand, surface runoff is computed relative to the saturation level of the soil column. When the entire soil column is saturated (WC = porosity), additional water input from precipitation or snowmelt is directly counted as surface runoff. On the other hand, subsurface runoff is computed relative to the field capacity of the ground, which is an input parameter of the model. When the water content (WC) of a ground cell exceeds this field capacity (FC), the amount of water corresponding to WC-FC is available to produce subsurface runoff. We use the lateral boundary condition LAT\_WATER\_RESERVOIR from the CryoGrid community model (Westermann et al., 2022) to account for this subsurface runoff. The speed at which this available water exits the soil column towards the lake is calculated with Darcy's law, using the hydrological conductivity of the ground and the mean slope of the catchment as hydraulic slope."

I did not see how the model takes the impacts of frozen soil on the connectivity between surface soil and groundwater. For example, in the early thawing seasons, although the top soil is thawed, but there is still frozen soil underneath, which inhibited the soil and groundwater connection. Also the impacts of frozen soil on groundwater discharge, i.e. the baseflow. These processes have huge impacts on catchment hydrology (Gao et al., 2022).

The CryoGrid community model aims at coupling the heat and water fluxes in the ground. In this regard, the hydraulic conductivity of each cell of the soil column during the simulation accounts for both the saturation of water and ice following Dall'Amico et al. (2011). In this approach, we first calculate the hydraulic conductivity based on liquid water saturation according to Mualem (1976) and then reduce the obtained value with an impedance factor equal to  $10^{-\omega q}$  with  $\omega$  being an empirical factor (that equals 7, following Zhao et al., 1997 and Hansson et al., 2004) and with q being the ice saturation. We now provide a short presentation of this approach in section 3.2.3 (The CryoGrid community model) of the revised manuscript :

"The soil matric potential and hydraulic conductivity follow van Genuchten, (1980) and Mualem (1976). Additionally, to represent the obstruction of connected porosity by ice formation, the hydraulic conductivity is reduced by a factor dependent on the local ice content, following Dall'Amico et al. (2011).".

Additionally, we want to clarify the fact that, consistently with the conceptual hydrological framework of our study (Fig. 2 of the revised and former manuscript), our setup does not include other groundwater components than the subsurface runoff. As such, the temperature-and-water-dependent calculation of the hydraulic conductivity applies for all the hydrologically active ground cells and drives variations in connectivity in response to variations of temperature around 0°C.

"physics-based approaches at the catchment scale aiming to connect the ground thermo-hydrological regime and the observed hydrological changes on the QTP changes remain scarce." "To our best knowledge, our study represents to date the most complete effort to include the variety of coupled climatological, surface and subsurface processes characterizing the climate, hydrology and ground thermal regime of high-mountain catchments in Tibet at a small scale with a high spatial resolution." These might be true when the authors wrote the paper, but I would like to recommend this new paper (Gao et al., 2022) for the authors' reference.

We agree with the reviewer, we re-phrased this sentence to the following:

"To our best knowledge, **along with Gao et al. (2022)** our study represents to date the most complete effort to include the variety of coupled climatological, surface and subsurface processes characterizing the climate, hydrology and ground thermal regime of high-mountain catchments in Tibet at a small scale with a high spatial resolution."

I did not find how the soil evaporation and plant transpiration were estimated. These processes are very important for water balance, especially for this basin, with only 10% precipitation generates runoff (as mentioned by the authors), and 90% goes back to atmosphere.

Potential evapotranspiration is derived from the latent heat fluxes in the frame of the surface energy balance calculation (using the latent heat of evaporation). This potential evaporation is then scaled to the available water in the ground . It occurs in the first grid cell only, but water can be drawn upwards due to matric potential differences We clarified this point in section 3.2.3 (The CryoGrid community model):

"At the surface, the model uses a surface energy balance module to calculate the ground surface temperature and water content. The turbulent fluxes of sensible and latent heat are calculated using a Monin– Obukhov approach (Monin and Obukhov, 1954). Evaporation is derived from the latent heat fluxes using the latent heat of evaporation and is adjusted to the available water in the soil. It occurs in the first grid cell only, but water can be drawn upwards due to matric potential differences."

Regarding the unraveling of evaporation and transpiration, we acknowledge that it is an important question in most hydrological studies, yet in the case of the Paiku catchment, vegetation is extremely scarce. During our field trip, we noted that most of the catchment corresponds to barren lands and that vegetation is limited to very sporadic herbaceous cover. Therefore, we do not expect transpiration to have a strong imprint on evapotranspiration in the catchment. For this reason, our approach does not separate evaporation from transpiration in our calculations. To include these considerations in the manuscript, we added the following sentence to the precedent quote:

"Because vegetation is very scarce in the catchment, we do not expect transpiration to have a strong imprint on evapotranspiration and our calculations do not unravel evaporation from transpiration."

We present below a photo to show the type of land cover we found on the field that supports this assumption.

Figure R2.1 Ground photo in the Paiku catchment (Credit: Fanny Brun).

Some terms need to be improved. For example, in Figure 5D, the y-axis should be changed to "Runoff depth (mm/y)".

**We changed the label of the y-axis according to the reviewer's suggestion.**

Gao, H., Hrachowitz, M., Fenicia, F., Gharari, S., and Savenije, H. H. G. (2014) Testing the realism of a topography-driven model (flex-topo) in the nested catchments of the upper Heihe, china, Hydrology and Earth System Sciences, 18, 1895-1915, 10.5194/hess-18-1895-2014.

Gao, H., Han, C., Chen, R., Feng, Z., Wang, K., Fenicia, F., and Savenije, H.: Frozen soil hydrological modeling for a mountainous catchment northeast of the Qinghai–Tibet Plateau, Hydrol. Earth Syst. Sci., 26, 4187–4208, https://doi.org/10.5194/hess-26-4187-2022, 2022.

McDonnell, J.J., Are all runoff processes the same? Hydrol. Process. 27, 4103-4111 (2013)

Seibert, J. Rodhe, A., and Bishop, K. Simulating interactions between saturated and unsaturated storage in a conceptual runoff model. Hydrol. Process. 17, 379–390 (2003)

**References**

- Dall'Amico, M., Endrizzi, S., Gruber, S. and Rigon, R.: A robust and energy-conserving model of freezing variablysaturated soil, Cryosph., 5(2), 469–484, doi:10.5194/tc-5-469-2011, 2011.
- Gao, H., Hrachowitz, M., Fenicia, F., Gharari, S. and Savenije, H. H. G.: Testing the realism of a topography-driven model (FLEX-Topo) in the nested catchments of the Upper Heihe, China, Hydrol. Earth Syst. Sci., 18(5), 1895–1915, doi:10.5194/hess-18-1895-2014, 2014.
- Gao, H., Han, C., Chen, R., Feng, Z., Wang, K., Fenicia, F. and Savenije, H.: Frozen soil hydrological modeling for a mountainous catchment northeast of the Qinghai–Tibet Plateau, Hydrol. Earth Syst. Sci., 26(15), 4187–4208, doi:10.5194/hess-26-4187-2022, 2022.
- van Genuchten, M. T.: A Closed-form Equation for Predicting the Hydraulic Conductivity of Unsaturated Soils, Soil Sci. Soc. Am. J., 44(5), 892–898, doi:10.2136/sssaj1980.03615995004400050002x, 1980.
- Hansson, K., Šimůnek, J., Mizoguchi, M., Lundin, L. and Genuchten, M. T.: Water Flow and Heat Transport in Frozen Soil: Numerical Solution and Freeze–Thaw Applications, Vadose Zo. J., 3(2), 693–704, doi:10.2136/vzj2004.0693, 2004.
- McDonnell, J. J.: Are all runoff processes the same?, Hydrol. Process., 27(26), 4103–4111, doi:10.1002/hyp.10076, 2013.
- Monin, A. S. and Obukhov, A. M.: Basic laws of turbulent mixing in the surface layer of the atmosphere, Contrib. Geophys. Inst. Acad. Sci. USSR, 151, 163–187, 1954.
- Mualem, Y.: A new model for predicting the hydraulic conductivity of unsaturated porous media, Water Resour. Res., 12(3), 513–522, doi:10.1029/WR012i003p00513, 1976.
- Seibert, J., Rodhe, A. and Bishop, K.: Simulating interactions between saturated and unsaturated storage in a conceptual runoff model, Hydrol. Process., 17(2), 379–390, doi:10.1002/hyp.1130, 2003.
- Westermann, S., Ingeman-Nielsen, T., Scheer, J., Aalstad, K., Aga, J., Chaudhary, N., Etzelmüller, B., Filhol, S., Kääb, A., Renette, C., Schmidt, L. S., Schuler, T. V, Zweigel, R. B., Martin, L., Morard, S., Ben-Asher, M., Angelopoulos, M., Boike, J., Groenke, B., Miesner, F., Nitzbon, J., Overduin, P., Stuenzi, S. M. and Langer, M.: The CryoGrid community model (version 1.0) -- a multi-physics toolbox for climate-driven simulations in the terrestrial cryosphere, Geosci. Model Dev. Discuss., 2022, 1–61, doi:10.5194/gmd-2022-127, 2022.
- Zhao, L., Gray, D. M. and Male, D. H.: During Infiltration Into Frozen Ground, J. Hydrol., 200, 345–363, 1997.

---

## Author Response (AR2)

**Referee 1**

I thank the authors for taking well of my suggestions, although I am still not fully convinced with the definition of surface and sub-surface runoff in this paper. The definition to separate surface and subsurface is important to draw one of the conclusions that "ground warming drives a strong increase in subsurface runoff", which is important for Tibetan Plateau hydrology study and decision making.

**We are grateful for the reviewer's comments and we understand his will to make precise and clear statements regarding surface and subsurface runoff, as well as connections with ground warming. It is also something we are aiming for with this study. With our model, we consider that surface runoff occurs when precipitation falls over a saturated ground. In comparison, subsurface runoff occurs for a soil water content lower than saturation, when the soil water content is above field capacity, because the slope drains it towards the lake. These approaches can be found in various publications (Samuel et al., 2008; Vörösmarty et al., 1989; Shaman et al., 2002; Kelleners et al., 2010; Kampf, 2011) and are consistent with field observations (Lai et al., 2018). We made clarifications in the text as detailed at the end of this answer.**

If more water goes to subsurface, with the same amount of total runoff, surface runoff will decrease. Surface runoff is largely related to sediment transport, thus decreased sediment yield can be expected in the case. However, dramatic increase of sediment yield was observed on the TP (https://www.science.org/doi/10.1126/science.abi9649). These two conclusions (more subsurface runoff and increase of sediment) seem like a paradox.

**We thank the reviewer for this interesting perspective on the distribution between surface and subsurface runoff and the connection with sediments. As pointed out by the reviewer, the paradox he mentions might only occur for a constant level of total runoff over time. Yet we believe that such an hypothesis on the total runoff might not be the most relevant given the climate changes observed and forecasted for High Asia. Indeed, the study from Li et al. (2021) mentioned by the reviewer suggests that climate change in High Asia implies a precipitation increase over time, which contributes to an overall increase in runoff. At the scale of the Paiku catchment, we report the same trend : an increase in precipitation that is strong enough to drive a concurrent increase of surface and subsurface runoff. To acknowledge connections between sediment transport and runoff in the study we added the following line to Sect. 5.2.2. (Evaporation and runoff changes):**

**_"These increases in runoff (especially surface runoff) are likely to have an influence on sediment transport. For instance, Li et al. (2021) showed that current precipitation augmentation over High Mountain Asia is driving a runoff increase, which contributes to a significant rise in fluvial sediment fluxes."_**

I agree that the surface and subsurface runoff are well connected, and in many cases are hard to be isolated. The key question might be "why do we need to split it apart?", since it is an integrated system in nature. The authors can find more detailed discussion about this issue and relevant topics in our HESSD Opinion paper (https://egusphere.copernicus.org/preprints/2023/egusphere-2023-125/). I don't want to use this reason to stop the publication of this paper. For me, the authors did lots of modeling work, and this is an excellent case study on frozen soil hydrology modeling in a small catchment on the TP. I'd like to suggest further improving the description and discussion on this critical issue before acceptance.

**We were not aware of this discussion and we thank the reviewer for pointing it out. We will keep in mind the possibility of an environment-driven approach rather than a soil-driven approach in the future. We believe that this is an open scientific debate and we are happy to bring here some arguments supporting our approach. We see several reasons to separate surface and subsurface runoff that we develop in the new paragraph we added to the discussion to account for the reviewer's comment. To summarize it, we believe that surface and subsurface flow have contrasted behaviors that reflect the different physical mechanisms that drive them, both hydrologically and thermally. And it seems to us that this is confirmed by field observations on Tibetan hillslopes (Hu et al., 2020). In turn, these contrasted behaviors have consequences on the rest of the hydrosystem, on the landscape and on the environments and for this reason we believe it is a relevant distinction for our work.**

**To acknowledge the suggestion of the reviewer to further improve the description and the discussion , the main text has been modified as follows.**

**In _Model Setup and Validation_ (Sect 3.2.4):**

**_"For this study, we rely on a simple approach that is based on thresholds regarding the soil water content (porosity and field capacity). This kind of approaches are thus based on soil properties and have been often_**

*used in hydrological modeling studies (Vörösmarty et al., 1989; Shaman et al., 2002; Kelleners et al., 2010; Kampf, 2011; Samuel et al., 2008). In detail, we compute surface and subsurface flow as follows.*"

In *Modeling Strategy* (Sect 5.1.2), entirely new paragraph:

*"Additionally, our approach regarding the modeling of runoff is relatively simple, i.e. partition between subsurface and surface runoff based on comparison between the soil water content and field capacity and porosity, respectively. More complex approaches split runoff into more sophisticated categories such as Horton overland flow, Dunne overland flow, subsurface stormflow… (e.g. Savenije, 2010; Gao et al., 2014; Mirus and Loague, 2013). However, over the last decade, the relevance of this type of partitioning between different types of runoff has been questioned (McDonnell, 2013; Gao et al., 2023) . In the frame of our study, we find it important to distinguish between surface and subsurface runoff because they generate flows with very contrasted speed. In a general perspective, this significant difference in flow velocities impacts the hydrological system as a whole (e.g. river discharge, evaporation…) and has various consequences throughout the catchment, such as the water availability for vegetation, erosion and sediment transport.*

*In the particular case of a cryo-hydrological study, separating surface from subsurface runoff is particularly relevant because both flows do not react in the same way to ground temperature changes. As such, we see our approach as a middle way that allows us to make this distinction based on simple hydrological considerations. Yet, we acknowledge that the classification and quantification of the different types of runoff represent a valuable direction for future investigation on catchment-scale cryo-hydrology in Tibet."*

**References**

Gao, H., Hrachowitz, M., Fenicia, F., Gharari, S., and Savenije, H. H. G.: Testing the realism of a topography-driven model (FLEX-Topo) in the nested catchments of the Upper Heihe, China, Hydrology and Earth System Sciences, 18, 1895–1915, https://doi.org/10.5194/hess-18-1895-2014, 2014.

Gao, H., Fenicia, F., and Savenije, H. H. G.: HESS Opinions: Are soils overrated in hydrology?, Catchment hydrology/Theory development, https://doi.org/10.5194/egusphere-2023-125, 2023.

Hu, G.-R., Li, X.-Y., and Yang, X.-F.: The impact of micro-topography on the interplay of critical zone architecture and hydrological processes at the hillslope scale: Integrated geophysical and hydrological experiments on the Qinghai-Tibet Plateau, Journal of Hydrology, 583, 124618, https://doi.org/10.1016/j.jhydrol.2020.124618, 2020.

Kampf, S. K.: Variability and persistence of hillslope initial conditions: A continuous perspective on subsurface flow response to rain events, Journal of Hydrology, 404, 176–185, https://doi.org/10.1016/j.jhydrol.2011.04.028, 2011.

Kelleners, T. J., Chandler, D. G., McNamara, J. P., Gribb, M. M., and Seyfried, M. S.: Modeling Runoff Generation in a Small Snow-Dominated Mountainous Catchment, Vadose Zone Journal, 9, 517–527, https://doi.org/10.2136/vzj2009.0033, 2010.

Lai, X., Zhou, Z., Zhu, Q., and Liao, K.: Comparing the spatio-temporal variations of soil water content and soil free water content at the hillslope scale, CATENA, 160, 366–375, https://doi.org/10.1016/j.catena.2017.10.008, 2018.

Li, D., Lu, X., Overeem, I., Walling, D. E., Syvitski, J., Kettner, A. J., Bookhagen, B., Zhou, Y., and Zhang, T.: Exceptional increases in fluvial sediment fluxes in a warmer and wetter High Mountain Asia, Science, 374, 599–603, https://doi.org/10.1126/science.abi9649, 2021.

McDonnell, J. J.: Are all runoff processes the same?, Hydrological Processes, 27, 4103–4111, https://doi.org/10.1002/hyp.10076, 2013.

Mirus, B. B. and Loague, K.: How runoff begins (and ends): Characterizing hydrologic response at the catchment scale: HOW DOES RUNOFF BEGIN (AND END)?, Water Resour. Res., 49, 2987–3006, https://doi.org/10.1002/wrcr.20218, 2013.

Samuel, J. M., Sivapalan, M., and Struthers, I.: Diagnostic analysis of water balance variability: A comparative modeling study of catchments in Perth, Newcastle, and Darwin, Australia: DIAGNOSTIC ANALYSIS OF WATER BALANCE VARIABILITY, Water Resour. Res., 44, https://doi.org/10.1029/2007WR006694, 2008.

Savenije, H. H. G.: HESS Opinions "Topography driven conceptual modelling (FLEX-Topo)", Hydrol. Earth Syst. Sci., 14, 2681–2692, https://doi.org/10.5194/hess-14-2681-2010, 2010.

Shaman, J., Stieglitz, M., Engel, V., Koster, R., and Stark, C.: Representation of subsurface storm flow and a more

responsive water table in a TOPMODEL-based hydrology model: TOPMODEL STORM FLOW AND A MORE REACTIVE WATER TABLE, Water Resour. Res., 38, 31-1-31–16, https://doi.org/10.1029/2001WR000636, 2002.

Vörösmarty, C. J., Moore, B., Grace, A. L., Gildea, M. P., Melillo, J. M., Peterson, B. J., Rastetter, E. B., and Steudler, P. A.: Continental scale models of water balance and fluvial transport: An application to South America, Global Biogeochem. Cycles, 3, 241–265, https://doi.org/10.1029/GB003i003p00241, 1989.

**Referee 2**

Major comments:

1. The title and abstract of this manuscript are likely to focus on whole Tibetan endorheic catchments. However, the major contents of the article focused on a typical lake with seasonal frozen ground, which is different from the most lakes on the north of the QTP with permafrost. Thus, the title and abstract should be revised to match the content of the article.

**Our study focuses on the Paiku catchment and lake. The title mentions "*a Tibetan endorheic catchment*", which indicates that it is a catchment scale study. Similarly, the abstract says "*This study focuses on the cryo-hydrology of the catchment of Lake Paiku*". Broader topics are tackled in the introduction and the discussion in order to connect our work with the rest of the literature and broad scientific questions, which we believe to be a common practice in scientific articles.**

**To make things even clearer we added the following elements to the title and abstract :**

**Title**

**"*Recent ground thermo-hydrological changes in a Southern Tibetan endorheic catchment and implications for lake level changes*"**

**Abstract**

**"*Although the present study was performed at catchment scale,* *we suggest that this ambivalent influence of permafrost may help to understand the contrasting lake level variations observed between the South and North of the QTP, opening new perspectives for future investigations.*"**

2. There are two critical assumptions that may cause larger errors and imbalance of water for the lake: (1) The water flows between the lake and potential aquifers surrounding are negligible due to it is difficult to quantify these flows. Because there is undoubtedly more potential water flows in nature, it is necessary to deeply discuss the errors or uncertainty caused by this assumption.

**We acknowledge the point of the reviewer on lake aquifer-interactions. For this study we developed a rather complex hydrological framework and this process is indeed absent from it. Additionally, we have no way to quantify the magnitude of these interactions and we think investigations in this direction would go beyond the scope of our study. Regarding the discussion, we have now extended the paragraph we had on this topic in sect. 5.1.3 (Reconstruction of the Lake hydrological budget and level variations). It now reads:**

**"*A possible reason for this mismatch is that the lake is connected to a larger aquifer that surrounds it. In the context of a decreasing lake level, an aquifer surrounding the lake can create an additional water inflow when the lake level passes below the piezometric level of the aquifer (Yechieli et al., 1995). Such an inflow could mitigate the lake level decrease and thus explain the missing water in our reconstruction (Fig. 6B). It could also explain the gradual stabilization of the lake level that our model does not reproduce. This flow is not part of our conceptual hydrological framework even though it likely exists in reality, especially since there is no permafrost near the lake (as we simulate it here), allowing for the existence of such an aquifer (Walvoord and Kurylyk, 2016). Groundwater has been identified as a potential contributor to lake level rise in other regions of the QTP (Lei et al., 2022). In the long run, lake-aquifer systems commonly follow oscillations of the net atmospheric flux of water (Precipitation – Evaporation) and of the runoff that forces its mass balance (Watras et al., 2014). During these oscillations, the lake can "pump" water from the aquifer or feed it depending on the relative difference of piezometric level between them (Almendinger, 1990; Liefert et al., 2018). Yet, this potential effect is difficult to account for and its magnitude remains unclear. Therefore, the reasons for the mismatch between observed and simulated lake levels could also be connected to other aspects of our methodology such as bias in the climatic forcing data and other shortcomings arising from the lack of field data, or hydrological processes, as developed in Sect. 5.1.1 and 5.1.2.*"**

(2) Because vegetation is very scarce in the catchment, the vegetation transpiration is ignored in evapotranspiration. I think this assumption may cause more errors and uncertainty. There were some researches reported that alpine meadow or swamp meadow with high evapotranspiration are distributed in a certain range around a lake and along a river, and vegetation transpiration and interception of alpine meadow and swamp meadow could account for 30-40% of the total evapotranspiration of the grassland area on the plateau.

**We understand the reviewers' concern on transpiration, yet we want to insist on the fact that vegetation is extremely scarce in the catchment. During our field trip, we noted that most of the catchment corresponds to barren lands (Fig. R2.1) and that vegetation is limited to very sporadic herbaceous cover. This assessment is confirmed when looking at satellite images of the catchment, which do not show any noticeable vegetated**

area, as well as NDVI values that correspond to barren land in the region (Liu et al., 2022; Mao et al., 2022). Therefore, we believe we have a meaningful argument to support the idea that transpiration is very unlikely to have a strong imprint on evapotranspiration in the catchment. We present below a photo that shows the type of land cover we found on the field that supports this assumption.

[Figure]

*Figure R2.1. Ground photo in the Paiku catchment in November 2019 (Credit: Fanny Brun).*

Additionally, the evaporation rates we report are extremely high (nearly 90% of the precipitation reaching the ground is evaporated in our historical scenario). Thus we think it is quite unlikely that our simulations underestimate evaporation. Nevertheless, to acknowledge the reviewer comments we added the following to the discussion in the *Modeling strategy* part (Sect. 5.1.2):

*"Another potential improvement in our modeling approach could be to unravel evaporation from transpiration. However, since vegetation is extremely scarce in the Paiku catchment, which is largely dominated by barren lands, we suggest that this would not significantly affect our results. However, this limitation should be explored in future field and modeling studies."*

3. The data used in model verification is too less to meet the requirements of model parameter calibration and simulation result verification. In a catchment area of 2 400 km2 with permafrost and glaciers distributed in high altitude areas, there is only one observation point at seasonal frozen ground to support the data of temperature, precipitation and ground temperature required by the CryoGrid community model simulation. Also, there are no observations in the lake area to use for lake water balance analysis.

We understand the reviewer's comment and we are aware that the amount of observations we have to frame our simulations is an important question for our study. Yet, we need to correct what is said here. On top of the data from our automatic weather station, our ground temperature loggers and geodetic mass balance reconstructions, we have access to very precise lake level observations from Lei et al. (2018, 2021). So in the end, we use all observations available, including lake level, which bring us observations on the water balance. In turn, any study in the Paiku catchment will have the same limitations. To clarify the existence and the use of the lake level data we modified the last paragraph of Section 2 (Study area: the Paiku catchment).

*"More recently, the lake level decreased by 3.7 m between 1972 and 2015, losing 4.2% of its surface and 8.5% of its volume. Measurements have been performed since the end of the 1970s and allow to accurately know the evolution of the lake level until today (Lei et al., 2021, 2018), they are used in this study to validate our hydrological results (Sect 3.2.1, Fig. 5D and 6B)."*

Additionally, none of our modeling works are driven by these lake observations, we use them to compare our simulated lake balance with the observed one, which provides validation to assess the model performances. In this regard, our calculations produced 95% of the runoff required to reproduce the lake variations, indicating that the magnitude of our reconstruction is correct. From the thermal point of view, on top of reproducing logger temperature values, our simulations find a very good consistency with larger scale studies covering the same area (such as studies tackling permafrost coverage, Sect. 5.2.1). Considering that our approach tackles water and heat flows and in a coupled and interdependent way, we think that providing this two-fold agreement with observations and other works brings confidence in the robustness of the reported results. To clarify this role of the lake observations we modified the last paragraph of Sect. 3.2.1 (Conceptual hydrological model for the catchment):

*"Our catchment-scale approach to represent the hydrological balance of the lake is summarized in Fig. 2. Based on this approach, we can evaluate the performance of our framework (Sect. 4.1.2), by comparing the simulated lake balance with the one derived from the detailed observations of lake level variations over the study period (Lei et al., 2018, 2021)."*

Additionally, we want to point out that, even though Tibet deserves major attention from hydrologists and cryosphere scientists, it is very challenging to acquire data in an area like the catchment of the Paiku lake.

It is a very remote region, hard to access for logistical reasons, even for Chinese scientists. And in this particular context, the COVID pandemic made it even harder to access the field. This limited our ability to collect more field observations. We are fully aware that our analysis contains large uncertainties and have therefore included a detailed discussion section ( Sect. 5.1.1 Data usage within the conceptual framework and data scarcity, which used to be even more extensive before the reviewer 1 of the first review round recommended to shorten it) in which we present different possible interpretations of our modeling result, in the light of the sparse available observations. To clarify this point we amended the last paragraph of this section:

*"A main limitation regarding our usage of the data is related to the limited amount of available field observations required to provide robust model parameterizing, climate forcing and in-depth validation of the simulations, both hydrologically and thermally. Regarding climatic forcing data, our AWS measurement offers sound observations to evaluate and adjust the ERA5 data processed with TopoSUB and downscaled with TopoSCALE. Yet, a period of observations longer than 2 years would have enabled more robust corrections and could have allowed us to perform a more advanced statistical downscaling approach, e.g. quantile mapping (Themeßl et al., 2011). As such, the spatiotemporal domain of relevance of these corrections is insufficient to correct data for the whole catchment and the 40 years of simulations. Overall, considering the strong bias we observe in the raw ERA5 data (Figure D0), these corrections do represent an important first-order improvement. Altogether, this scarcity of field observations is likely to bring significant uncertainties to our analysis. Future efforts should focus on acquiring additional data or developing validation methods based on remotely sensed observations."*

As indicated in the paper, based on all these considerations, we believe that the following conclusions can be drawn in the light of this uncertainty:

- Lost of the permafrost extent (20% loss)
- Average ground warming around 1.7°C per decade (at 2m deep)
- Increased duration of seasonal thaw (mainly due to later end date of the thaw period)
- Evaporation acts as an energy sink limiting active layer deepening
- Increase in evaporation, surface and subsurface runoff
- Increase of the runoff/(runoff+evaporation) ratio
- Connections between permafrost disappearance and subsurface runoff increase
- Increased availability of liquid water in the ground connected to higher evaporation rates
- Precipitation increase drive a concomitant increase of runoff and evaporation
- Potentially ambivalent influence of permafrost on evaporation, that seems to be climate-dependent.

Altogether, we believe that our results shed light on important cryo-hydrological trends that have the potential to foster new research and improve our understanding of the impact of climate changes on High Mountain Asia and particularly on the understanding of the lake variations across the QTP.

Minor comments:

Abstract: Larger part related to the introduction, however, some key results and conclusion are deficiency.

We believe it is natural to connect the detailed work we did in the Paiku catchment with larger scale scientific questions. Therefore the first paragraph of the abstract aims at framing our study and demonstrating its relevance. Based on the reviewer comments, we have shorten this paragraph, which now reads:

*"Climate change modifies the water and energy fluxes between the atmosphere and the surface in mountainous regions such as the Qinghai-Tibet Plateau (QTP), which has shown substantial hydrological changes over the last decades, including rapid lake level variations. The ground across the QTP hosts either permafrost or seasonally frozen and, in this environment, the ground thermal regime influences liquid water availability, evaporation and runoff. Therefore, climate-driven modifications of the ground thermal regime may contribute to lake level variations, yet this hypothesis has been relatively overlooked until now."*

The final part of the abstract is based on the results and discussion sections. To make this clearer, we modified the following in the abstract:

*"Our results show that both seasonal frozen ground and permafrost…"*

L25-27: The last sentence is not suitable and lack of sufficient supports.

This last sentence of the abstract summarizes considerations from the discussion where we try to be prospective and aim at connecting our study with broader questions. To make this clear, we now phrase it

with even more caution. The new version is already quoted higher up, in the answer to the major point number 1.

**References**

Almendinger, J. E.: Groundwater control of closed-basin lake levels under steady-state conditions, Journal of Hydrology, 112, 293–318, https://doi.org/10.1016/0022-1694(90)90020-X, 1990.

Lei, Y., Yao, T., Yang, K., Bird, B. W., Tian, L., Zhang, X., Wang, W., Xiang, Y., Dai, Y., Lazhu, Zhou, J., and Wang, L.: An integrated investigation of lake storage and water level changes in the Paiku Co basin, central Himalayas, Journal of Hydrology, 562, 599–608, https://doi.org/10.1016/j.jhydrol.2018.05.040, 2018.

Lei, Y., Yao, T., Yang, K., Ma, Y., and Bird, B. W.: Contrasting hydrological and thermal intensities determine seasonal lake-level variations – a case study at Paiku Co on the southern Tibetan Plateau, Hydrology and Earth System Sciences, 25, 3163–3177, https://doi.org/10.5194/hess-25-3163-2021, 2021.

Lei, Y., Yang, K., Immerzeel, W. W., Song, P., Bird, B. W., He, J., Zhao, H., and Li, Z.: Critical Role of Groundwater Inflow in Sustaining Lake Water Balance on the Western Tibetan Plateau, Geophysical Research Letters, 49, https://doi.org/10.1029/2022GL099268, 2022.

Liefert, D. T., Shuman, B. N., Parsekian, A. D., and Mercer, J. J.: Why Are Some Rocky Mountain Lakes Ephemeral?, Water Resour. Res., 54, 5245–5263, https://doi.org/10.1029/2017WR022261, 2018.

Liu, Y., Li, Z., Chen, Y., Li, Y., Li, H., Xia, Q., and Kayumba, P. M.: Evaluation of consistency among three NDVI products applied to High Mountain Asia in 2000–2015, Remote Sensing of Environment, 269, 112821, https://doi.org/10.1016/j.rse.2021.112821, 2022.

Mao, X., Ren, H.-L., and Liu, G.: Primary Interannual Variability Patterns of the Growing-Season NDVI over the Tibetan Plateau and Main Climatic Factors, Remote Sensing, 14, 5183, https://doi.org/10.3390/rs14205183, 2022.

Themeßl, M. J., Gobiet, A., and Leuprecht, A.: Empirical-statistical downscaling and error correction of daily precipitation from regional climate models, International Journal of Climatology, 31, 1530–1544, https://doi.org/10.1002/joc.2168, 2011.

Walvoord, M. A. and Kurylyk, B. L.: Hydrologic Impacts of Thawing Permafrost—A Review, Vadose Zone Journal, 15, 0, https://doi.org/10.2136/vzj2016.01.0010, 2016.

Watras, C. J., Read, J. S., Holman, K. D., Liu, Z., Song, Y.-Y., Watras, A. J., Morgan, S., and Stanley, E. H.: Decadal oscillation of lakes and aquifers in the upper Great Lakes region of North America: Hydroclimatic implications: DECADAL WATER LEVEL OSCILLATION, Geophys. Res. Lett., 41, 456–462, https://doi.org/10.1002/2013GL058679, 2014.

Yechieli, Y., Ronen, D., Berkowitz, B., Dershowitz, W. S., and Hadad, A.: Aquifer Characteristics Derived From the Interaction Between Water Levels of a Terminal Lake (Dead Sea) and an Adjacent Aquifer, Water Resour. Res., 31, 893–902, https://doi.org/10.1029/94WR03154, 1995.

**Referee 3**

In the manuscript titled "Recent ground thermo-hydrological changes in a Tibetan endorheic catchment and implications for lake level changes", the authors performed a physical land surface model and quantify thermo-hydrological changes in the Paiku catchment in the Qinghai-Tibet Plateau. This study contains some interesting findings and are valuable for the understanding of climate-driven ground thermal changes on hydrological cycle in alpine basin. However, the structure of the manuscript needs improvement. Therefore, a major revision is needed before this manuscript could be accepted for publication.

**We are grateful for the positive perception of our work.**

Major comments:

1. The results of section 4.1 of the manuscript show that glacial runoff exceeds land runoff, suggesting that the literature review of study on glaciers together with glacial runoff in the basin should be added to the introduction section. And how can you get the initial glacier ice volume for the simulation?

**We modified and extended the state of the art regarding glacier mass loss and glacier runoff over the QTP in the introduction with new references:**

**"*Overall glacier shrinkage has also been observed since the 1960s with a persistent increase in glacier mass loss rates (Bhattacharya et al., 2021; Hugonnet et al., 2021).*"**

**"*The majority of these lakes have experienced a pronounced increase in water levels since the 1990s (Lei et al., 2013, 2014), a trend that was suggested to be mainly driven by changes in precipitation and evaporation patterns (Yao et al., 2018) rather than by an increase in glacier mass loss and runoff (Brun et al., 2020; Zhang et al., 2021).*"**

**Additionally, a state of the art of glacier changes in the Paiku catchment is present in Sect. 3.2.6 (Quantification of glacier mass change). Following the reviewer's comment, to include glacial runoff, we added the following new sentences to this part:**

**"*Regarding glacial runoff, it was estimated to 320 ± 4 mm per year for the 2001-2010 period by Biskop et al. (2016) using a temperature-index approach for ice melt. For the 2000-2018 period, Zhang et al. (2020) derived a runoff value of 52 ± 12 mm per year (1.24 ± 0.29 $10^8$ $m^3$ per year that we scaled to the basin area). The value we derive of 39 ± 13 mm per year thus finds good consistency with the latter one (Sect. 4.1).*"**

**Regarding the initial glacier volume, we want to point out that our study does not include glacier simulation. Glacier volume change is derived from geodetic data. Glacier runoff is derived from the glacier volume change calculation and the precipitation from the climate forcing data. Concerning the initial volume of glacier ice, we did not need it because the volume change is directly derived from the difference in elevation change obtained from the DEMs and the area of the glacier.**

2. "Result" section:It is proposed that section 4.4 be merged into section 4.1.

**We followed the recommendation of the reviewer and merged section 4.4 into section 4.1. We do not reproduce the text here because it is an extensive and relatively straightforward change.**

3. "Discussion" section:The scenario experiment reveals the main findings and it is recommended to put the scenario experiment in the results section.

**We followed this recommendation, the results of this experiment are now presented in the new section 4.4 (Sensitivity test on evaporation and runoff). What we considered was discussion (and thus not relevant within the Results part) was kept within the discussion part (Sect. 5.3. Evaporation vs runoff and sensitivity to climate conditions).**

4. Section 5.3:It is suggested to add a table to give the value of runoff, evaporation, and precipitation in each permafrost region under two scenarios;

**We added the suggested table to this section (which is now section 4.4 after the modification from Major Comment 3).**

5. please explain Figure 11C specifically.

**We included in the new Section 4.4 the following explanations:**

**"*Figure 10C aggregates over the whole catchment this distribution of this precipitation input to the ground between runoff and evaporation for both scenarios. In between them, it also includes the distribution associated with the steady lake level scenario of Fig. 9C, which is based on the hypothesis listed as bullet points in Sect. 4.3 (climate forcing of the historical scenario, same glacier contribution, only land runoff increases).*"**

Minor comments

1.   Line 2-12 : You can summarize these sentences into 2-3 short sentences.

   **We shortened the first paragraph of the abstract to the following:**

   *"Climate change modifies the water and energy fluxes between the atmosphere and the surface in mountainous regions such as the Qinghai-Tibet Plateau (QTP), which has shown substantial hydrological changes over the last decades, including rapid lake level variations. The ground across the QTP hosts either permafrost or seasonally frozen and, in this environment, the ground thermal regime influences liquid water availability, evaporation and runoff. Therefore, climate-driven modifications of the ground thermal regime may contribute to lake level variations, yet this hypothesis has been relatively overlooked until now."*

2.   Line 14-15 : "We use TopoSCALE and TopoSUB to downscale ERA5 data and capture the spatial variability of the climate in our forcing data". This sentence is excessive.

   **We rephrased for:**

   *"We use TopoSCALE and TopoSUB to downscale ERA5 data, in an effort to account for the spatial variability of the climate in our forcing data."*

3.   The title has the "implications for lake level changes", but why does the abstract not reflect the results of the study on the water level?

   **We modified the abstract and from now on,  references to the lake level variations appear in 3 different places :**

   *"This study focuses on the cryo-hydrology of the catchment of Lake Paiku (Southern Tibet) for the 1980-2019 period. We use TopoSCALE and TopoSUB to downscale ERA5 data, in an effort to account for the spatial variability of the climate in our forcing data. We use a distributed setup of the CryoGrid community model (version 1.0) to quantify thermo-hydrological changes in the ground during this period. Forcing data and simulation outputs are validated with weather station data, surface temperature logger data and observations of lake level variations. Our lake budget reconstruction shows that the main water input to the lake is direct precipitation (310 mm per year), followed by glacier runoff (280 mm per year) and land runoff (180 mm per year). However, altogether these components do not offset evaporation (860 mm per year).*

   *Our results show that both seasonal frozen ground and permafrost have warmed (0.17 °C per decade 2 m deep), increasing the availability of liquid water in the ground and the duration of seasonal thaw. Correlations with annual values suggest that both phenomena promote evaporation and runoff. Yet, ground warming drives a strong increase in subsurface runoff, so that the runoff/(evaporation + runoff) ratio increases over time. This increase likely contributed to stabilizing the lake level decrease after 2010.*

   *Summer evaporation is an important energy sink and we find active layer deepening only where evaporation is limited. The presence of permafrost is found to promote evaporation at the expense of runoff, consistent with recent studies suggesting that a shallow active layer maintains higher water contents close to the surface. However, this relationship seems to be climate-dependent and we show that a colder and wetter climate produces the opposite effect. Although the present study was performed at catchment scale, we suggest that this ambivalent influence of permafrost may help to understand the contrasting lake level variations observed between the South and North of the QTP, opening new perspectives for future investigations."*

4.   Line 23-24 : "consistent with recent studies" is proposed to be excessive.

   **We added some precision to clarify which studies we were mentioning and thus avoid ambiguity:**

   *"The presence of permafrost is found to promote evaporation at the expense of runoff, consistent with recent studies suggesting that a shallow active layer maintains higher water contents close to the surface."*

5.   Line 148-150 : "It reached 4665 masl (85 m higher than the present level) prior to 25 ka BP and at the onset of the Holocene (11.9-9.5 ka BP)", references to the study should be marked.

   **The long term evolution of the lake level was spread over this sentence and the next one, which carries the bibliographic reference. But based on the suggestion from the reviewer, and to make things clearer, we merged the 2 sentences. The description of the long term evolution now reads:**

   *"Previous studies reported lake level fluctuations over different time scales. It reached 4665 masl (85 m higher than the present level) prior to 25 ka BP and at the onset of the Holocene (11.9-9.5 ka BP), afterwards, the lake shrank gradually (Wünnemann et al., 2015)."*

6.   Line 236: Please check the number "510-3", perhaps it is "0.005".

**We replaced 5 10⁻³ by 0.005.**

7. Figure 7(A): Please mark the start date.

**We thank the reviewer for noticing this typo. We corrected it and the gray bottom curve is now labeled "start date".**

8. Line 447: It is recommended that "AL thickness" be modified to "ALT".

**We did this modification. To be consistent, we also applied it to previous and following occurrences of the "active layer thickness".**

9. Line 644: It is suggested to change "AL" there to "Active Layer (AL)", and the abbreviation will be logical afterwards.

**We implemented this change.**

**References**

Bhattacharya, A., Bolch, T., Mukherjee, K., King, O., Menounos, B., Kapitsa, V., Neckel, N., Yang, W., and Yao, T.: High Mountain Asian glacier response to climate revealed by multi-temporal satellite observations since the 1960s, Nature Communications, 12, 4133, https://doi.org/10.1038/s41467-021-24180-y, 2021.

Biskop, S., Maussion, F., Krause, P., and Fink, M.: Differences in the water-balance components of four lakes in the southern-central Tibetan Plateau, Hydrology and Earth System Sciences, 20, 209–225, https://doi.org/10.5194/hess-20-209-2016, 2016.

Brun, F., Treichler, D., Shean, D., and Immerzeel, W. W.: Limited Contribution of Glacier Mass Loss to the Recent Increase in Tibetan Plateau Lake Volume, Frontiers in Earth Science, 8, 1–14, https://doi.org/10.3389/feart.2020.582060, 2020.

Hugonnet, R., McNabb, R., Berthier, E., Menounos, B., Nuth, C., Girod, L., Farinotti, D., Huss, M., Dussaillant, I., Brun, F., and Kääb, A.: Accelerated global glacier mass loss in the early twenty-first century, Nature, 592, 726–731, https://doi.org/10.1038/s41586-021-03436-z, 2021.

Wünnemann, B., Yan, D., and Ci, R.: Morphodynamics and lake level variations at Paiku Co, southern Tibetan Plateau, China, Geomorphology, 246, 489–501, https://doi.org/10.1016/j.geomorph.2015.07.007, 2015.

Zhang, G., Bolch, T., Chen, W., and Crétaux, J.-F.: Comprehensive estimation of lake volume changes on the Tibetan Plateau during 1976–2019 and basin-wide glacier contribution, Science of The Total Environment, 772, 145463, https://doi.org/10.1016/j.scitotenv.2021.145463, 2021.

Zhang, X., Wang, R., Yao, Z., and Liu, Z.: Variations in glacier volume and snow cover and their impact on lake storage in the Paiku Co Basin, in the Central Himalayas, Hydrological Processes, 34, 1920–1933, https://doi.org/10.1002/hyp.13703, 2020.

---

## Author Response (AR3)

**Referee 5**

The authors present a study aiming at the ground thermo-hydrological changes in a plateau lake. Overall, the topic is interesting and the authors provide detailed model construction and data analysis. This study is of great significance for the water level changes and water resource utilization of plateau lakes under climate change. But, I still have some concerns before considering for acceptance. Thus, this manuscript is subject to major revision.

**We are grateful to Referee 5 for the review we received and for the positive perception on the significance of the study. We present below our detailed answers to each of the discussed points.**

1. There are several questions regarding the modeling process of the lake water balance:

(1)In analyzing the lake water balance, the water input is considered to include direct precipitation, land surface and subsurface runoff, and glacier runoff. Does subsurface runoff include the recharge from permafrost/frozen soil thawing?

**Yes, recharge from permafrost is included. In the model, when frozen water in the ground melts, it becomes available for subsurface runoff if the liquid water content exceeds the field capacity. This is true for places with permafrost and with seasonally frozen ground. To clarify this point we added the following line to Sect. 3.2.4. (model setup and validation):**

**"Because the model couples thermal and hydrological fluxes, all of these changes in the soil water content can be driven by precipitation input, evaporation but also water phase change in the ground such as ice melt."**

And in the model construction (Line 191), the important part of the subsurface runoff was ignored.

**Our intention was to merge surface and subsurface runoff in a variable that we called $Runoff_{Land}$ but we see it leads to misinterpretation. So now, to solve this problem, we split both terms in the equation:**

**"$\Delta zLake = Precipitation_{Lake} + Runoff_{LandSurf} + Runoff_{LandSub} + Runoff_{Glacier} - Evaporation_{Lake}$"**

Additionally, considering that the lake is located in a closed basin of surface water systems, the water output is only considered to be evaporation. However, does the lake have any water exchange with underground aquifers?

**We are aware of the possibility of water flow between the lake and a surrounding aquifer. We believe that attempting to close the hydrological budget of such a catchment, that present glaciers, snowfall, rainfall permafrost, seasonally frozen ground and a lake is very challenging. Therefore, as modelers, we have to choose the processes we want to represent, based on the scientific questions we want to tackle and based on our ability to quantify these processes. For this reason, as we explained in Sect. 3.2.1. (Conceptual hydrological model for the catchment) we cannot quantify such a flow and we had no choice but to ignore it in our calculations. Yet because we know it can impact our results, we include an extensive paragraph on the topic in the discussion. To clarify this point, we modified it. It now reads:**

**"Additionally, our approach ignores potential water fluxes between the lake and a surrounding aquifer. This can be a possible reason for this mismatch. In the context of a decreasing lake level, an aquifer surrounding the lake can create an additional water inflow when the lake level passes below the piezometric level of the aquifer (Yechieli et al., 1995). We suggest that such an inflow could mitigate the lake level decrease and thus explain the missing water in our reconstruction (Fig. 6B). It could also explain the gradual stabilization of the lake level that our model does not reproduce. This flow is not part of our conceptual hydrological framework even though it likely exists in reality, especially since there is no permafrost near the lake (as we simulate it here), allowing for the existence of such an aquifer (Walvoord and Kurylyk, 2016). Groundwater has been identified as a potential contributor to lake level rise in other regions of the QTP (Lei et al., 2022). In the long run, lake-aquifer systems commonly follow oscillations of the net atmospheric flux of water (Precipitation – Evaporation) and of the runoff that forces its mass balance (Watras et al., 2014). During these oscillations, the lake can "pump" water from the aquifer or feed it depending on the relative difference of piezometric level between them (Almendinger, 1990; Liefert et al., 2018). Yet, this potential effect is difficult to account for and its magnitude remains unclear. Therefore, the reasons for the mismatch between observed and simulated lake levels could also be connected to other aspects of our methodology such as bias in the climatic forcing data and other shortcomings arising from the lack of field data, or hydrological processes, as developed in Sect. 5.1.1. and 5.1.2."**

(2)This study focuses on the impact of ground thermal processes on hydrological processes. From the conceptual diagram in Figure 2, it can be seen that this study considers the process of glacier melting, but does not take into account the process of permafrost/frozen soil thawing, which is also sensitive to temperature.

**We regret a misunderstanding here, as stated above, our model couples heat and water flows, therefore the relevant processes to account for potential hydrological impacts of thermal changes in permafrost and/or seasonally frozen ground are represented in our approach. See further details on this point below.**

On the one hand, permafrost/frozen soil thawing can provide water,

**Our model accounts for this effect. As stated earlier, when frozen water in the ground melts, it becomes available for subsurface runoff if the liquid water content exceeds the field capacity, regardless if this happens in permafrost or in seasonally frozen ground.**

**Seasonally frozen ground is fully thawed in summer, therefore its ice stock is fully depleted each year according to a seasonal pattern. In this condition, it cannot represent a long term water stock that ground warming would empty year after year to fill the lake. This is different in the case of permafrost. In a catchment with important permafrost disappearance and active layer deepening over the years, permafrost could indeed represent a long term water input to the lake. Yet, as presented in Sect. 4.2., Active Layer deepening is only observed for 44% of the warm permafrost simulations, which represents in the end only 8% of the total land area (44% of 19%, Table 1). Additionally, this AL deepening trend is rather weak (5.2.1.). Therefore, even under the assumption that the thawing permafrost layer is ice saturated, the water release from this process would be several orders of magnitude below the flows affecting the lake balance, as we quantify them in Sect. 4.1.2.**

**In the case of disappearing permafrost, the AL already neighbors or exceeds 2m at the beginning of the simulations in 1980 (0 to 2 m is the hydrologically active ground layer in our setup), therefore we do not derive any significant flow from this process.**

And on the other hand, it can change soil permeability and the connectivity between lakes and groundwater.

**Since our model couples heat and water flows and account for phase changes, it therefore provides a relevant and consistent representation of cryohydrological changes in the ground as demonstrated in Sect. 5.3.1. and Fig. 11. As such, parameters such as hydraulic conductivity, which is used to calculate the subsurface runoff, accounts for changes of the thermal regime. We here quote an example from Sect. 3.2.3. (The CryoGrid community model (version 1.0)).**

*"Additionally, to represent the obstruction of connected porosity by ice formation, the hydraulic conductivity is reduced by a factor dependent on the local ice content, following Dall'Amico et al. (2011)."*

**Regarding lake-aquifer interactions, we developed our take on this question earlier in our answer to referee's point 1.(1). Additionally, we want to point out that there is no permafrost near the lake and that large stores of liquid groundwater are unlikely in higher elevations areas of the catchment.**

Thus, can the water flows between the lake and potential aquifers be ignored, especially in the context of increasing temperatures and significant melting of frozen soil? Would this portion of the error be significant?

**See the answer to 1.(1).**

(3)The study conducted a one-dimensional vertical simulation of water and heat processes. Is the impact of lateral water and heat transfer ignored?

**Indeed, as described in Sect. 3.2.4., there are neither heat nor water fluxes between the land simulations. In this study we tackle complex questions involving numerous physical processes (surface energy fluxes, infiltration, heat conduction, phase change, soil suction…), we therefore have to make compromises regarding the processes we represent. Regarding lateral heat fluxes, we want to point out that they are most likely insignificant in the near surface, especially when compared to the vertical heat fluxes.**

**Additionally, because we want to capture the important spatial variability of the climate in a high mountain catchment (spanning vertically over 2400 m), we chose to work with downscaled and clustered climatic data (100 x 100 m resolution). This setup limits our ability to represent lateral water fluxes between the simulation units and we see it as a necessary compromise to carry out our approach and preserve its strengths. Indeed, a different framework with larger and interconnected response units would allow to derive these water fluxes but would not allow to map the cryological types of ground as we did it. Yet, we show that each cryological type of ground has a very specific behavior.**

**Therefore, we think it is important to evaluate our work in the frame of the processes we want to investigate and against what is doable today with other approaches. In this regard, we are confident that our approach is part of the state of the art and one of the most complete effort to include the variety of coupled**

climatological, surface and subsurface processes characterizing the climate, hydrology and ground thermal regime of high-mountain catchments in Tibet at a small scale with a high spatial resolution (Sect. 5.3.1.). Yet we are aware of the limitations brought by the lack of water routing. To clarify this point, we rephrased the part of the Discussion (Sect. 5.1.2. Modeling strategy ) tackling this point. It now reads:

*"A limitation in our study is that lateral water flows between land simulation units is ignored. By giving access to the timing of water transport across the catchment, water routing would allow to investigate temporal hydrological patterns at a monthly or seasonal scale. Because we work at annual and decadal time scales, this limitation has limited consequences on our results. The main consequence is to ignore potential storage effects on the land that would delay the arrival of runoff to the lake. We suggest that it is possible that this limitation partly explains the limited match between computed and required runoff at the annual time scale (Fig. 5). Yet, our subdivision of the catchment based on the different cryological states of the ground allows us to work with hydrological units that are smaller than the catchment and thus present shorter hydrological response time to precipitation."*

Additionally, all 368 simulations are independent and use the same parameterization. Considering the large range of 2400 km2, the same vertical distribution of soil condition in different elevations, mountain hillslopes - valleys, and under different land uses and vegetation cover, may introduce significant errors.

Indeed, because of the absence of specific datasets for the basin or even borehole observations, all the simulations use the same setup. To clarify this methodological choices and explain how we adapted to data scarcity, we now added the following consideration in the Discussion in Sect. 5.1.1. (Data usage within the conceptual framework and data scarcity):

*"Additionally, in absence of borehole data that would allow us to anchor our parameters into observations, we rely on gridded values designed for hydrological and/or land surface modeling (Sect. 3.2.4. and Appendix A). Because these values might be less reliable than field observations, we chose to average them over the catchment to derive some more robust values."*

Regarding the potential impact of vegetation, we present below a photo to show the type of land cover we found on the field that supports our approach. As we precise it in Sect. 5.1.2.:

*"…since vegetation is extremely scarce in the Paiku catchment, which is largely dominated by barren lands, we suggest that this would not significantly affect our results. However, this limitation should be explored in future field and modeling studies."*

[Figure]

**Figure R5.1 Ground photo in the Paiku catchment (Credit: Fanny Brun).**

We are aware that the limitations of the model approach can influence our results and we acknowledge it in the discussion, in Sect. 5.1.1.:

*"Altogether, this scarcity of field observations is likely to bring significant uncertainties to our analysis. Future efforts should focus on acquiring additional data or developing validation methods based on remotely sensed observations."*

2. The research topic mentioned "implications for lake level changes", so what are the specific trend and influence factors of lake water level changes? What is the reason for the transformation from rapid lake shrinkage to relative stable stage in recent years? What are the impacts on the lake level from water and heat changes? The authors could further summarize the answers to these questions to help readers more intuitively understand the impact of water and heat processes on lake water level changes.

We understand the reviewer's point and, in combination with comment number 3, we significantly modified the structure of the Discussion. Within this new structure, the Discussion now has a Sect. 5.4. called *Implication for lake level changes* that gathers our findings on the question. We reproduce it below:

*"    5.4        Implications for lake level changes*

*At the scale of the Paiku catchment and in regard of lake level variations, the results we present highlight that:*

- *The sum of the direct precipitation in the lake, the land runoff and the glacier runoff are not enough to compensate for the lake evaporation over the study period, hence driving the observed lake level decrease.*
- *Long-term hydrological trends in the catchment are led by trends in climate; and precipitation increase, jointly with glacier melt, provides enough water to drive a concomitant increase of runoff and evaporation.*
- *Ground thermal changes increase the distribution of liquid vs. frozen water in the ground and the duration of seasonal thaw, correlations suggest that these modifications increase evaporation. The warming of the ground is also related to the increase of subsurface runoff towards the lake.*
- *Ground warming and permafrost thawing promote subsurface runoff over time, contributing to an increase in the runoff/evaporation ratio of the catchment.*
- *Over the 40 years we studied, the presence of permafrost seems to promote evaporation at the expense of runoff. Yet this trend appears to be climate-dependent and the cryological state of the ground might shift the runoff/evaporation distribution in the other direction under colder and wetter climates.*

*At the scale of the QTP, these results have several implications. First, a better understanding of the recent and future lake level variations will come with a better knowledge of spatial patterns and temporal trends in precipitation. Second, climate changes are modifying the ground thermal regime of Tibetan catchments. Ground warming may lead to active layer deepening, permafrost disappearance and/or changes in the seasonal freeze/thaw cycles, affecting evaporation, runoff volumes and pathways and overall, changing the hydrological functioning of Tibetan catchments (and the waterflow provided to the lakes). Finally, the effect of permafrost on the distribution between evaporation and runoff seems to be dependent on the climate settings and the permafrost coverage of the catchment. Further studies should investigate this phenomenon and how it might contribute to explaining the contrasting lake level evolutions across the QTP."*

3. The paper structure can be improved, not only just presenting simulation results, but also highlighting specific targeted questions. For example, in the discussion section, the key points of the study are not highlighted, and only the various aspects of the results including data, model, permafrost changes, ground temperature, evaporation, and runoff changes, are listed. What is the main focus of this study? How are water and heat related and how do they affect lake water level changes? I think the structure of the discussion section could be more focused.

**We understand the reviewer's concern and see how the many changes we have already made to the discussion have probably weakened its structure. To correct this point and answer the reviewer's comment, we have now fully restructured the Discussion as follows:**

**5. Discussion**

    **5.1. Limitation and potential of the approach**

        **5.1.1. Data usage within the conceptual framework and data scarcity**

        **5.1.2. Modeling strategy**

    **5.2. Trends in the catchment and across the QTP**

        **5.2.1.    Lake hydrological budget and level variations**

        **5.2.2.    Permafrost and ground temperature trends**

        **5.2.3.    Evaporation and runoff trends**

    **5.3. Cryo-hydrological couplings at catchment scale and implication for lake level variations**

        **5.3.1.    Interdependence of thermal and hydrological variables**

        **5.3.2.    Influence of the ground thermal regime on the distribution between runoff and evaporation**

    **5.4. Implications for lake level changes over the QTP**

Here, some detailed comments are listed:

(1)Line6: "Therefore, climate-driven modifications of the ground thermal regime may contribute to lake level variations, yet this hypothesis has been relatively overlooked until now.". Previous studies have already conducted a lot of relevant research. The most important thing is that this study did not effectively reveal how the ground thermal regime affects the lake water level. The author should further revise the manuscript.

**To account for the reviewer's comment and keep the abstract short, we modified it as follows:**

*"Consequently, climate-induced changes in the ground thermal regime may contribute to variations in lake levels, but the validity of this hypothesis has yet to be established."*

(2)Line58: "Whereas the northern and central QTP have recorded lake expansion, the southern parts of the plateau have experienced lake shrinkage". What is the cause of lake shrinkage? After all, most lakes on the Qinghai-Tibet Plateau experience water level increases, while the lake in this study experienced water level decreases. Therefore, it is worthwhile for the author to further summarize existing research findings and take their influence into account when explaining lake shrinkage in their writing.

**We now added the following paragraph to the introduction :**

*"Shrinking lakes have received less attention in the literature than rising lakes because they are fewer. For this reason the drivers of this shrinkage are still unclear. Qiao et al. (2019) reported that recent lake shrinkage over the QTP could be driven by local precipitation decrease and/or evaporation increase (in relation to air temperature increase). Zhang et al. (2020a) suggests that the divergent trends in lake level variations across the QTP could be linked to the contrasting evolution of moisture transport between the north and south of the plateau. On longer timescales, lake shrinkage over the QTP during the Holocene seems to be related to variations in the intensity of the Asian monsoon (Chen et al., 2013). Overall, such a complex pattern of rising and shrinking lakes challenges our understanding of the hydrological changes occurring in these high Asian watersheds."*

(3)Line115: " We show the interplay in the water and energy fluxes occurring between the atmosphere, the surface and the subsurface and discuss their impact on the hydrology of the catchment and their implication regarding lake level variations.". What are the factors that affect lake level changes, and how can the interaction between lakes and groundwater be considered?

**This line closes the Introduction and thus aims at giving an overview of the rest of study. The lake budget and its positive and negative terms are presented in Sect. 4.1.2. and Discussed in Sect. 5.2.1., including a discussion paragraph on lake-groundwater interactions (see answer to Reviewer's point 1.(1)).**

(4)Line149: "Because it is hydrologically closed, the lake mainly loses water through evaporation". The basin is hydrologically closed in surface water system does not mean that the underground is also closed. Whether there is water exchange between lakes and the underground, and whether there is seepage of lake water?

**As stated above, lake-aquifer interactions are ignored by our approach. This is described in the methodology (Sect. 3.2.1.), discussed in the Discussion (Sect. 5.2.1.) and our argument to do so are described in the answer to Reviewer's point 1.(1).**

(5)Line277: "see section 3.2.2". The full form (section) and abbreviation (sect.) of the same word should be consistent throughout the text.

**Done. All references to a given section are now written "Sect.".**

(6)Line288: "infiltration according to Richards equation only occurs in the top and bottom soil units". Does this study only consider the infiltration to the depth of 2m, and is the result reliable?

**As mentioned in the response to Reviewer's point 1.(3), no observations/boreholes data are available for the catchment. Therefore, the only dataset available for modeling are large scale gridded datasets that we decide to average for the sake of simplicity and robustness. As stated in the study (Sect. 3.2.4. Model setup and validation):**

*"In absence of direct observation of the soil stratigraphy within the catchment, the soil column was designed to agree with field observations in the region (Yuan et al., 2020; Wang et al., 2009; Hu et al., 2020; Luo et al., 2020; Yang et al., 2014b; Wang et al., 2008), to be consistent with similar modeling approaches*

*across Tibet (Chen et al., 2018; Song et al., 2020) and to be consistent with input datasets (Shangguan et al., 2017, 2013)."*

(7)Line431: "(Sect. 4.1.1.)". The format should be consistent: "4.1.1." or "4.1.1"

**Done. All references to a given section now have a numbering finishing with a dot.**

(8)Line432: " This pattern of a too strong decrease followed by an increase is consistent with the comparison between simulated and required runoff presented on Fig. 5D. ". Lake water level variation is a key focus of this study, while there are significant errors in simulating lake water levels. What is the source of these errors?

**We understand that, when looking at lake levels, the difference may look significant but we believe that this take on our results exacerbates the mismatch. Indeed, when we look at it rather in terms of lake hydrological budget over the 40 years, the missing water input to the lake (from land and glaciers) that leads us to overestimate the lake decrease only corresponds to a 5% error on the total water input (Sect. 4.1.2., Fig. Fig. 6A, red line). In the framework of a study that attempts to couple glacier mass change, land thermo-hydrological changes and lake level changes altogether, with very limited available data, we find the agreement acceptable.**

**Following our approach, this mismatch can arise from bias (i) in the forcing data (and mainly in the precipitation) used for the land and lake simulations, (ii) in the glacier mass balance estimate and/or (iii) in the quantification of hydrological processes for the land or for the lake (evaporation, runoff). For all these variables, we believe that we provide the best possible estimates because we use all the available data that could possibly contribute to correct and validate them.**

Especially after 2005, there is a trend difference that may require further improvement in the simulation process. Could the rapid decrease in lake water level during the early stages of the simulation be related to the ignorance of the water source provided by the melting of frozen soil?

**There are two aspects of our methodology that can limit our ability to accurately match the observed lake level trends. Firstly, our estimates of glacier mass changes correspond to two time averages, one for the 1980-2000 period and one for the 2000-2020 period. Therefore our values of glacier runoff are very smooth and a different distribution of the same amount of water towards the lake over the 40 years (i.e. more before 2010 and less after) could limit the discrepancy between the observed and simulated lake levels.**

**Secondly, the absence of water routing also limits the realism of the lake level variations. The routing could induce delays or storage effects on the land runoff towards the lake that would change the year to year variation of the lake level. Discussions on the absence of water routing in our study are developed in the Discussion (Sect. 5.1.2.) and higher in this response.**

**Thanks to the reviewer's comment, we now see that these different elements on the mismatch between the simulated and observed lake levels were not gathered in one paragraph but scattered in the Discussion. Therefore, we modified the first paragraph of Sect. 5.2.1. (Lake hydrological budget and level variations) to include these considerations:**

**"*The reason for the overall mismatch of 1.06 m can arise from bias (i) in the forcing data (and mainly in the precipitation) used for the land and lake simulations, (ii) in the glacier mass balance estimate and/or (iii) in the quantification of hydrological processes for the land or for the lake (evaporation, runoff). On top of these potential biases, the difference in trends for the end of the simulation time can be influenced by (i) our estimates of glacier mass changes, which are made of two time averages (one for the 1980-2000 period and one for the 2000-2020 period) and therefore produce very smoothed glacier runoff values that cannot capture variations at the scale of the decade of less and (ii) the absence of water routing that prevent us from accounting for delays of storage effects on the water supply from the land to the lake.*"**

(9)Line442: "Disappearing permafrost". This study simulated the areas where permafrost will disappear. What are the characteristics of these places, including topographic factors, precipitation and temperature factors, or other reasons that cause these areas to disappear earlier than other place under the same warming conditions? This is of significance for studying the distribution of permafrost.

**As it is visible on Fig. 7A and in Tab. 1, elevation (and the climate variability it creates) is the main factor affecting the distribution of disappearing permafrost. Indeed, the mean elevation of disappearing permafrost is 200 m lower than the one of warm permafrost and 600 lower than the one of cold permafrost (Tab. 1). As such, it always occurs at the boundary between permafrost and permafrost free areas (i.e., the lowest possible place). As stated in Sect. 4.2. (Ground thermal results):**

*"Permafrost disappearance (grey zones in Fig. 7D) mainly happens for low-lying permafrost of the south and the center of the catchment. It occurs for the most part on the outer slopes of the permafrost regions and at the bottom of steep glacial valleys."*

**Otherwise, regarding the information we make available, we remind that for each cryological type of ground, including disappearing permafrost, we provide in the study:**

- **In Tab. 1: % of catchment area, elevation, slope**
- **In Tab. 2: Precipitation input, runoff, evaporation**
- **in Fig. 7A: spatial extent and distribution within the catchment**
- **in Fig. 7B: yearly mean annual temperature at 2m deep**

**Specifically for disappearing permafrost we also provide:**

- **in Fig. 11D: the relationship between ground warming and subsurface runoff**

(10)Line461: What is the mean of MAGT in Figure 7.

**MAGT stands for *Mean Annual Ground Temperature*, we modified the caption of Fig. 7 so that it now explains this acronym.**

(11)Line494: "we distinguished ALT for locations experiencing an average evaporation lower or higher than 150 mm per year during the simulations". Line 509 shows an average evaporation of 180 mm per year, how was the value of 150 mm per year determined?

**The choice of this value is based on the relationship between evaporation and AL deepening as it can be seen on Fig. 11. This value marks a threshold among warm permafrost simulations, between those with a clear deepening trend and the others. We now clarified this point by adding to this paragraph the sentence:**

*"This threshold value of 150 mm per year is based on further investigations on the relationships between evaporation and ALT provided in Sect. 5.3.1."*

(12)Line495: "evaporation below 150 mm per year record an active layer deepening trend". In areas with low evaporation, the active layer thickens, while in areas with high evaporation, which should have more heat and more melting, the active layer becomes shallower. What is the reason for this?

**This point is discussed in Sect. 5.2.2.:**

*"Active layer (AL) evolution is contrasting throughout the catchment and a deepening signal is only visible for the locations with limited evaporation (<150 mm per year). Given the strong drive of summer climate on ALT, this overall lack of a deepening trend highlights how evaporation can act as an energy intake at the surface (Yang et al., 2014a), limiting the surface and subsurface heat fluxes and thus AL deepening. In this regard, our results fall in line with the conclusions of Fisher et al. (2016)…"*

(13)Line519: "We also present the catchment average of the runoff / (runoff + evaporation) ratio (Fig. 9C), which is equivalent to runoff / (rain + snow – snow sublimation) given the negligible contribution of soil storage variations. ". As glaciers and permafrost melt, does the soil water storage change? Line 538 suggests an increase in unfrozen water content in the soil, is this assumption valid?

**Firstly, because there is no water routing in our approach, glacier runoff cannot increase soil water storage. Glacier runoff is directly accounted as a water flux towards the lake. Secondly, as we explained it in detail in the response to reviewer's point 1.(2), liquid water input from active layer deepening or disappearing permafrost is negligible compared to any other water flux in the catchment and does not have the power to impact soil water storage. Finally, line 538 goes together with Fig. 9D, which shows the annual proportion of liquid / total water averaged for the whole catchment. Therefore it is a ratio which does not indicate an increase in the absolute quantity of water in the soil per year but rather an increase in the time the water spends unfrozen in the ground compared to frozen. To clarify this point we rephrased this line to:**

*"The graph shows that the proportion of liquid water in the total water content increases at around +1.41% per decade (p=1×10⁻⁴), indicating that water spends more and more time in the ground in a liquid form, being thus increasingly available for hydrological processes such as evaporation or runoff."*

(14)Line 540: "Figure 9. Hydrological results. A: Annual evaporation averaged over the whole catchment. ". Over the whole catchment, is this correct? Land area only?

**We thank the reviewer for pointing out this unprecise formulation. The results indeed concern the land area of the catchment. We added the mention (landed area only) in the caption to clarify this point.**

(15)Line547: "4.4. Sensitivity test on evaporation and runoff". This study adds a simulation result comparison by adding one scenario that cannot be referred to as sensitivity analysis.

**To be precise, the wording "sensitivity analysis" is nowhere to be found in our study. We initially referred to a "sensitivity test" because we believe that a test is a smaller thing than an analysis. Yet, to cut short discussions revolving around the usage of a single word, Sect. 4.4. is now called "Sensitivity of evaporation and runoff". We also added the following lines to this section:**

*"We conducted a simple sensitivity test on the climatic conditions (i.e. not a full-scale sensitivity test)."*

(16)Line556: "Figure 10C aggregates…". Figure or Fig. needs to be consistent.

**Based on this comment, we now apply consistently the following rule:**

- **Figure captions are written "Figure"**
- **Mentions to figures are written "Fig."**

(17)Line584: "Table 2. Distribution of between runoff and evaporation for the 2 scenarios ". The table caption should be corrected.

**We removed "of".**

(18)Line633: "Gao et al., n.d.". Format correction and corresponding reference format.

**Reference updated.**

(19)Line674: "Such an inflow could mitigate the lake level decrease and thus explain the missing water in our reconstruction (Fig. 6B). It could also explain the gradual stabilization of the lake level that our model does not reproduce.". This explanation is too speculative.

**We did not come up with this hypothesis. It is based on the literature review we did to document this section of the discussion. Yet, to account for the reviewer's comment, we added some more cautious wording. The full paragraph now reads as follow:**

*"In the context of a decreasing lake level, an aquifer surrounding the lake can create an additional water inflow when the lake level passes below the piezometric level of the aquifer (Yechieli et al., 1995). We suggest that such an inflow could mitigate the lake level decrease and thus explain the missing water in our reconstruction (Fig. 6B). It could also explain the gradual stabilization of the lake level that our model does not reproduce. [...] In the long run, lake-aquifer systems commonly follow oscillations of the net atmospheric flux of water (Precipitation – Evaporation) and of the runoff that forces its mass balance (Watras et al., 2014). During these oscillations, the lake can "pump" water from the aquifer or feed it depending on the relative difference of piezometric level between them (Almendinger, 1990; Liefert et al., 2018)."*

(20)Line 782: "Given the strong drive of summer climate on Active Layer Thickness (ALT), ". The previous text has already provided the abbreviation.

**We replaced "Active Layer Thickness (ALT)" by "ALT".**

(21)Line834: "5.3. Evaporation vs runoff and sensitivity to climate conditions". From the view of title, is it similar content with 5.2.2 Evaporation and runoff changes?

**Titles of the Discussion have been significantly modified to follow comment 3 from the reviewer. We believe the new ones provide a clearer understanding of the content of each subsection.**

(22)Line883: "hence driving the observed lake level decrease.". Is the reason for the lake water level decline really due to high evaporation? So why do most lakes on the Tibetan Plateau still experience rising water levels against the backdrop of rising temperatures?

**We state here the results of our quantification of the lake hydrological budget (Sect. 4.1.2.). According to our reconstruction, and despite the increasing trend in precipitation over the catchment, evaporation exceeds the sum of direct precipitation in the lake and land and glacier runoff. Unfortunately our catchment scale study does not allow us to derive firm conclusions for the rest of the Tibetan plateau. Each catchment**

**has indeed its own characteristics, and from one catchment to another many factors can vary such as elevation, climate (and in particular precipitation and temperature), glacier coverage, lake/land area ratio, evaporation… As stated earlier, much is still to be understood about the spatial distribution of rising and shrinking lakes over the QTP. Yet, thanks to our results linking climate, permafrost coverage and the partition between evaporation and runoff (Sect. 4.4.) we believe our study indicates a worthy direction for future investigations on the topic.**

(23)Line891: "Ground warming and permafrost thawing promote subsurface runoff over time, contributing to an increase in the runoff/evaporation ratio of the catchment.". The impact on runoff is greater than evaporation, why does the lake level decrease? As mentioned earlier, it is due to increased evaporation under climate warming, which leads to a decrease in lake water level.

**Indeed, we report trends on land runoff and land runoff/evaporation ratio showing that more and more water reaches the lake over time. This likely explains the gradual stabilization of the lake level decrease. Yet, these trends are not enough to overturn the dominance of lake evaporation over all the input variables of the lake budget.**

(24)The authors should carefully check the format of cited references and the reference lists. Such as the abbreviation of Journal of Geophysical Research: Atmospheres, IEEE Geoscience and Remote Sensing Letters, Vadose Zone Journal... in the reference list.

**Our reference list is formatted with a bibliography software. We have updated the reference list which should now fit the formatting of HESS.**

**References**

[revised manuscript text omitted]